# GEOMETRY INFORMED TOKENIZATION OF MOLECULES FOR LANGUAGE MODEL GENERATION

## ABSTRACT

We consider molecule generation in 3D space using language models (LMs), which requires discrete tokenization of 3D molecular geometries. Although tokenization of molecular graphs exists, that for 3D geometries is largely unexplored. Here, we attempt to bridge this gap by proposing the Geo2Seq, which converts molecular geometries into $SE(3)$-invariant 1D discrete sequences. Geo2Seq consists of canonical labeling and invariant spherical representation steps, which together maintain geometric and atomic fidelity in a format conducive to LMs. Our experiments show that, when coupled with Geo2Seq, various LMs excel in molecular geometry generation, especially in controlled generation tasks.

## 1 INTRODUCTION

The generation of novel molecules with desired properties is an important step in drug discovery. Specifically, the design of three-dimensional (3D) molecular geometries is particularly important because 3D information plays a critical role in determining many molecular properties. Different generative models have been used for 3D molecule generation. Early studies such as G-SchNet (Gebauer et al., 2019) use autoregressive generative models to generate 3D molecules by sequentially placing atoms in 3D space. It was observed that these models often yield results with low chemical validity. Recently, diffusion models (Hoogeboom et al., 2022; Xu et al., 2023a) achieve better performance in 3D molecule generation tasks. However, they typically need thousands of diffusion steps, resulting in long generation time.

Language models (LMs) (Vaswani et al., 2017; Devlin et al., 2018; Brown et al., 2020; Gu et al., 2021), with their streamlined data processing and powerful generation capabilities, have shown success across various domains, particularly in natural language processing (NLP). Recently, large language models (LLMs) (Zhao et al., 2023b) show extraordinary capabilities in learning complex patterns (Zhang et al., 2024) and generating meaningful outputs (Touvron et al., 2023; Achiam et al., 2023; Chowdhery et al., 2023). Despite their potential, the application of LLMs to the direct generation of 3D molecules is largely under-explored. This is primarily due to the fact that geometric graph structures of molecular data are fundamentally different from texts. However, 3D geometric information is crucial in molecular tasks, since different conformations of the same molecule topology have different properties, such as per-atom forces. This gap reveals a unique challenge of how to make use of the powerful pattern recognition and generative capabilities of LLMs to handle complicated molecular graph structures, especially geometries. On the other hand, solutions to this challenge with model-level modifications cannot effectively leverage the rapidly developing power of LMs. These solutions require specific module designs, which needs to be done separately for each LM architecture and can be infeasible for modern LMs released via APIs.

In this work, we bridge this gap by applying LMs to the task of 3D molecule generation. We employ a novel approach translating the intricate geometry of molecules into a format that can be effectively processed by LMs. This is achieved by our proposed tokenization method Geo2Seq, which converts 3D molecular structures into $SE(3)$-invariant one-dimensional (1D) discrete sequences. The transformation is based on canonical labeling, which allows dimension reduction with no information loss outside graph isomorphism groups, and invariant spherical representations, which guarantees $SE(3)$-invariance under the equivariant global frame. By doing so, we harness the advanced sequence-processing capabilities and efficiency of LMs while retaining essential geometric and atomic information. Note that since Geo2Seq operates solely on input data, our method is

agnostic to the subsequent LMs used. and can seamlessly adapt to any state-of-the-art sequence model, maximizing LM capabilities while avoiding additional architecture design or redundant computations. When combined with powerful modern LLMs, Geo2Seq can achieve highly accurate modeling of 3D molecular structures. In addition, Geo2Seq can benefit conditional generation by including real-world chemical properties in sequences because modern LLMs are capable of capturing long-context correlations to comprehend global structure and information in sequences. Our experimental results demonstrate these advantages. We show that using different LMs with Geo2Seq can reliably produce valid and diverse 3D molecules and outperform the strong diffusion-based baselines by a large margin in conditional generation. These results validate the feasibility of using LMs for 3D molecule generation and highlight their potential to aid in the discovery and design of new molecules, paving the way for applications such as drug development and material science.

## 2 PRELIMINARIES AND RELATED WORK

### 2.1 3D MOLECULE GENERATION

In this work, we study the problem of generating 3D molecules from scratch. Note that this problem is different from the 3D molecular conformation generation problem studied in the literature (Mansimov et al., 2019; Simm & Hernandez-Lobato, 2020; Gogineni et al., 2020; Xu et al., 2021a;b; Shi et al., 2021; Ganea et al., 2021; Xu et al., 2022; Jing et al., 2022), where 3D molecular conformations are generated from 2D molecular graphs. We represent a 3D molecule with $n$ atoms in the form of a 3D point cloud (*i.e.*, a set of points with different positions in 3D Euclidean space) as $G = (\boldsymbol{z}, \boldsymbol{R})$. Here, $\boldsymbol{z} = [z_1, \cdots, z_n] \in \mathbb{Z}^n$ is the atom type vector where $z_i$ is the atomic number (nuclear charge number) of the $i$-th atom, and $\boldsymbol{R} = [\boldsymbol{r}_1, \cdots, \boldsymbol{r}_n] \in \mathbb{R}^{3 \times n}$ is the atom coordinate matrix, where $\boldsymbol{r}_i$ is the 3D coordinate of the $i$-th atom. Note that 3D atom coordinates $\boldsymbol{R}$ are commonly called 3D molecular conformations or geometries in chemistry. We aim to solve the following two generation tasks in this work:

- **Random generation.** Given a 3D molecule dataset $\mathcal{G} = \{G_j\}_{j=1}^m$, we aim to learn an unconditional generative model $p_\theta(\cdot)$ on $\mathcal{G}$ so that the model can generate valid and diverse 3D molecules.

- **Controllable generation.** Given a 3D molecule dataset $\mathcal{G} = \{(G_j, s_j)\}_{j=1}^m$ where $s_j$ is a certain property value of $G_j$, we aim to learn a conditional generative model $p_\theta(\cdot|s)$ on $\mathcal{G}$ so that for a given $s$, the model can generate 3D molecules whose quantum property values are $s$. The equivalent task is also known as "conditional generation", while in this work we follow Hoogeboom et al. (2022) to use the term "controllable generation".

A major technical challenge of 3D molecule generation lies in maintaining invariant to $SE(3)$ transformations, including rotation and translation. In other words, ideal models should assign the same probability to $G = (\boldsymbol{z}, \boldsymbol{R})$ and $G' = (\boldsymbol{z}, \boldsymbol{R}')$ if $\boldsymbol{R}' = \boldsymbol{Q}\boldsymbol{R} + \boldsymbol{b}\mathbf{1}^T$, where $\mathbf{1}$ is an $n$-dimensional vector whose elements are all one, $\boldsymbol{b} \in \mathbb{R}^3$ is an arbitrary translation vector, and $\boldsymbol{Q} \in \mathbb{R}^{3 \times 3}$ is a rotation matrix satisfying $\boldsymbol{Q}\boldsymbol{Q}^T = \boldsymbol{I}, |\boldsymbol{Q}| = 1$. To achieve $SE(3)$-invariance in 3D molecule generation, existing studies have proposed various strategies. Early studies propose to generate 3D atom positions by $SE(3)$-invariant features, such as interatomic distances, angles and torsion angles. They construct 3D molecular structures through either atom-by-atom generation (Gebauer et al., 2019; Luo & Ji, 2022a) or generating full distance matrices (Hoffmann & Noé, 2019) in one shot. Recently, more and more studies have applied generative models to generate 3D atom coordinate directly. These studies include E-NFs (Satorras et al., 2021a) and EDM (Hoogeboom et al., 2022), which combine equivariant atom coordinate alignment process with equivariant EGNN (Satorras et al., 2021b) model for 3D molecule generation. Following EDM, many other studies have proposed to improve diffusion-based 3D molecule generation frameworks by stochastic differential equation (SDE) based diffusion models (Wu et al., 2022; Bao et al., 2023) or latent diffusion models (Xu et al., 2023a). Besides, some recent studies (Qiang et al., 2023) have explored generating 3D molecules through generating and connecting fragments first, then aligning atom coordinates with software like RDKit. We refer readers to Du et al. (2022); Zhang et al. (2023b) for a comprehensive review.

While generating 3D molecules in the form of 3D point clouds have been well studied, few studies have tried applying powerful language models to this problem. In this work, different from mainstream methods, we convert 3D point clouds to $SE(3)$-invariant 1D discrete sequences, and show that generating sequences by LMs achieves promising performance in the 3D molecule generation task.

## 2.2 CHEMICAL LANGUAGE MODEL

LMs have catalyzed significant advancements across a spectrum of fields. Recently, LLMs have revolutionized the landscape of NLP (Touvron et al., 2023; Achiam et al., 2023; Chowdhery et al., 2023) and beyond, extending to fields such as computer vision, speech and acoustics, scientific discovery, and multi-modalities. Drawing inspiration from NLP methodologies, chemical language models (CLMs) have emerged as a competent way for representing molecules (Bran & Schwaller, 2023; Janakarajan et al., 2023; Bajorath, 2024; Zhang et al., 2024). Due to the superiority LMs show in generation tasks, most CLMs are designed as generative models. Variants of LMs have been adapted for molecular science, producing a variety of works such as MolGPT (Bagal et al., 2021), MolReGPT (Li et al., 2023a), MolT5 (Edwards et al., 2022), MoleculeGPT (Zhang et al., 2023a), InstructMol (Cao et al., 2023), DrugGPT (Li et al., 2023b), and many others.

CLMs learn the chemical vocabulary and syntax used to represent molecules, as well as the conditional probabilities of character occurrence at given positions of sequences depending on preceding characters. This vocabulary covers all characters from the adopted molecule representation. All inputs including chemical structures and properties should be converted into sequence form and tokenized for compatibility with language models. Commonly, SMILES (Weininger, 1988) is used for this sequential representation, although other formats like SELFIES (Krenn et al., 2019), atom type strings, and custom strings with positional or property values are also viable options. To learn representations, CLMs are usually pre-trained on extensive molecular sequences through self-supervised learning. Subsequently, models are fine-tuned on more focused datasets with desired properties, such as activity against a target protein. Generative CLMs generally adopt an autoregressive training approach of next token prediction, *i.e.*, iteratively predicting each subsequent token in a sequence based on the preceding tokens. Traditional autoregressive models use the Transformer architecture with causal self-attention (Brown et al., 2020) due to its superior efficacy, while other sequence models like recurrent neural networks (RNNs) and state space models (SSMs) (Gu et al., 2021; Özçelik et al., 2024; 2023) also show considerable functionality.

Given a dataset of sequences, $\boldsymbol{U} = \{U_1, U_2, \cdots, U_N\}$, where $U_i$ is transformed from the representation, property conditions and/or descriptions of a molecule $G_i$ with $n_i$ nodes, let $U_i = \{u_1, u_2, \cdots, u_{n_i}\}$ and all tokens $u_i$ belong to vocabulary $V$. An autoregressive CLM has parameters $\theta$ encoding a distribution with conditional probabilities of each token given its predecessors, $p(U_i; \theta) = \prod_{j=1}^{n_i} p(u_j | u_0 : u_{j-1}; \theta)$. The optimization process involves maximizing the probabilities of the entire dataset $p(\boldsymbol{U}; \theta) = \prod_{i=1}^{N} p(U_i; \theta)$. Each conditional distribution $p(u_j | u_0 : u_{j-1}; \theta)$ is a categorical distribution over the vocabulary size $|V|$; thus the loss for each term aligns with the standard cross-entropy loss. To generate new sequences, the model samples each token sequentially from these conditional distributions. To introduce randomness and control into generation, the sampling process is typically modulated with Top-K $(k)$ and temperature $(\tau)$ hyperparameters, enabling a balance between adherence and diversity.

Most existing CLM works consider chemical structures as well as other modalities such as natural language captions (Bagal et al., 2021; Li et al., 2023a;b; Edwards et al., 2022; Xie et al., 2023; Chen et al., 2023b; Tysinger et al., 2023; Xu et al., 2023b; Chen et al., 2023a; Pei et al., 2023; Liu et al., 2023b; Wang et al., 2023), while some focus on pure text of chemical literature (Luo et al., 2022a) or molecule strings (Haroon et al., 2023; Mao et al., 2023b; Blanchard et al., 2023; Mazuz et al., 2023; Fang et al., 2023; Kyro et al., 2023; Izdebski et al., 2023; Yoshikai et al., 2023; Wu et al., 2023; Mao et al., 2023a). Notably, all these works solely consider 2D molecules for representation learning and downstream tasks, overlooking 3D geometric structures which is crucial in many molecular predictive and generative tasks. For example, different conformations of the same 2D molecule have different potentials and per-atom forces. In order to use pivotal 3D information, another line of work incorporate geometric models such as GNNs in parallel with the CLM (Xia et al., 2023; Zhang et al., 2023a; Cao et al., 2023; Liang et al., 2023; Liu et al., 2023a; Frey et al., 2023), which requires additional design and training techniques to mitigate alignment issues. Some works extend the architecture of CLM to include 3D-geometric-model-like modules in the attention block (Fuchs et al., 2020; Shi et al., 2022; Liao & Smidt, 2022; Thölke & De Fabritiis, 2021; Luo et al., 2022b; Masters et al., 2022; Ünlü et al., 2023; Zhao et al., 2023a), capturing 3D information as positional encodings with considerable computations and framework design. In contrast, Flam-Shepherd & Aspuru-Guzik (2023) make an initial attempt showing language models trained directly on contents of XYZ format chemical files can generate molecules with three coordinates, implying pure LMs'

Figure 1: Overview of Geo2Seq. We use the canonical labeling order to arrange nodes in a row, fill in the place of each node with vector $[z_i, d_i, \theta_i, \phi_i]$, and concatenate all elements into a sequence. Each node vector contains atom type and spherical coordinates. Notably, the spherical coordinates are $SE(3)$-invariant.

potential to directly explore 3D chemical space. In this work, we propose an invariant 3D molecular sequencing algorithm, Geo2Seq, to empower CLMs with structural completeness and geometric invariance, showing LMs' capabilities of understanding molecules precisely in 3D space. We extend beyond the conventional Transformer architecture of CLMs and additionally employ SSMs as LM backbones. Furthermore, Geo2Seq operates solely on the input data, which allows independence from model architecture and training techniques and provides reuse flexibility.

**Representation techniques.** Our proposed Geo2Seq leverages spherical representation and canonical labeling techniques. Spherical representation has been applied in various molecule-related tasks (Van Kempen et al., 2024), including molecular property prediction (Liu et al., 2022; Gao et al., 2022) and molecule generation (Luo & Ji, 2022b). A crucial step in using spherical representation is defining the coordinate frame, *i.e.*, x, y, and z axes. One straightforward approach is to directly use the frame as for the input coordinates, *i.e.*, (1, 0, 0) as x-axis, but this case fails to ensure $SE(3)$-invariance. Instead, SphereNet (Liu et al., 2022) defines local frames based on a central edge and one reference node to ensure invariance (Liu et al., 2022; Gao et al., 2022). Similarly, G-SphereNet defines local frames based on focal atoms to compute distance and angle for model generation (Luo & Ji, 2022b). These approaches demonstrate the importance of spherical representations in molecule-related tasks. Canonical labeling (CL) has been adopted from the graph theory and used in molecular representation, enabling the conversion of molecules into 1D sequences. This allows for efficient processing and analysis of chemical structures. One of the most popular canonical sequence is canonical SMILES (Weininger et al., 1989), which represents molecules as a string of characters based on the Morgan Algorithm (Morgan, 1965) and additional defined rules. SELFIES provides a more robust sting representation to overcome the limitation that some strings do not correspond to valid molecules. However, the application of CL to 3D molecules has yet been studied.

## 3 TOKENIZATION OF 3D MOLECULES

A fundamental difference between LMs and other models is that LMs use discrete inputs, *i.e.*, tokens. In this section, we introduce our tokenization method to map input 3D molecules with atomic coordinates to discrete token sequences appropriate for LM learning.

A main challenge in tokenization design is to develop bijective mappings between 3D molecules and token sequences, *i.e.*, obtaining the same token sequence for the same input 3D molecule, while obtaining different sequences for different inputs. In this section, we present our solutions to tackle this challenge. We first reorder the atoms in the input molecule to a canonical order (Section 3.1), such that any two isomorphic graphs result in the same canonical form, and any non-isomorphic graphs yield different canonical forms. We then convert 3D Cartesian coordinates to $SE(3)$-invariant spherical representations, including distances and angles (Section 3.2). Combining them together, we obtain our geometry informed tokenization method Geo2Seq (Section 3.3). We provide rigorous proof of all theorems supporting the bijective mapping relation in Appendix B.

### 3.1 SERIALIZATION VIA CANONICAL ORDERING

As the first step in 3D molecule tokenization, we need to transform a graph to a 1D sequential representation. We resort to canonical labeling as a solution for dimension reduction without information loss.

Canonical labeling (CL), in the context of graph theory, is a process to assign a unique form to each graph in a way that two graphs receive the same canonical form only if they are isomorphic (McKay

et al., 1981). The canonical form is a re-indexed version of a graph, which is unique for the whole isomorphism class of a graph. The new indexes naturally establish the order of nodes in the graph. The order, which we refer to as canonical labels, is not necessarily unique if the graph has symmetries and thus has an automorphism group larger than 1. However, all canonical labels are strictly equivalent when used for serialization. The canonical label essentially re-assigns an index $\ell_i$ to each node originally indexed with $i$ in graph $G$. Since canonical labeling can precisely distinguish non-isomorphic graphs, it fully contains the structure information of a graph $G$. Thus, by arranging nodes with attributes in the labeling order $\ell_1, \ell_2, \cdots$, we obtain a sequential representation of attributed graphs with all structural information preserved.

The Nauty algorithm (McKay & Piperno, 2014), tailored for CL and computing graph automorphism groups, presents a rigorous implementation of CL. In this paper, we adopt the Nauty algorithm for CL calculation, while all analyses and derivations apply to other rigorous algorithms. The bijective mapping relation between CL-obtained sequential representation and graphs can be be proved based on graph isomorphism. We first formally define graph isomorphism as Def. B.1. Due to the geometric needs in our case, we move a step forward and extend the isomorphism problem to node/edge-attributed graphs. This leads us to the guarantee below.

**Lemma 3.1.** *[Canonical Labeling for Colored Graph Isomorphism] Let $G_1 = (V_1, E_1, A_1)$ and $G_2 = (V_2, E_2, A_2)$ be two finite, undirected graphs where $V_i$ denotes the set of vertices, $E_i$ denotes the set of edges, and $A_i$ denotes the node attributes of the graph $G_i$ for $i = 1, 2$. Let $\boldsymbol{L} : \mathcal{G} \to \mathcal{L}$ be a function that maps a graph $G \in \mathcal{G}$, the set of all finite, undirected graphs, to its canonical label $\boldsymbol{L}(G) \in \mathcal{L}$, the set of all possible canonical labels, as produced by the Nauty algorithm. Then the following equivalence holds:*

$$\boldsymbol{L}(G_1) = \boldsymbol{L}(G_2) \Leftrightarrow G_1 \cong G_2$$

*where $G_1 \cong G_2$ denotes that $G_1$ and $G_2$ are isomorphic.*

Lemma 3.1 indicates that the CL process is both complete (sufficient to distinguish non-isomorphic graphs) and sound (not distinguishing actually isomorphic graphs). Note that if $\boldsymbol{L}(G)$ corresponds to multiple automorphic labels, we can randomly select one since they are all equivalent and produce the same sequence later through Geo2Seq, as detailed in Appendix B. However, this is a very uncommon case for real-world 3D attributed graphs like molecules.

## 3.2 INVARIANT SPHERICAL REPRESENTATIONS

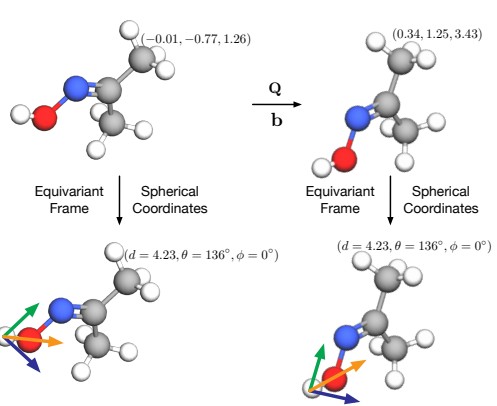

Figure 2: Illustrations of the equivariant frame and invariant spherical representations. If the molecule is rotated and translated by a rotation matrix $\boldsymbol{Q}$ and a translation vector $\boldsymbol{b}$, the atom coordinates change accordingly. But our spherical representations remain invariant since the frame is equivariant to the $SE(3)$-transformation.

In this section, we describe how to incorporate 3D structure information into our sequences. One main challenge here is to ensure the $SE(3)$-invariance property described in Section 2.1. Specifically, given a 3D molecule, if it is rotated or translated in the 3D space, its 3D representation should be unchanged. Another challenge is to ensure no information loss (Liu et al., 2022; Wang et al., 2022). Specifically, given the 3D representation, we can recover the given 3D structure. If two 3D structures cannot be matched via a $SE(3)$ transformation, the representations should be different. This property is important to the discriminative ability of models.

We address these challenges by **spherical representations**, *i.e.*, using spherical coordinates to represent 3D structures. Compared to Cartesian coordinates, spherical coordinate values are bounded in a smaller region, namely, a range of $[0, \pi]$ or $[0, 2\pi]$. This makes spherical coordinates advantageous in discretized representations and thus easier to be modeled by LMs. Given the same decimal place constraints, spherical coordinates require a smaller vocabulary size, and given the same vocabulary size, spherical coordinates present less information loss. This is also supported by empirical results and analysis when using different methods to represent 3D molecular structures, as detailed in Appendix C.

We propose to maintain $SE(3)$-invariance while ensuring no information loss. Given a 3D molecule $G$ with atom types $z$ and atom coordinates $R$, we first build a **global coordinate frame** $F = (x, y, z)$ based on the input. Specifically, as shown in Figure 1, the frame is built based on the first three non-collinear atoms in the canonical ordering $L(G)$. Let $\ell_1, \ell_2$, and $\ell_F$ be the indices of these three atoms. Then the global frame $F = (x, y, z)$ is calculated as

$$
\begin{aligned}
x &= \text{normalize}(r_{\ell_2} - r_{\ell_1}), \\
y &= \text{normalize}\left((r_{\ell_F} - r_{\ell_1}) \times x\right), \\
z &= x \times y.
\end{aligned}
\tag{1}
$$

Here $\text{normalize}(\cdot)$ is the function to normalize a vector to unit length. Note that the global frame is equivariant to the rotation and translation of the input molecule, as shown in Figure 2 and Appendix B.2. After obtaining the global frame, we use a function $f(\cdot)$ to convert the coordinates of each atom to **spherical coordinates** $d, \theta, \phi$ under this frame. Specifically, for each node $\ell_i$ with coordinate $r_{\ell_i}$, the corresponding spherical coordinate is

$$
\begin{aligned}
d_{\ell_i} &= \|r_{\ell_i} - r_{\ell_1}\|_2, \\
\theta_{\ell_i} &= \arccos\left((r_{\ell_i} - r_{\ell_1}) \cdot z / d_{\ell_i}\right), \\
\phi_{\ell_i} &= \text{atan2}\left((r_{\ell_i} - r_{\ell_1}) \cdot y, (r_{\ell_i} - r_{\ell_1}) \cdot x\right).
\end{aligned}
\tag{2}
$$

The spherical coordinates show the relative position of each atom in the global frame $F$. As shown in Figure 2, if the input coordinates are rotated by a matrix $Q$ and translated by a vector $b$, the transformed spherical coordinates remain the same, so the spherical coordinates are $SE(3)$-invariant.

Next, we demonstrate that there is no information loss in our method. We show that given our $SE(3)$-invariant spherical representations, we can recover the given 3D structures. For each node $\ell_i$, we convert the spherical coordinate $[d_{\ell_i}, \theta_{\ell_i}, \phi_{\ell_i}]$ to coordinate $r'_{\ell_i}$ in 3D space as

$$
[d_{\ell_i} \sin(\theta_{\ell_i}) \cos(\phi_{\ell_i}), d_{\ell_i} \sin(\theta_{\ell_i}) \sin(\phi_{\ell_i}), d_{\ell_i} \cos\theta_{\ell_i}].
\tag{3}
$$

Note that our reconstructed coordinate $r'_{\ell_i}$ may not be exactly the same as the original coordinate $r_{\ell_i}$. However, there exists a $SE(3)$-transformation $g$, such that $g(r'_{\ell_i}) = r_{\ell_i}$ for all $i$. Note that the same transformation $g$ is applied to all nodes. Formally, by applying the function $f(\cdot)$ to the 3D coordinate matrix $R$, we can demonstrate the following properties of spherical representations.

**Lemma 3.2.** *Let $G = (z, R)$ be a 3D graph with node type vector $z$ and node coordinate matrix $R$. Let $F$ be the equivariant global frame of graph $G$ built based on the first three non-collinear nodes in $L(G)$. $f(\cdot)$ is our function that maps 3D coordinate matrix $R$ of $G$ to spherical representations $S$ under the equivariant global frame $F$. Then for any 3D transformation $g \in SE(3)$, we have $f(R) = f(g(R))$. Given spherical representations $S = f(R)$, there exist a transformation $g \in SE(3)$, such that $f^{-1}(S) = g(R)$.*

Lemma 3.2 indicates that our spherical representation is $SE(3)$-invariant, and we can reconstruct (a transformation of) the original coordinates. Therefore, our method can convert 3D structures into $SE(3)$-invariant representations with no information loss. Detailed proofs are provided in Appendix B.

## 3.3 GEO2SEQ: GEOMETRY INFORMED TOKENIZATION

In this section, we describe the process and properties of our 3D tokenization method, Geo2Seq. Equipped with canonical labeling that reduces graph structures to 1D sequences with no information loss regarding graph isomorphism, and $SE(3)$-invariant spherical representations that ensure no 3D information loss, we develop Geo2Seq, a reversible transformation from 3D molecules to 1D sequences. Figure 1 shows an overview of Geo2Seq. Specifically, given a graph $G$ with $n$ nodes, Geo2Seq concatenates the node vector $[z_i, d_i, \theta_i, \phi_i]$ of every node in $G$ to a 1D sequence by its canonical order, $\ell_1, \cdots, \ell_n$. To formulate the properties of Geo2Seq, we extend the concept of graph isomorphism in Definition B.1 to 3D graphs.

**Definition 3.3.** [3D Graph Isomorphism] Let $G_1 = (z_1, R_1)$ and $G_2 = (z_2, R_2)$ be two 3D graphs, where $z_i$ is the node type vector and $R_i$ is the node coordinate matrix of the molecule $G_i$. Let $V_i$ denote the set of vertices, $A_i$ denote node attributes, and no edge exists. Two 3D graphs $G_1$ and $G_2$ are **3D isomorphic**, denoted as $G_1 \cong_{3D} G_2$, if there exists a bijection $b : V_1 \to V_2$ such

that $G_1 \cong G_2$ given $A_i = [\boldsymbol{z}_i, \boldsymbol{R}_i]$, and there exists a 3D transformation $g \in SE(3)$ such that $\boldsymbol{r}_i^{G_1} = g(\boldsymbol{r}_{b(i)}^{G_2})$. If a small error $\boldsymbol{\epsilon}$ is allowed such that $|\boldsymbol{r}_i^{G_1} - g(\boldsymbol{r}_{b(i)}^{G_2})| \leq \boldsymbol{\epsilon}$, we call the two 3D graphs $\boldsymbol{\epsilon}$-**constrained 3D isomorphic**.

Considering Lemma 3.1, we specify $G = (V, E, A)$ with $A = [\boldsymbol{z}, \boldsymbol{R}]$ and define the CL function for 3D molecules as $\boldsymbol{L}_m$, which extends the equivalence of Lemma 3.1 to $\boldsymbol{L}_m$ with 3D isomorphism. We formulate Geo2Seq and our major theoretical derivations below.

**Theorem 3.4.** *[Bijective Mapping between 3D Graph and Sequence] Following Definition 3.3, let $G_1 = (\boldsymbol{z}_1, \boldsymbol{R}_1)$ and $G_2 = (\boldsymbol{z}_2, \boldsymbol{R}_2)$ be two 3D graphs. Let $\boldsymbol{L}_m(G)$ be the canonical label for 3D graph $G$ and $f : \mathcal{R} \to \mathcal{S}$ be the function that maps 3D coordinates to its spherical representations. Given a graph $G$ with $n$ nodes and $\boldsymbol{X} = [\boldsymbol{x}_1, \cdots, \boldsymbol{x}_n]^T \in \mathbb{R}^{n \times m}$, where $m \in \mathbb{Z}$, we define $\boldsymbol{L}_m(G) \otimes \boldsymbol{X} = concat(\boldsymbol{x}_{\ell_1}, ..., \boldsymbol{x}_{\ell_n})$, where $\ell_i$ is the index of the node labeled $i$ by $\boldsymbol{L}_m(G)$, and $concat(\cdot)$ concatenates elements as a sequence. We define*

$$Geo2Seq(G) = \boldsymbol{L}_m(G) \otimes (\boldsymbol{z}, f(\boldsymbol{R})) = \boldsymbol{L}_m(G) \otimes \boldsymbol{X},$$

*where $\boldsymbol{x}_i = [z_i, d_i, \theta_i, \phi_i]$. Then $Geo2Seq : \mathcal{G} \to \mathcal{U}$ is a surjective function, and the following equivalence holds:*

$$Geo2Seq(G_1) = Geo2Seq(G_2) \Leftrightarrow G_1 \cong_{3D} G_2,$$

*where $G_1 \cong_{3D} G_2$ denotes $G_1$ and $G_2$ are 3D isomorphic.*

Theorem 3.4 establishes the following guarantees for Geo2Seq: (1) Given a 3D molecule, we can uniquely construct a 1D sequence using Geo2Seq. (2) If two molecules are 3D isomorphic, their sequence outputs from Geo2Seq are identical. (3) Given a sequence output of Geo2Seq, we can uniquely reconstruct a 3D molecule. (4) If two constructed sequences from Geo2Seq are identical, their corresponding molecules must be 3D isomorphic. This enable sequential tokenization of 3D molecules, preserving structural completeness and geometric invariance.

Due to the necessity of discreteness in serialization and tokenization for LMs, in reality, numerical values need to be discretized before concatenation. In practice, we round up numerical values to certain decimal places. Thus Theorem 3.4 can be extended with constraints, as below.

**Corollary 3.5.** *[Constrained Bijective Mapping between 3D Graph and Sequence] Following the notations and definitions of Theorem 3.4, let spherical coordinate values be rounded up to $b$ decimal places. Then $Geo2Seq : \mathcal{G} \to \mathcal{U}$ is a surjective function, and the following equivalence holds:*

$$Geo2Seq(G_1) = Geo2Seq(G_2) \Leftrightarrow G_1 \cong_{3D-|10^{-b}|/2} G_2,$$

*where $G_1 \cong_{3D-|10^{-b}|/2} G_2$ denotes graphs $G_1$ and $G_2$ are $(|10^{-b}|/2)$-constrained 3D isomorphic.*

Corollary 3.5 extends Theorem 3.4's guarantees for the practical use of Geo2Seq. If we allow a round-up error below $|10^{-b}|/2$ for coordinates when distinguishing 3D isomorphism, all properties still hold. This implies that the practical Geo2Seq implementation retains near-complete geometric information and invariance, with numerical precision of $\boldsymbol{\epsilon} \leq |10^{-b}|/2$.

With discreteness incorporated, we can collect a finite vocabulary covering all accessible molecule samples to enable tokenization for LMs. Specifically, we use vocabularies of approximately 1K-16K tokens consisting of atom type tokens '$C, N, O \cdots$', and spherical coordinate tokens such as '$-1.98$', '$1.57°$' or '$-0.032°$'. Specifically, the vocabulary size is approximately 1.8K for the QM9 dataset, and 16K for the Geom-Drug dataset. Note that we consider chirality for atoms and use the special token suffixes '@' and '@@' to distinguish clockwise and counterclockwise chiral centers, for example, '$C@$' and '$C@@$'. The numerical tokens range from the smallest to the largest distance and angle values with restricted precision of 2 or 3 decimal places. Experimental results show the benefits in using this level of tokenization, as detailed in Appendix C.

## 4  3D MOLECULE GENERATION

**Training and Sampling**. Now that we have defined a canonical and robust sequence representation for 3D molecules, we turn to the method of modeling such sequences, $U$. Here, we attempt to train a model $M$ with parameters $\theta$ to capture the distribution of such sequences, $p_\theta(U)$, in our dataset. As this is a well-studied problem within language modeling, we opt to use two language models, GPT

Table 1: Random generation performance on QM9 and GEOM-DRUGS datasets. Here, larger numbers indicate better performance. **bold** and underline highlight the best and second best performance, respectively. Note that for GEOM-DRUGS dataset, molecule stability and unique percentage are close to 0% and 100% for all methods so they are not presented. Following Hoogeboom et al. (2022) and Xu et al. (2023a), we report the mean and standard deviation over three runs on QM9 dataset.

| Method | QM9 | | | | GEOM-DRUGS | |
|---|---|---|---|---|---|---|
| | Atom Sta (%) | Mol Sta (%) | Valid (%) | Valid & Unique (%) | Atom Sta (%) | Valid (%) |
| Data | 99.0 | 95.2 | 97.7 | 97.7 | 86.5 | 99.9 |
| E-NFs | 85.0 | 4.9 | 40.2 | 39.4 | - | - |
| G-SchNet | 95.7 | 68.1 | 85.5 | 80.3 | - | - |
| GDM | 97.0 | 63.2 | - | - | 75.0 | 90.8 |
| GDM-AUG | 97.6 | 71.6 | 90.4 | 89.5 | 77.7 | 91.8 |
| EDM | 98.7±0.1 | 82.0±0.4 | 91.9±0.5 | 90.7±0.6 | 81.3 | 92.6 |
| EDM-Bridge | 98.8±0.1 | 84.6±0.3 | 92.0±0.1 | 90.7±0.1 | 82.4 | 92.8 |
| GEOLDM | **98.9**±0.1 | 89.4±0.5 | 93.8±0.4 | **92.7**±0.5 | **84.4** | **99.3** |
| Geo2Seq with GPT | 98.3±0.1 | 90.3±0.1 | 94.8±0.2 | 80.6±0.4 | 82.6 | 87.4 |
| Geo2Seq with Mamba | **98.9**±0.2 | **93.2**±0.2 | **97.1**±0.2 | 81.7±0.4 | 82.5 | 96.1 |

(Radford et al., 2018) and Mamba (Gu & Dao, 2023), which have shown effective sequence modeling capabilities on a range of tasks. Both models are trained using a standard next-token prediction cross-entropy loss $\ell$ for all elements in the sequence:

$$\min_{\theta} \; \mathbb{E}_{u \in U} \left[ \sum_{i=1}^{|u|-1} \ell\left(M_{\theta}(u_1, \cdots, u_i), u_{i+1}\right) \right].$$

To sample from a trained model, we first select an initial atom token by sampling from the multinomial distribution of first-tokens in the training data (we note that in almost all cases this is 'H'). We then perform a standard autoregressive sampling procedure by iteratively sampling from the conditional distribution $p_{\theta}(u_{i+1}|u_1, \cdots, u_i)$ until the stop token or max length is reached. We sample from this distribution using top-$k$ sampling (Fan et al., 2018) and a softmax temperature $\tau$ (Ackley et al., 1985; Ficler & Goldberg, 2017). Unless otherwise noted, $\tau = 0.7$ and $k = 80$.

**Controllable Generation**. For controllable generation, we follow Bagal et al. (2021) and use a conditioning token for the desired property. This token is created by projecting the desired properties through a trainable linear layer to create a vector with the model's initial token embedding space. This property token is then used as the initial element in the molecular sequence. Training and sampling are performed as before with this new sequence formulation. Sampling begins with the desired property's token as input.

## 5 Experimental Studies

In this section, we evaluate the method of generating 3D molecules in the form of our proposed Geo2Seq representations by LLMs. We show that in the random generation task (see Section 2.1), the performance of Geo2Seq with GPT (Radford et al., 2018) or Mamba (Gu & Dao, 2023) models is better than or comparable with state-of-the-art 3D point cloud based methods, including EDM (Hoogeboom et al., 2022) and GEOLDM (Xu et al., 2023a). In addition, in the controllable generation task (see Section 2.1), we show that Geo2Seq with Mamba models outperform previous 3D point cloud based methods by a large margin.

### 5.1 Random Generation

**Data.** We adopt two datasets, QM9 (Ramakrishnan et al., 2014) and GEOM-DRUGS (Axelrod & Gomez-Bombarelli, 2022), to evaluate performances in the random generation task. The QM9 dataset collects over 130k 3D molecules with 3D structures calculated by density functional theory (DFT). Each molecule in QM9 has less than 9 heavy atoms and its chemical elements all belong to H, C, N, O, F. Following Anderson et al. (2019), we split the dataset into train, validation and test sets with 100k, 18k and 12k samples, separately. The GEOM-DRUGS dataset consists of over 450k large molecules with 37 million DFT-calculated 3D structures. Molecules in GEOM-DRUGS has up to 181 atoms and 44.2 atoms on average. We follow Hoogeboom et al. (2022) to select 30 3D structures with the lowest energies per molecule for model training.

**Setup.** On the QM9 dataset, we set the training batch size to 32, base learning rate to 0.0004, and train a 12-layer GPT model and a 26-layer Mamba model by AdamW (Loshchilov & Hutter, 2019)

Table 2: Controllable generation performance of property MAE on QM9 datasets. Smaller numbers indicate better performance, and **bold** and underline highlight the best and second best performances.

| Property (Units) | $\alpha$ (Bohr$^3$) | $\Delta\epsilon$ (meV) | $\epsilon_{\text{HOMO}}$ (meV) | $\epsilon_{\text{LUMO}}$ (meV) | $\mu$ (D) | $C_v$ ($\frac{\text{cal}}{\text{mol}}$K) |
|---|---|---|---|---|---|---|
| Data | 0.10 | 64 | 39 | 36 | 0.043 | 0.040 |
| Random | 9.01 | 1470 | 645 | 1457 | 1.616 | 6.857 |
| $N_{\text{atoms}}$ | 3.86 | 866 | 426 | 813 | 1.053 | 1.971 |
| EDM | 2.76 | 655 | 356 | 584 | 1.111 | 1.101 |
| GEOLDM | 2.37 | 587 | 340 | 522 | 1.108 | 1.025 |
| Geo2Seq with Mamba | **0.46** | **98** | 57 | 71 | 0.164 | **0.275** |
| Geo2Seq with GPT | 0.53 | 102 | **48** | **53** | **0.097** | 0.325 |

optimizers. On the GEOM-DRUGS dataset, we set the training batch size to 32, base learning rate to 0.0004, and train a 14-layer GPT model and a 28-layer Mamba model by AdamW optimizers. See Appendix D for more information about hyperparameters and other settings. When model training is completed, we randomly generate 10,000 molecules, and evaluate the performance on these molecules. Specifically, we first transform 3D molecular structures to 2D molecular graphs using the bond inference implementation of EDM. Then, we evaluate the performance by **atom stability**, which is the percentage of atoms with correct bond valencies, and **molecule stability**, which is the percentage of molecules whose all atoms have correct bond valencies. We also report the percentage of **valid** molecules that can be successfully converted to SMILES strings by RDKit, and the percentage of **valid and unique** molecules that can be converted to unique SMILES strings.

**Baselines.** We compare GPT and Mamba models with several strong baseline methods. Specifically, we compare with an autoregressive generation method G-SchNet (Gebauer et al., 2019) and an equivariant flow model based method E-NFs (Satorras et al., 2021a). We also compare with some recently proposed diffusion based methods, including EDM (Hoogeboom et al., 2022), GDM (the non-equivariant variant of EDM) and GDM-AUG (GDM trained with random rotation as data augmentation). Besides, we compare with EDM-Bridge (Wu et al., 2022) and GEOLDM (Xu et al., 2023a), which are two latest 3D molecule generation methods improving EDM by SDE based diffusion models and latent diffusion models, respectively. To ensure that the comparison is fair, our methods and baseline methods use the same data split and evaluation metrics.

**Results.** We present the random generation results of different methods on QM9 and GEOM-DRUGS datasets in Table 1. Note that for GEOM-DRUGS dataset, all methods achieve nearly 0% molecule stability percentage and 100% uniqueness percentage. Thus, following previous studies, these two metrics are omitted. According to the results in Table 1, on QM9 dataset, generating 3D molecules in Geo2Seq representations with either GPT or Mamba models achieve better performance than all 3D point cloud based baseline methods in molecule stability and valid percentage, and achieves atom stability percentages close to the upper bound (99%). This demonstrates that our method can model 3D molecular structure distribution and capture the underlying chemical rules more accurately. It is worth noticing that our method does not achieve very high uniqueness percentage, showing that it is not easy for our method to generate a large number of diverse molecules. We believe this is due to that the conversion from real numbers to discrete tokens limits the search space of 3D molecular structures, especially on a small dataset like QM9, while it is easier to generate more diverse molecules for 3D point cloud based methods as they directly generate real numbers. This is reflected by the fact that our method achieves nearly 100% uniqueness percentage on the large GEOM-DRUGS dataset. On GEOM-DRUGS dataset, both GPT and Mamba models achieve reasonably high atom stability and valid percentage. The performance of our method is comparable with strong diffusion based baseline methods, showing that LLMs have the potential to model very complicated drug molecular structures well. We will explore further improvements on GEOM-DRUGS with larger LLMs in the future.

See Appendix D.3 for additional experiments on more baselines (Huang et al., 2023; Vignac et al., 2023), and metrics including percentage of novel/complete molecules. See Appendix C for ablation studies about atom order, 3D representation and tokenization, Appendix D for generation complexity analysis and results with pretraining, and Appendix F for token embedding and molecule visualization.

## 5.2 CONTROLLABLE GENERATION

**Data.** In the controllable generation task, we train our models on molecules and their property labels in the QM9 (Ramakrishnan et al., 2014) dataset. Specifically, we try taking a certain quantum property value as the conditional input to LLMs, and train LLMs to generate molecules with the conditioned

quantum property values. Following Hoogeboom et al. (2022), we split the training dataset of QM9 to two subsets where each subset has 50k samples, and train our conditional generation models and an EGNN (Satorras et al., 2021b) based quantum property prediction models on these two subsets, respectively. We conduct the controllable generation experiments on six quantum properties from QM9, including (1) polarizability ($\alpha$), tendency of a molecule to acquire an electric dipole moment when subjected to anexternal electric field, (2) HOMO energy ($\epsilon_{\text{HOMO}}$), highest occupied molecular orbital energy, (3) LUMO energy ($\epsilon_{\text{LUMO}}$), lowest unoccupied molecular orbital energy, (4) HOMO-LUMO gap ($\Delta\epsilon$), energy difference between HOMO and LUMO, (5) dipole moment ($\mu$) and (6) heat capacity at 298.15K ($C_v$). All properties are dependent on the 3D molecular conformation. For example, the dipole moment vector quantity depends on the orientation of the 3D conformer, and heat capacity is related to the vibration of molecule in 3D space.

**Setup.** For the controllable generation experiment, we train 16-layer Mamba (Gu & Dao, 2023) models with the same hyperparameters as the random generation experiments in Section 5.1. To evaluate the performance, we sample 10000 quantum property values, generate molecules conditioned on these property values by trained models, and compute the mean absolute difference (**MAE**) between the given property values and the property values of the generated molecules. Note that we use the trained EGNN based property prediction models to calculate the property values of the generated molecules.

**Baselines.** We compare our models with two equivariant diffusion models, EDM (Hoogeboom et al., 2022) and GEOLDM (Xu et al., 2023a). In addition, we use several baselines that are based on dataset molecules. One baseline (Data) is directly taking the molecules from the QM9 dataset and use their property values as conditions. The MAE metric simply reflects the prediction error of the trained property prediction model, which can be considered as a lower bound. The second baseline (Random) is taking the molecules from the dataset but uses the randomly shuffled property values as conditions, and its MAE can be considered as an upper bound. The third baseline ($N_{\text{atoms}}$) uses the molecules from the dataset but uses property values predicted from the number of atoms as conditions. Achieving better performance than this baseline shows that models can use conditional information beyond the number of atoms.

**Results.** Controllable generation results of different methods are summarized in Table 2. As shown in the table, among all six properties, our method outperforms the strong diffusion based baseline methods EDM and GEOLDM by a large margin. Our method moves a significant step in pushing the performance of controllable generation task towards the lower bound, *i.e.*, Data baseline. As we use the same training set as EDM and GEOLDM to train the conditional generation model, the good performance of our method shows that LLMs have more powerful capacity in incorporating conditional information into the 3D molecular structure generation process. We believe that the powerful long-context correlation capturing structures from LLMs, *e.g.*, attention mechanism, play significant roles in achieving the good control of 3D molecule generation by the conditioned property values. The huge success of LLMs in controllable molecule generation will motivate broader applications of LLMs in goal-directed or constrained drug design. In addition, our method has the potential to generate new molecules with desired properties such as smaller HOMO-LUMO gaps, thereby accelerating the discovery of new materials. See Appendix F for visualization of molecules generated from given polarizability values.

## 6 CONCLUSION AND DISCUSSION

Geo2Seq showcases the potential of pure LMs in revolutionizing molecular design and drug discovery when geometric information is properly transformed. Traditional diffusion-based models fall short in terms of efficiency, scalability, and the ability to learn from extensive databases or transfer knowledge across different tasks. In contrast, LMs exhibit inherent advantages in these areas. We envision the development of efficient, large-scale models trained on vast chemical databases that can function across multiple datasets and molecular tasks. By introducing LMs into the 3D molecule generation field, we unlock substantial potential for broad scientific impact. The framework has certain limitations, particularly in the generalization abilities across the continuous domain of real numbers. Due to the discrete nature of vocabularies, LMs rely on large pre-training corpus, fine-grained tokenization or emergent abilities for better generalization, as a trade-off to high precision and versatility. Future works points towards directions such as advanced tokenization techniques and more tasks. Despite these challenges, our work represents a significant step forward in this new field.

REPRODUCIBILITY STATEMENT

To ensure the reproducibility of this work, for our theoretical results, all assumptions and proofs are included in Appendix B. For the experiments, we provide full details including all the training setup, architecture, and hyper-parameter searching spaces in Appendix D.1. Licenses are provided in Appendix D.2. The finalized code will be released upon acceptance.

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

## A  BROADER IMPACTS AND LIMITATIONS

Our work demonstrates the significant potential of pure language models (LMs) in revolutionizing molecular design and drug discovery by effectively transforming geometric information. The challenge of molecule design is particularly daunting when scientific experiments are cost-prohibitive or impractical. In many real-world scenarios, data collection is confined to specific chemical domains, yet the ability to generate molecules for broader tasks where experimental validation is difficult remains crucial. Traditional diffusion-based models fall short in terms of efficiency, scalability, and the ability to learn from extensive databases or transfer knowledge across different tasks. In contrast, LMs exhibit inherent advantages in these areas. We envision the development of efficient, large-scale models trained on vast chemical databases that can function across multiple datasets and molecular tasks. By introducing LMs into the 3D molecule generation field, we unlock substantial potential for broad scientific impact.

Our research adheres strictly to ethical guidelines, with no involvement of human subjects or potential privacy and fairness issues. This work aims to advance the field of Machine Learning and AI for drug discovery, with no immediate societal consequences requiring specific attention. We foresee no potential for malicious or unintended usage beyond known chemical applications. However, we recognize that all technological advancements carry inherent risks, and we advocate for ongoing evaluation of the broader implications of our methodology in various contexts.

We admit certain limitations, including that rounding up numerical values to certain decimal places bring information loss and discretized numbers impair generalization abilities across the continuous domain of real numbers. However, this is a trade-off betweeen advantages brought by our model-agnostic framework. Due to the discrete nature of vocabularies, LMs depend on extensive pre-training corpora, fine-grained tokenization, or emergent abilities for better generalization, balancing high precision and versatility. Geo2Seq operates solely on the input data, which allows independence from model architecture and training techniques and provides reuse flexibility. This also means that we can effortlessly apply Geo2Seq on the latest generative language models, making seamless use of their capabilities. Future work points towards expanding on conditional tasks and exploring advanced tokenization techniques to enhance the model's performance and applicability.

## B  PROOFS

### B.1  PROOF OF LEMMA 3.1

First, we define the isomorphism problem for attributed graphs as follows.

**Definition B.1.** [Graph Isomorphism] Let $G_1 = (V_1, E_1, A_1)$ and $G_2 = (V_2, E_2, A_2)$ be two graphs, where $V_i$ denotes the set of vertices, $E_i$ denotes the set of edges, and $A_i$ denotes the node attributes of $G_i$ for $i = 1, 2$. Let $attr(v)$ denote the node attributes of vertex $v$. The graphs $G_1$ and $G_2$ are said to be isomorphic, denoted as $G_1 \cong G_2$, if there exists a bijection $b : V_1 \to V_2$ such that for every vertex $v \in V_1$, $attr(v) \in A_1 = attr(b(v)) \in A_2$, and for every pair of vertices $u, v \in V_1$,
$$(u, v) \in E_1 \Leftrightarrow (b(u), b(v)) \in E_2.$$

Next we prove Lemma 3.1.

**Lemma** (Colored Canonical Labeling for Graph Isomorphism). *Let $G_1 = (V_1, E_1, A_1)$ and $G_2 = (V_2, E_2, A_2)$ be two finite, undirected graphs where $V_i$ denotes the set of vertices, $E_i$ denotes the set of edges, and $A_i$ denotes the node attributes of the graph $G_i$ for $i = 1, 2$. Let $\boldsymbol{L} : \mathcal{G} \to \mathcal{L}$ be a function that maps a graph $G \in \mathcal{G}$, the set of all finite, undirected graphs, to its canonical labeling $\boldsymbol{L}(G) \in \mathcal{L}$, the set of all possible canonical labelings, as produced by the Nauty algorithm. Then the following equivalence holds:*
$$\boldsymbol{L}(G_1) = \boldsymbol{L}(G_2) \iff G_1 \cong G_2$$
*where $G_1 \cong G_2$ denotes that the graphs $G_1$ and $G_2$ are isomorphic.*

The Nauty algorithm, tailored for CL and computing graph automorphism groups, presents rigorous mathematical underpinnings to guarantee the CL properties. Here we leave out the proof of Nauty algorithm's rigor for canonical labeling, which is detailed in the work of McKay & Piperno (2014). The key is the refinement process ensuring that the partitioning of the graph's vertices is done in such a way that any two isomorphic graphs will end with the same partition structure.

## B.2   PROOF OF LEMMA 3.2

**Lemma.** *Let $G = (z, R)$ be a 3D graph with node type vector $z$ and node coordinate matrix $R$. Let $F$ be the equivariant global frame of graph $G$ built based on the first three non-collinear nodes in $L(G)$. $f(\cdot)$ is our function that maps 3D coordinate matrix $R$ to spherical representations $S$ under the equivariant global frame $F$. Then for any $SE(3)$ transformation $g$, we have $f(R) = f(g(R))$. Given spherical representations $S = f(R)$, there exist a $SE(3)$ transformation $g$, such that $f^{-1}(S) = g(R)$.*

*Proof.* Let $\ell_1, \ell_2$, and $\ell_F$ be the indices of the first three non-collinear atoms in $G$. Then the global frame $F = (x, y, z)$ is

$$x = \text{normalize}(r_{\ell_2} - r_{\ell_1})$$
$$y = \text{normalize}\left((r_{\ell_F} - r_{\ell_1}) \times x_1\right)$$
$$z = x \times y$$

For a $SE(3)$ transformation $g$, let $R' = g(R) = QR + b$. Then the global frame $F' = (x', y', z')$ is

$$x' = \text{normalize}(r_{\ell_2} - r_{\ell_1}) = \text{normalize}(g(r_{\ell_2}) - g(r_{\ell_1})) = Qx$$
$$y' = \text{normalize}\left((r_{\ell_F} - r_{\ell_1}) \times x_2\right) = \text{normalize}\left((g(r_{\ell_F}) - g(r_{\ell_1})) \times x_2\right) = Qy$$
$$z' = x' \times y' = (Qx) \times (Qy) = Qz$$

Thus $F' = QF$. Here normalize($\cdot$) is the function to normalize a vector to the corresponding unit vector. Then $\forall i$, the spherical representations $f(R)_{\ell_i}$ is

$$d_{\ell_i} = \|r_{\ell_i} - r_{\ell_1}\|_2$$
$$\theta_{\ell_i} = \arccos\left((r_{\ell_i} - r_{\ell_1}) \cdot z / d_{\ell_i}\right)$$
$$\phi_{\ell_i} = \text{atan2}\left((r_{\ell_i} - r_{\ell_1}) \cdot y, (r_{\ell_i} - r_{\ell_1}) \cdot x\right)$$

Similarly, the spherical representations $f(R')_{\ell_i}$ is

$$d'_{\ell_i} = \|r'_{\ell_i} - r'_{\ell_1}\|_2 = \|g(r_{\ell_i}) - g(r_{\ell_1})\|_2 = d_{\ell_i}$$
$$\theta'_{\ell_i} = \arccos\left((r'_{\ell_i} - r'_{\ell_1}) \cdot z' / d'_{\ell_i}\right) = \arccos\left((g(r_{\ell_i}) - g(r_{\ell_1})) \cdot z' / d'_{\ell_i}\right) = \theta_{\ell_i}$$
$$\phi'_{\ell_i} = \text{atan2}\left((r_{\ell'_i} - r_{\ell'_1}) \cdot y', (r_{\ell'_i} - r_{\ell'_1}) \cdot x'\right) = \text{atan2}\left((g(r_{\ell_i}) - g(r_{\ell_1})) \cdot y', (g(r_{\ell_i}) - g(r_{\ell_1})) \cdot x'\right) = \phi_{\ell_i}$$

Therefore, we show that $f(R) = f(g(R))$. Next, we consider the function $f^{-1}(\cdot)$. For all $i$, the three terms in $f^{-1}(S)_{\ell_i}$ are

$$d_{\ell_i} \sin(\theta_{\ell_i}) \cos(\phi_{\ell_i})$$
$$d_{\ell_i} \sin(\theta_{\ell_i}) \sin(\phi_{\ell_i}) \tag{4}$$
$$d_{\ell_i} \cos\theta_{\ell_i}$$

Then we have $r_{\ell_i} = f^{-1}(S)_{\ell_i}^T F + r_{\ell_0}$. Therefore, we show that there exist a $SE(3)$ transformation $g$, such that $g(f^{-1}(S)) = R$. $\qquad\square$

## B.3   PROOF OF THEOREM 3.4

First we establish a lemma and provide its proof.

**Lemma B.2.** *Let $G_1 = (z_1, R_1)$ and $G_2 = (z_2, R_2)$ be two 3D graphs, where $z_i$ is the node type vector and $R_i$ is the node coordinate matrix of the molecule $G_i$ for $i = 1, 2$. Let $L(G)$ be the canonical label of graph $G$. We have $G_1 \cong G_2$. Let $\ell_i$ and $\ell'_i$ denote the indexes of the node labeled $i$ correspondingly in $L(G_1)$ and $L(G_2)$, respectively. Let $F$ be the equivariant global frame of graph $G$ built based on the first three non-collinear atoms in $L(G)$. Let $f : \mathcal{G} \to \mathcal{S}$ be a surjective function that maps a 3D graph $G \in \mathcal{G}$ to its spherical representations $S = f(G) \in \mathcal{S}$ under the equivariant global frame $F$. Then the following equivalence holds:*

$$\forall i \in V_1, f(G_1)_{\ell_i} = f(G_2)_{\ell'_i} \iff G_1 \cong_{3D} G_2$$

*where $G_1 \cong_{3D} G_2$ denotes that the graphs $G_1$ and $G_2$ are 3D isomorphic.*

*Proof.* Let $\boldsymbol{L}(G)$ be the canonical labeling of graph $G$. Let $\ell_i$ and $\ell_i'$ denote the index of the node labeled $i$ correspondingly in $\boldsymbol{L}(G_1)$ and $\boldsymbol{L}(G_2)$, respectively. We have

$$G_1 \cong_{3D} G_2 \iff \begin{cases} G_1 \cong G_2, \text{ and} \\ \text{there exists a 3D transformation } g \in SE(3) \text{ such that } \boldsymbol{r}_{\ell_i'}^{G_2} = g(\boldsymbol{r}_{\ell_i}^{G_1}). \end{cases}$$

Specifically, $g(\boldsymbol{r}_{\ell_i}) = \boldsymbol{Q}\boldsymbol{r}_{\ell_i} + \boldsymbol{b}$. Here $\boldsymbol{Q}$ is a rotation matrix, and $\boldsymbol{b}$ is a translation vector.

Let $\ell_1, \ell_2$, and $\ell_F$ be the indices of the first three non-collinear atoms in $G_1$. Then the equivariant global frame $\boldsymbol{F}_1 = (\boldsymbol{x}_1, \boldsymbol{y}_1, \boldsymbol{z}_1)$ is

$$\boldsymbol{x}_1 = \text{normalize}(\boldsymbol{r}_{\ell_2} - \boldsymbol{r}_{\ell_1})$$
$$\boldsymbol{y}_1 = \text{normalize}\left((\boldsymbol{r}_{\ell_F} - \boldsymbol{r}_{\ell_1}) \times \boldsymbol{x}_1\right)$$
$$\boldsymbol{z}_1 = \boldsymbol{x}_1 \times \boldsymbol{y}_1$$

Here normalize($\cdot$) is the function to normalize a vector to the corresponding unit vector. Then $\forall i$, the spherical representations $f(G_1)_{\ell_i}$ is

$$d_{\ell_i} = \|\boldsymbol{r}_{\ell_i} - \boldsymbol{r}_{\ell_1}\|_2$$
$$\theta_{\ell_i} = \arccos\left((\boldsymbol{r}_{\ell_i} - \boldsymbol{r}_{\ell_1}) \cdot \boldsymbol{z}_1/d_{\ell_i}\right)$$
$$\phi_{\ell_i} = \text{atan2}\left((\boldsymbol{r}_{\ell_i} - \boldsymbol{r}_{\ell_1}) \cdot \boldsymbol{y}_1, (\boldsymbol{r}_{\ell_i} - \boldsymbol{r}_{\ell_1}) \cdot \boldsymbol{x}_1\right)$$

Similarly, for $G_2$, let $\ell_1', \ell_2'$, and $\ell_F'$ be the indices of the first three non-collinear atoms. Then the equivariant global frame $\boldsymbol{F}_2 = (\boldsymbol{x}_2, \boldsymbol{y}_2, \boldsymbol{z}_2)$ is

$$\boldsymbol{x}_2 = \text{normalize}(\boldsymbol{r}_{\ell_2'} - \boldsymbol{r}_{\ell_1'}) = \text{normalize}(g(\boldsymbol{r}_{\ell_2}) - g(\boldsymbol{r}_{\ell_1})) = \boldsymbol{Q}\boldsymbol{x}_1$$
$$\boldsymbol{y}_2 = \text{normalize}\left((\boldsymbol{r}_{\ell_F'} - \boldsymbol{r}_{\ell_1'}) \times \boldsymbol{x}_2\right) = \text{normalize}\left((g(\boldsymbol{r}_{\ell_F}) - g(\boldsymbol{r}_{\ell_1})) \times \boldsymbol{x}_2\right) = \boldsymbol{Q}\boldsymbol{y}_1$$
$$\boldsymbol{z}_2 = \boldsymbol{x}_2 \times \boldsymbol{y}_2 = (\boldsymbol{Q}\boldsymbol{x}_1) \times (\boldsymbol{Q}\boldsymbol{y}_1) = \boldsymbol{Q}\boldsymbol{z}_1$$

Then $\forall i$, the spherical representations $f(G_2)_{\ell_i'}$ is

$$d_{\ell_i'} = \|\boldsymbol{r}_{\ell_i'} - \boldsymbol{r}_{\ell_1'}\|_2 = \|g(\boldsymbol{r}_{\ell_i}) - g(\boldsymbol{r}_{\ell_1})\|_2 = d_{\ell_i}$$
$$\theta_{\ell_i'} = \arccos\left((\boldsymbol{r}_{\ell_i}' - \boldsymbol{r}_{\ell_1}') \cdot \boldsymbol{z}_2/d_{\ell_i'}\right) = \arccos\left((g(\boldsymbol{r}_{\ell_i}) - g(\boldsymbol{r}_{\ell_1})) \cdot \boldsymbol{z}_2/d_{\ell_i'}\right) = \theta_{\ell_i}$$
$$\phi_{\ell_i'} = \text{atan2}\left((\boldsymbol{r}_{\ell_i'} - \boldsymbol{r}_{\ell_1'}) \cdot \boldsymbol{y}_2, (\boldsymbol{r}_{\ell_i'} - \boldsymbol{r}_{\ell_1'}) \cdot \boldsymbol{x}_2\right) = \text{atan2}\left((g(\boldsymbol{r}_{\ell_i}) - g(\boldsymbol{r}_{\ell_1})) \cdot \boldsymbol{y}_2, (g(\boldsymbol{r}_{\ell_i}) - g(\boldsymbol{r}_{\ell_1})) \cdot \boldsymbol{x}_2\right) = \phi_{\ell_i}$$

Therefore, we show that $G_1 \cong_{3D} G_2 \iff \forall i, \in V_1, f(G_1)_{\ell_i} = f(G_2)_{\ell_i'}$ holds. $\square$

Then we prove Theorem 3.4.

**Theorem** (Bijective mapping between 3D graph isomorphism and sequence)**.** *Let $G_1 = (\boldsymbol{z}_1, \boldsymbol{R}_1)$ and $G_2 = (\boldsymbol{z}_2, \boldsymbol{R}_2)$ be two 3D graphs, where $\boldsymbol{z}_j$ is the node type vector and $\boldsymbol{R}_j$ is the node coordinate matrix of the molecule $G_j$ for $j = 1, 2$. Let $\boldsymbol{L}_m(G)$ be the canonical label for 3D graph and $f : \mathcal{G} \to \mathcal{S}$ be the function that maps a 3D graph $G$ to its spherical representations. Given graph $G$ with $n$ nodes and $\boldsymbol{X} = [\boldsymbol{x}_1, ..., \boldsymbol{x}_n]^T \in \mathbb{R}^{n \times m}$, where $m \in \mathbb{Z}$, we define $\boldsymbol{L}_m(G) \otimes \boldsymbol{X} = concat(\boldsymbol{x}_{\ell_1}, ..., \boldsymbol{x}_{\ell_n})$, where $\ell_i$ is the node index of the node labeled $i$ in $\boldsymbol{L}_m(G)$, and $concat(\cdot)$ concatenates elements as a sequence. Define*

$$Geo2Seq(G_i) = \boldsymbol{L}_m(G) \otimes (\boldsymbol{z}, f(G)) = \boldsymbol{L}_m(G) \otimes \boldsymbol{X},$$

*where $\boldsymbol{x}_i = [z_i, d_i, \theta_i, \phi_i]$. Then $Geo2Seq : \mathcal{G} \to \mathcal{U}$ is a surjective function, and the following equivalence holds:*

$$Geo2Seq(G_1) = Geo2Seq(G_2) \iff G_1 \cong_{3D} G_2$$

*where $G_1 \cong_{3D} G_2$ denotes that the graphs $G_1$ and $G_2$ are 3D isomorphic.*

*Proof.* First, we prove that $Geo2Seq : \mathcal{G} \to \mathcal{U}$ is a surjective function. Given the definition

$$Geo2Seq(G_i) = \boldsymbol{L}_m(G) \otimes (\boldsymbol{z}, f(G)) = \boldsymbol{L}_m(G) \otimes \boldsymbol{X},$$

where $\boldsymbol{x}_i = [z_i, d_i, \theta_i, \phi_i]$, we need to prove that all operations are deterministic. $\otimes$ and $\boldsymbol{z}$ are defined to be deterministic, and $f : \mathcal{G} \to \mathcal{S}$ is a function. $\boldsymbol{L}_m(G_j)$ outputs the automorphism group of $G_j$'s canonical label. By definition, the automorphism group contain different labels of the strictly

identical graph. Let $\ell_i$ and $\ell_i'$ describe two different sets of labels of the same automorphism group with $n$ nodes; since the graphs are identical,

$$[z_{\ell_i}, d_{\ell_i}, \theta_{\ell_i}, \phi_{\ell_i}] = [z_{\ell_i'}, d_{\ell_i'}, \theta_{\ell_i'}, \phi_{\ell_i'}] \text{for} i = 1, ..., n.$$

Thus $\text{concat}(\boldsymbol{x}_{\ell_1}, ..., \boldsymbol{x}_{\ell_n}) = \text{concat}(\boldsymbol{x}_{\ell_1'}, ..., \boldsymbol{x}_{\ell_n'})$, *i.e.*, different labels of one automorphism group produce identical sequences with Geo2Seq. Therefore, Geo2Seq $: \mathcal{G} \to \mathcal{U}$ is a well-defined function; given a 3D molecule, we can uniquely construct a 1D sequence from Geo2Seq.

Next we prove Geo2Seq's surjectivity. Given any output sequence $q \in \mathcal{U}$ of Geo2Seq, the sequence is in the format

$$q = concat([z_1, d_1, \theta_1, \phi_1], ..., [z_n, d_n, \theta_n, \phi_n]).$$

For the nodes in $q$, we denote with $S = [[d_1, \theta_1, \phi_1], ..., [d_n, \theta_n, \phi_n]]$. Given the surjectivity of the spherical representation function $f : \mathcal{G} \to \mathcal{S}$ and the defined $f^{-1} : \mathcal{S} \to \mathcal{G}$, there must be a unique $G(\boldsymbol{z}, \boldsymbol{R}) \in \mathcal{G}$ where $S = f(G)$. Therefore, $\forall$ output sequence $q \in \mathcal{U}$ there exists

$$G(\boldsymbol{z}, \boldsymbol{R}) \in \mathcal{G} \quad s.t. \quad q = \text{Geo2Seq}(G),$$

*i.e.*, Geo2Seq is surjective; given a sequence output of Geo2Seq, we can uniquely reconstruct a 3D molecule.

Now we prove the equivalence $\text{Geo2Seq}(G_i) = Geo2Seq(G_i) \iff G_1 \cong_{3D} G_2$, starting from right to left. Considering Lemma 3.1 for molecule $G = (\boldsymbol{z}, \boldsymbol{R})$, we specify $G = (V, E, A)$ with $A = [\boldsymbol{z}, \boldsymbol{R}]$ and define the CL function for 3D molecule graphs as $\boldsymbol{L}_m$, which extends the equivalence in Lemma 3.1 to $\boldsymbol{L}_m$ on molecules with 3D isomorphism. If $G_1 \cong_{3D} G_2$, *i.e.*, graphs $G_1$ and $G_2$ are 3D isomorphic, then from Lemma 3.1 we know the canonical forms $\boldsymbol{L}_m(G_1) = \boldsymbol{L}_m(G_2)$. Let graphs $G_1$ and $G_2$ have numbers of node $n$. Let $\ell_i$ and $\ell_i'$ be the denotations of a corresponding pair of canonical labelings from $\boldsymbol{L}_m(G_1)$ and $\boldsymbol{L}_m(G_2)$, respectively. Since graphs $G_1$ and $G_2$ are 3D isomorphic, from Def.3.3 we know $\forall i \in \mathcal{V}(G_1), z_{\ell_i} = z_{l_{i'}}$; and from Lemma B.2 we know $\forall i \in \mathcal{V}(G_1), f(G_1)_{\ell_i} = f(G_2)_{\ell_i'}$. Thus, we have

$$\begin{aligned}
\text{Geo2Seq}(G_1) &= \boldsymbol{L}_m(G_1) \otimes (\boldsymbol{z}_1, f(G_1)) \\
&= \text{concat}_{z_j \in \boldsymbol{z}_1, d_j, \theta_j, \phi_j \in f(G_1), i=1, ...n}([z_{\ell_i}, d_{\ell_i}, \theta_{\ell_i}, \phi_{\ell_i}]) \\
&= \text{concat}_{z_j \in \boldsymbol{z}_2, d_j, \theta_j, \phi_j \in f(G_2), i=1, ...n}([z_{\ell_i'}, d_{\ell_i'}, \theta_{\ell_i'}, \phi_{\ell_i'}]) \\
&= \boldsymbol{L}_m(G_2) \otimes (\boldsymbol{z}_2, f(G_2)) = \text{Geo2Seq}(G_2).
\end{aligned} \tag{5}$$

Note that if $\boldsymbol{L}_m(G_1)$ and $\boldsymbol{L}_m(G_2)$ contain automorphism groups larger than 1, we can include all possible labelings, which will all produce the same sequence later through Geo2Seq, as we have shown in detail above. However, this is a very rare case for real-world 3D graphs like molecules. Therefore, we have shown that if two molecules are 3D isomorphic considering atoms, bonds, and coordinates, their sequences resulting from Geo2Seq must be identical.

Finally, we prove the equivalence from left to right. We provide proof by contradiction. Given that $\text{Geo2Seq}(G_1) = Geo2Seq(G_2)$, we assume that the graphs $G_1$ and $G_2$ are not 3D isomorphic. We denote with $G_1 = (\boldsymbol{z}_1, \boldsymbol{R}_1)$ and $G_2 = (\boldsymbol{z}_2, \boldsymbol{R}_2)$. If $G_1$ and $G_2$ are not even isomorphic for $A_i = \boldsymbol{z}_i$, then from Def.B.1, there does not exist a node-to-node mapping from $G_1$ to $G_2$, where each node is identically attributed and connected. And from Lemma 3.1, we know the canonical forms $\boldsymbol{L}_m(G_1) \neq \boldsymbol{L}_m(G_2)$. Thus for

$$\text{Geo2Seq}(G_1) = \text{concat}_{z_j \in \boldsymbol{z}_1, d_j, \theta_j, \phi_j \in f(G_1), i=1, ...n}([z_{\ell_i}, d_{\ell_i}, \theta_{\ell_i}, \phi_{\ell_i}]),$$

and

$$\text{Geo2Seq}(G_2) = \text{concat}_{z_j \in \boldsymbol{z}_2, d_j, \theta_j, \phi_j \in f(G_2), i=1, ...n}([z_{\ell_i'}, d_{\ell_i'}, \theta_{\ell_i'}, \phi_{\ell_i'}]),$$

there must be at least one pair of $z_{\ell_i}, z_{\ell_i'}$ where $z_{\ell_i} \neq z_{\ell_i'}$. Therefore, $\text{Geo2Seq}(G_1) \neq \text{Geo2Seq}(G_2)$, which is a contradiction to the initial condition that $\text{Geo2Seq}(G_1) = Geo2Seq(G_2)$ and ends the proof.

If $G_1$ and $G_2$ are isomorphic for $A_i = \boldsymbol{z}_i$, we continue with the following analyses. Let $\ell_i$ and $\ell_i'$ be the denotations of a corresponding pair of canonical labelings from $\boldsymbol{L}_m(G_1)$ and $\boldsymbol{L}_m(G_2)$, respectively. Let $f : \mathcal{G} \to \mathcal{S}$ be the surjective function mapping a 3D graph to its spherical representations. Since $G_1$ and $G_2$ are not 3D isomorphic, from Lemma B.2, we know there exists at least one

$$i \in V_1, s.t. f(G_1)_{\ell_i} \neq f(G_2)_{\ell_i'};$$

otherwise, we would have

$$\forall i \in V_1, f(G_1)_{\ell_i} = f(G_2)_{\ell'_i} \Rightarrow G_1 \cong_{3D} G_2,$$

contradicting the above condition. Thus for

$$\text{Geo2Seq}(G_1) = \text{concat}_{z_j \in \boldsymbol{z}_1, d_j, \theta_j, \phi_j \in f(G_1), i=1,\dots n}([z_{\ell_i}, d_{\ell_i}, \theta_{\ell_i}, \phi_{\ell_i}]),$$

and

$$\text{Geo2Seq}(G_2) = \text{concat}_{z_j \in \boldsymbol{z}_2, d_j, \theta_j, \phi_j \in f(G_2), i=1,\dots n}([z_{\ell'_i}, d_{\ell'_i}, \theta_{\ell'_i}, \phi_{\ell'_i}]),$$

$G_1$ and $G_2$ are isomorphic, so

$$\forall i = 1, \dots n, z_{\ell_i} = z_{\ell'_i};$$

at least one pair of spherical coordinates does not correspond, so there must be at least one pair of $(d_{\ell_i}, \theta_{\ell_i}, \phi_{\ell_i})$ and $(d_{\ell'_i}, \theta_{\ell'_i}, \phi_{\ell'_i})$ where

$$(d_{\ell_i}, \theta_{\ell_i}, \phi_{\ell_i}) \neq (d_{\ell'_i}, \theta_{\ell'_i}, \phi_{\ell'_i}).$$

Thus, $\text{Geo2Seq}(G_1) \neq \text{Geo2Seq}(G_2)$, which contradicts the initial condition that $\text{Geo2Seq}(G_1) = Geo2Seq(G_2)$. Therefore, we have shown that if two constructed sequences from Geo2Seq are identical, their corresponding molecules must be 3D isomorphic considering atoms, bonds, and coordinates. This ends the proof.

$\square$

## B.4  PROOF OF COROLLARY 3.5

**Corollary** (Constrained bijective Mapping between 3D graph and sequence). *Let $G_1 = (\boldsymbol{z}_1, \boldsymbol{R}_1)$ and $G_2 = (\boldsymbol{z}_2, \boldsymbol{R}_2)$ be two 3D graphs, where $\boldsymbol{z}_j$ is the node type vector and $\boldsymbol{R}_j$ is the node coordinate matrix of the molecule $G_j$ for $j = 1, 2$. Let $\boldsymbol{L}_m(G)$ be the canonical labeling for 3D graph and $f : \mathcal{G} \to \mathcal{S}$ be the function that maps a 3D graph $G$ to its spherical representations. Given graph $G$ with $n$ nodes and $\boldsymbol{X} = [\boldsymbol{x}_1, \dots, \boldsymbol{x}_n] \in \mathbb{R}^{n \times m}$, where $m \in \mathbb{Z}$, we define $\boldsymbol{L}_m(G) \otimes \boldsymbol{X} = concat(\boldsymbol{x}_{\ell_1}, \dots, \boldsymbol{x}_{\ell_n})$, where $\ell_i$ is the node index of the node labeled $i$ in $\boldsymbol{L}_m(G)$, and $concat(\cdot)$ concatenates elements as a sequence. Define*

$$Geo2Seq(G_i) = \boldsymbol{L}_m(G) \otimes (\boldsymbol{z}, f(G)) = \boldsymbol{L}_m(G) \otimes \boldsymbol{X},$$

*where $\boldsymbol{x}_i = [z_i, d_i, \theta_i, \phi_i]$. Let the truncation of spherical coordinate values be after $b$ decimal digits. Then $Geo2Seq : \mathcal{G} \to \mathcal{U}$ is a surjective function, and the following equivalence holds:*

$$Geo2Seq(G_i) = Geo2Seq(G_i) \iff G_1 \cong_{3D - \frac{\lfloor 10^{-b} \rfloor}{2}} G_2$$

*where $G_1 \cong_{3D - \frac{\lfloor 10^{-b} \rfloor}{2}} G_2$ denotes that the graphs $G_1$ and $G_2$ are $\frac{\lfloor 10^{-b} \rfloor}{2}$-constrained 3D isomorphic.*

*Proof.* First, we prove that $Geo2Seq : \mathcal{G} \to \mathcal{U}$ is a surjective function, which resembles the proof for Theorem 3.4. Given the definition

$$\text{Geo2Seq}(G_i) = \boldsymbol{L}_m(G) \otimes (\boldsymbol{z}, f(G)) = \boldsymbol{L}_m(G) \otimes \boldsymbol{X},$$

where $\boldsymbol{x}_i = [z_i, d_i, \theta_i, \phi_i]$, we need to prove that all operations are deterministic. $\otimes$ and $\boldsymbol{z}$ are defined to be deterministic, and $f : \mathcal{G} \to \mathcal{S}$ with truncation after certain decimal places is still a well-defined function. $\boldsymbol{L}_m(G_j)$ outputs the automorphism group of $G_j$'s canonical label. By definition, the automorphism group contain different labels of the strictly identical graph. Let $\ell_i$ and $\ell'_i$ describe two different sets of labels of the same automorphism group with $n$ nodes; since the graphs are identical,

$$[z_{\ell_i}, d_{\ell_i}, \theta_{\ell_i}, \phi_{\ell_i}] = [z_{\ell'_i}, d_{\ell'_i}, \theta_{\ell'_i}, \phi_{\ell'_i}] \text{for} i = 1, \dots, n.$$

Thus $\text{concat}(\boldsymbol{x}_{\ell_1}, \dots, \boldsymbol{x}_{\ell_n}) = \text{concat}(\boldsymbol{x}_{\ell'_1}, \dots, \boldsymbol{x}_{\ell'_n})$, *i.e.*, different labels of one automorphism group produce identical sequences with Geo2Seq. Therefore, Geo2Seq : $\mathcal{G} \to \mathcal{U}$ is still a well-defined function; given a 3D molecule, we can uniquely construct a 1D sequence from Geo2Seq.

Next we prove Geo2Seq's surjectivity. Given any output sequence $q \in \mathcal{U}$ of Geo2Seq, the sequence is in the format

$$q = concat([z_1, d_1, \theta_1, \phi_1], \dots, [z_n, d_n, \theta_n, \phi_n]).$$

For the nodes in $q$, we define $S_{trun} = [[d_1, \theta_1, \phi_1], ..., [d_n, \theta_n, \phi_n]]$. Given the surjectivity of the spherical representation function $f : \mathcal{G} \to \mathcal{S}$ and the defined $f^{-1} : \mathcal{S} \to \mathcal{G}$, there must be a unique $G(\boldsymbol{z}, \boldsymbol{R}) \in \mathcal{G}$ where $S_{trun} = f(G)$. Therefore, $\forall$ output sequence $q \in \mathcal{U}$ there exists

$$G(\boldsymbol{z}, \boldsymbol{R}) \in \mathcal{G} \quad s.t. \quad q = \text{Geo2Seq}(G),$$

*i.e.*, Geo2Seq is surjective; given a sequence output of Geo2Seq, we can uniquely reconstruct a 3D molecule.

Now we prove the equivalence $\text{Geo2Seq}(G_i) = Geo2Seq(G_i) \iff G_1 \cong_{3D} G_2$, starting from right to left. When a number is truncated after $b$ decimal places, according to the rounding principle, the maximum error caused is $\boldsymbol{\epsilon} \leq \frac{|10^{-b}|}{2}$. Considering Lemma 3.1 for molecule $G = (\boldsymbol{z}, \boldsymbol{R})$, we specify $G = (V, E, A)$ with $A = [\boldsymbol{z}, \boldsymbol{R}]$ and define the CL function for 3D molecule graphs as $\boldsymbol{L}_m$, which extends the equivalence in Lemma 3.1 to $\boldsymbol{L}_m$ on molecules with 3D isomorphism. If $G_1 \cong_{3D - \frac{|10^{-b}|}{2}} G_2$, *i.e.*, graphs $G_1$ and $G_2$ are $\frac{|10^{-b}|}{2}$-constrained 3D isomorphic, then from Lemma 3.1 we know $G_1$ and $G_2$ are still isomorphic for $A_i = \boldsymbol{z}_i$, and the canonical forms $\boldsymbol{L}_m(G_1) = \boldsymbol{L}_m(G_2)$. Let graphs $G_1$ and $G_2$ have numbers of node $n$. Let $\ell_i$ and $\ell'_i$ be the denotations of a corresponding pair of canonical labelings from $\boldsymbol{L}_m(G_1)$ and $\boldsymbol{L}_m(G_2)$, respectively. Since graphs $G_1$ and $G_2$ are $\frac{|10^{-b}|}{2}$-constrained 3D isomorphic, from Def.3.3 we know $\forall i \in \mathcal{V}(G_1), z_{\ell_i} = z_{l'_i}$; and from Lemma B.2 we know $\forall i \in \mathcal{V}(G_1), f(G_1)_{\ell_i} = f(G_2)_{\ell'_i}$ with $\frac{|10^{-b}|}{2}$ error range allowed for each numerical value. Thus, we still have

$$\begin{aligned}
\text{Geo2Seq}(G_1) &= \boldsymbol{L}_m(G_1) \otimes (\boldsymbol{z}_1, f(G_1)) \\
&= \text{concat}_{z_j \in \boldsymbol{z}_1, d_j, \theta_j, \phi_j \in f(G_1), i=1,...n}([z_{\ell_i}, d_{\ell_i}, \theta_{\ell_i}, \phi_{\ell_i}]) \\
&= \text{concat}_{z_j \in \boldsymbol{z}_2, d_j, \theta_j, \phi_j \in f(G_2), i=1,...n}([z_{\ell'_i}, d_{\ell'_i}, \theta_{\ell'_i}, \phi_{\ell'_i}]) \\
&= \boldsymbol{L}_m(G_2) \otimes (\boldsymbol{z}_2, f(G_2)) = \text{Geo2Seq}(G_2).
\end{aligned} \tag{6}$$

Note that if $\boldsymbol{L}_m(G_1)$ and $\boldsymbol{L}_m(G_2)$ contain automorphism groups larger than 1, we can include all possible labelings, which will all produce the same sequence later through Geo2Seq, as we have shown in detail above. However, this is a very rare case for real-world 3D graphs like molecules. Therefore, we have shown that if two molecules are 3D isomorphic considering atoms, bonds, and coordinates within the round-up error range $\frac{|10^{-b}|}{2}$, their sequences resulting from Geo2Seq must be identical.

Finally, we prove the equivalence from left to right. We provide proof by contradiction. Given that $\text{Geo2Seq}(G_1) = Geo2Seq(G_2)$, we assume that the graphs $G_1$ and $G_2$ are not $\frac{|10^{-b}|}{2}$-constrained 3D isomorphic. We denote with $G_1 = (\boldsymbol{z}_1, \boldsymbol{R}_1)$ and $G_2 = (\boldsymbol{z}_2, \boldsymbol{R}_2)$. If $G_1$ and $G_2$ are not even isomorphic for $A_i = \boldsymbol{z}_i$, then from Def.B.1, there does not exist a node-to-node mapping from $G_1$ to $G_2$, where each node is identically attributed and connected. And from Lemma 3.1, we know the canonical forms $\boldsymbol{L}_m(G_1) \neq \boldsymbol{L}_m(G_2)$. Thus for

$$\text{Geo2Seq}(G_1) = \text{concat}_{z_j \in \boldsymbol{z}_1, d_j, \theta_j, \phi_j \in f(G_1), i=1,...n}([z_{\ell_i}, d_{\ell_i}, \theta_{\ell_i}, \phi_{\ell_i}]),$$

and

$$\text{Geo2Seq}(G_2) = \text{concat}_{z_j \in \boldsymbol{z}_2, d_j, \theta_j, \phi_j \in f(G_2), i=1,...n}([z_{\ell'_i}, d_{\ell'_i}, \theta_{\ell'_i}, \phi_{\ell'_i}]),$$

there must be at least one pair of $z_{\ell_i}, z_{\ell'_i}$ where $z_{\ell_i} \neq z_{\ell'_i}$. Therefore, $\text{Geo2Seq}(G_1) \neq \text{Geo2Seq}(G_2)$, which is a contradiction to the initial condition that $\text{Geo2Seq}(G_1) = Geo2Seq(G_2)$ and ends the proof.

If $G_1$ and $G_2$ are isomorphic for $A_i = \boldsymbol{z}_i$, we continue with the following analyses. Let $\ell_i$ and $\ell'_i$ be the denotations of a corresponding pair of canonical labelings from $\boldsymbol{L}_m(G_1)$ and $\boldsymbol{L}_m(G_2)$, respectively. Let $f : \mathcal{G} \to \mathcal{S}$ be the surjective function mapping a 3D graph to its spherical representations. Since $G_1$ and $G_2$ are not $\frac{|10^{-b}|}{2}$-constrained 3D isomorphic, from Lemma B.2, we know there exists at least one

$$i \in V_1, s.t. f(G_1)_{\ell_i} \neq f(G_2)_{\ell'_i},$$

even with error range $\frac{|10^{-b}|}{2}$ allowed; otherwise, we would have

$$\forall i \in V_1, f(G_1)_{\ell_i} = f(G_2)_{\ell'_i} \Rightarrow G_1 \cong_{3D} G_2,$$

contradicting the above condition. Thus for

$$\text{Geo2Seq}(G_1) = \text{concat}_{z_j \in \boldsymbol{z}_1, d_j, \theta_j, \phi_j \in f(G_1), i=1,...n}([z_{\ell_i}, d_{\ell_i}, \theta_{\ell_i}, \phi_{\ell_i}]),$$

and

$$\text{Geo2Seq}(G_2) = \text{concat}_{z_j \in \boldsymbol{z}_2, d_j, \theta_j, \phi_j \in f(G_2), i=1,...n}([z_{\ell'_i}, d_{\ell'_i}, \theta_{\ell'_i}, \phi_{\ell'_i}]),$$

$G_1$ and $G_2$ are isomorphic, so

$$\forall i = 1, ...n, z_{\ell_i} = z_{\ell'_i};$$

at least one pair of spherical coordinates does not correspond, so there must be at least one pair of $(d_{\ell_i}, \theta_{\ell_i}, \phi_{\ell_i})$ and $(d_{\ell'_i}, \theta_{\ell'_i}, \phi_{\ell'_i})$ where

$$\min(|d_{\ell_i}, \theta_{\ell_i}, \phi_{\ell_i}| - |d_{\ell'_i}, \theta_{\ell'_i}, \phi_{\ell'_i}|) > \frac{|10^{-b}|}{2}.$$

Thus $\text{Geo2Seq}(G_1) \neq \text{Geo2Seq}(G_2)$, which contradicts the initial condition that $\text{Geo2Seq}(G_1) = Geo2Seq(G_2)$. Therefore, we have shown that if two constructed sequences from Geo2Seq are identical, their corresponding molecules must be 3D isomorphic considering atoms, bonds, and coordinates within the round-up error range $\frac{|10^{-b}|}{2}$. This ends the proof.

$\square$

## C  ABLATION STUDIES

Table 3: Random generation performance with different atom generation orders.

| Order | Atom Sta (%) | Mol Sta (%) | Valid (%) | Valid & Unique (%) |
|---|---|---|---|---|
| Canonical-locality | **97.39** | **86.77** | **92.97** | **84.71** |
| Canonical-nonlocality | 96.45 | 81.36 | 90.89 | 83.37 |
| Canonical-SMILES | 97.35 | 85.86 | 92.97 | 84.05 |
| DFS (Thomas et al., 2009) | 95.95 | 81.54 | 90.45 | 82.48 |
| BFS (Lee, 1961) | 96.85 | 80.92 | 90.49 | 76.13 |
| Dijkstra (Dijkstra, 2022) | 95.29 | 77.25 | 88.97 | 73.52 |
| Cuthill–McKee (Cuthill & McKee, 1969) | 93.56 | 71.57 | 85.36 | 76.23 |
| Hilbert-curve (Hilbert & Hilbert, 1935) | 90.11 | 64.99 | 80.40 | 67.83 |
| Random | 64.87 | 20.14 | 43.16 | 38.44 |

Table 4: Random generation performance with different 3D representations.

| 3D representation | Atom Sta (%) | Mol Sta (%) | Valid (%) | Valid & Unique (%) |
|---|---|---|---|---|
| Original coordinates | 91.1 | 58.1 | 75.6 | 55.1 |
| Normalized coordinates | 92.7 | 63.2 | 83.1 | 72.5 |
| Invariant Cartesian coordinates | 96.0 | 78.5 | 89.7 | 74.1 |
| Inv-spherical coordinates | **97.3** | **83.4** | 91.0 | **82.7** |
| Inv-spherical coordinates-local distances | 97.1 | 82.8 | **91.7** | 79.6 |

Table 5: Random generation performance with different tokenization.

| Tokenization | Atom Sta (%) | Mol Sta (%) | Valid (%) | Valid & Unique (%) |
|---|---|---|---|---|
| Char-tokenization | 90.5 | 43.7 | 71.5 | 71.0 |
| BPE | 85.3 | 55.3 | 74.4 | 57.6 |
| Sub-tokenization | 96.4 | 80.3 | 89.9 | 74.4 |
| Comp-tokenization | **97.0** | **82.2** | **91.0** | **75.5** |

To study the effects of atom order, 3D representations and tokenization of Geo2Seq on the generation performance of LLMs, we conduct a series of ablation experiments. Among all ablation experiments,

we train 8-layer GPT models on QM9 dataset for 250 epochs with the same hyperparameters as Section 5.1 and use the random generation metrics in Section 5.1 to compare the performance under different settings.

**Ablation on atom order.** First, we show that our proposed canonical order of atoms in Geo2Seq sequence representation is significant for LLMs to achieve good 3D molecular structure modeling. Specifically, we conduct an extended study of ordering algorithms, comparing our Geo2Seq with alternative canonicalization strategies as well as established traversing baselines. As we specified in Sec 3.1, theoretically, analyses and derivations apply to all rigorous CL algorithms. In the paper, we select Nauty Algorithm because its implementation has the best time efficiency among all existing CL algorithms. We implemented Nauty Algorithm for 3D molecules, where multiple strategies can be applied for the partitioning of graph vertices (a step in Nauty). We compare canonicalization strategies with/without locality considered. Canonicalization with locality considered can lead to better results, due to the importance of neighboring atom interactions in molecular evaluations. Given the similar nature, canonical SMILES produces a very similar ordering with "Nauty with locality", thus close in performances. The traversing baselines includes Breadth-First Search (BFS) (Lee, 1961), Depth-First Search (DFS) (Thomas et al., 2009), Dijkstra's algorithm (Dijkstra, 2022), Cuthill–McKee algorithm (Cuthill & McKee, 1969), and Hilbert curve (Hilbert & Hilbert, 1935). We also compare with a Random sequence representation where atoms are randomly ordered. All the other settings of sequence representations remain the same. As Table 3 shows, canonicalization with locality considered can lead to better results, due to the importance of neighboring atom interactions in molecular evaluations. In addition, we can clearly observe that well-designed canonical ordering as in Geo2Seq significantly outperforms basic traverse strategies and the random order, which validates the significance of canonical order.

**Advantage of Nauty Algorithm.** Note that in the paper, we implement Nauty Algorithm for 3D molecules because: (1) its implementation has the best time efficiency among all existing CL algorithms; (2) it is naturally rigorous. The widely used canonical SMILES is based on the Morgan CL Algorithm, which is proven to be incomplete for isomorphism corner cases (such as two triangles versus one hexagon). While canonical SMILES solve corner cases by manual restrictions, Nauty Algorithm is elegantly rigorous. Still, we emphasize that all rigorous CL algorithms are usable for our method, while our contribution lies in achieving structural completeness and geometric invariance for LM learning of 3D molecules.

**Ablation on 3D representation.** Besides, we explore using different methods to represent 3D molecular structures. We compare the spherical coordinates in Geo2Seq with directly using the 3D Cartesian coordinates of atoms from QM9 xyz data files in sequences. We also study whether normalizing the xyz coordinates is effective by subtracting the xyz coordinates with the mass-center coordinates of each molecule. Additionally, we compare with using the $SE(3)$-invariant Cartesian coordinates that are projected to the equivariant frame proposed in Section 3.2. We also explore adopting to manage distances in a more local scheme, which reduces the scale of the distances. We compare with "local distances", where our "distances to the global frame" are replaced with "relative distances to the previous atom" (except for the first atom) while the angles remain the same. Results in Table 4 demonstrate that LLMs achieve the best performance on spherical coordinates. We believe this is due to that the numerical values of distances and angles of spherical coordinates lie in a smaller region than coordinates, which reduces outliers and makes it easier for LLMs to capture their correlation. Furthermore, both our spherical coordinates and that replaced with local distances achieve comparable results, while outperforming Cartesian coordinates. From these empirical results, we can analyze that the representation of azimuth and polar angles has brought sufficient advantage for LM learning over Cartesian coordinates, thus spherical representations with both distance schemes are showing promising performances. In addition, the similar performances could be attributed to that molecular systems often exhibit localized spatial structures (e.g., compact subunits or functional groups), which naturally constrain distances for most small molecules.

**Advantage of invariant spherical representations.** The above experiments show the superiority of invariant spherical coordinates over invariant Cartesian coordinates. While invariant Cartesian coordinates when our proposed equivariant frame is applied can also $SE(3)$-invariance, spherical coordinates are advantageous in discretized representations. Compared to Cartesian coordinates, spherical coordinate values are bounded in a smaller region, namely, a range of $[0, \pi]$ or $[0, 2\pi]$. Given the same decimal place constraints, spherical coordinates require a smaller vocabulary size, and given the same vocabulary size, spherical coordinates present less information loss. This makes

spherical coordinates advantageous in discretized representations and thus easier to be modeled by LMs. Lemma 3.2 and its proof aim to guarantee the validity that our proposed invariant spherical representations possess $SE(3)$-invariance. We consider it as a part of our theoretical contribution towards the derivation of Theorem 3.4.

**Ablation on tokenization.** Finally, we explore other ways to tokenize real numbers in spherical coordinates. Instead of simply taking the complete real number as a token (Comp-tokenization), we try splitting it by the decimal point and treat every part as an individual token (Sub-tokenization). We also explore the common NLP tokenization method, including treating each character as a token (Char-tokenization) and Byte-Pair Encoding (BPE). We compare these tokenization methods in Table 5. Results show that our used Comp-tokenization leads to better performance. This shows that treating the complete real number as an individual token enables LLMs to capture 3D molecular structures more effectively.

Overall, through a series of ablation experiments, we show that canonical atom order, spherical coordinate representation and Comp-tokenization in Geo2Seq are all very useful in parsing 3D molecules to good sequence representations.

## D    EXPERIMENTAL DETAILS AND ADDITIONAL RESULTS

### D.1    HYPERPARAMETERS AND EXPERIMENTAL DETAILS

In the random generation experiment (Section 5.1), we apply two LMs, GPT (Radford et al., 2018) and Mamba (Gu & Dao, 2023), to our proposed Geo2Seq representations. For GPT models, we adopt the architecture of GPT-1, set the hidden dimension to 768, the number of attention head to 8, and the number of layers to 12 and 14 for QM9 and GEOM-DRUGS datasets, respectively. For Mamba models, we set the hidden dimension to 768 and the number of layers to 26 and 28 for QM9 and GEOM-DRUGS datasets, respectively. On QM9 dataset, we set the batch size to 32, base learning rate to 0.0004, the number of training epochs to 600 and 210 for GPT and Mamba models, respectively. On GEOM-DRUGS dataset, we set the batch size to 32, base learning rate to 0.0004, the number of training epochs to 20 and 25 for GPT and Mamba models, respectively. During model training, we use AdamW (Loshchilov & Hutter, 2019) optimizer and follow the commonly used linear warm up and cosine decay scheduler to adjust learning rates. Specifically, the learning rate first linearly increases from zero to the base learning rate 0.0004 when handling the first 10% of total training tokens, then gradually decreases to 0.00004 by the cosine decay scheduler. Besides, the tokenization of real numbers uses the precision of two and three decimal places for QM9 and GEOM-DRUGS datasets, respectively. In the controllable generation experiment (Section 5.2), we train 16-layer Mamba models for 200 epochs, and all the other hyperparameters and settings are the same as the random generation experiment. Based on data statistics, we set the context length to 512 for QM9 dataset and 744 for GEOM-DRUGS dataset throughout the experiments. All experiments on the QM9 dataset are conducted using a single NVIDIA A6000 GPU. Experiments on the GEOM-DRUGS dataset are deployed on 4 NVIDIA A100 GPUs.

### D.2    LICENSES

We strictly follow all licenses when using the public assets in this work. The QM9 dataset is under license CC-BY 4.0. The GEOM-DRUGS dataset is under license CC0 1.0. The code of EDM, GEOLDM, JODO, and MiDi is under MIT License.

### D.3    EXPERIMENTS ON ADDITIONAL BASELINES AND METRICS

We extend our experiments with two more baselines, JODO (Huang et al., 2023) and MiDi (Vignac et al., 2023), which are diffusion models jointly generating 2D and 3D molecular information. We exclude them in experiments of the main paper, since the setting is not the same as ours. Our method follows works on 3D molecule generation without 2D information, such as bonds.

We extend the metrics of our evaluation for more comprehensive comparisons on random generation of QM9 dataset. We report the percentage of valid, unique and novel molecules, *i.e.*, that are not present in the training set. We also report the percentage of complete molecules in which all atoms

are connected. Following JODO (Huang et al., 2023), we also include 2D metrics. Frechet ChemNet Distance (FCD) measures the distance between the test set and the generated set with the activation of the penultimate layer of ChemNet. Lower FCD values indicate more similarity between the two distributions. Similarity to the nearest neighbor (SNN) calculates an average Tanimoto similarity between the fingerprints of a generated molecule and its closest molecule in the test set. Fragment similarity (Frag) compares the distributions of BRICS fragments in the generated and test sets, and Scaffold similarity (Scaf) compares the frequencies of Bemis-Murcko scaffolds between them. Additionally, we include alignment metrics. For RDKit generated bonds, we compute the Maximum Mean Discrepancy (MMD) distances of the bond length (Bond), bond angle (Angle), and dihedral angle (Dihedral) distributions, and report their mean MMD distances. To ensure fair comparison, we evaluate the metrics of all methods on the generated 3D structures, and use RDKit to convert 3D structures to 2D graphs if needed. We use the same model and settings as the main paper for Geo2Seq, and follow the released codes for the baselines' respective hyperparameter and settings. Table 6 reports the random generation results on QM9 dataset. According to the results, though our model is not designed to directly learn 2D information, the performance of our method is better than or comparable with baseline methods on all metrics including the 2D metrics, which demonstrates the effectiveness of our design.

Table 6: Additional random generation results on QM9 dataset.

| Metric | EDM | GEOLDM | JODO | MiDi | Geo2Seq with Mamba |
|---|---|---|---|---|---|
| Atom Sta (%) | 98.7 | **98.9** | **98.9** | 98.2 | **98.9** |
| Mol Sta (%) | 82.0 | 89.4 | 89.0 | 83.5 | **93.2** |
| Valid (%) | 91.9 | 93.8 | 94.9 | 95.2 | **97.1** |
| Valid & Unique (%) | 90.7 | 91.8 | **92.8** | **92.8** | 81.7 |
| Valid & Unique & Novel (%) | 83.0 | 83.1 | 85.2 | **85.5** | 71.2 |
| Complete (%) | 90.9 | 93.3 | 94.4 | 94.4 | **97.3** |
| Bond Length MMD | 0.18 | 0.12 | 0.27 | 1.09 | **0.08** |
| Bond Angle MMD | 0.04 | 0.04 | 0.05 | 0.05 | **0.04** |
| Dihedral Angle MMD | 0.003 | 0.003 | 0.0022 | 0.0033 | **0.0011** |
| FCD | 1.16 | **0.94** | 1.55 | 1.28 | 2.04 |
| SNN | 0.47 | **0.49** | 0.47 | 0.47 | **0.49** |
| Frag | **0.94** | **0.94** | **0.94** | **0.94** | 0.83 |
| Scaf | 0.29 | 0.33 | 0.25 | 0.26 | **0.38** |

Reporting the percentage of novel molecules is important in showing that language models can generate new molecules instead of merely memorizing the training dataset. Given our improvements on controllable generation is significant, we explore whether the generated molecules are different from the molecules in the training set. Thus we also extend the metric on controllable generation experiments. We use the same model and setting as the main paper. Table 7 presents the novelty results of controllable generation compared with EDM and JODO. Results show that our method achieves reasonably high novelty scores, which demonstrates that our method is not simply memorizing training data.

Table 7: Additional controllable generation results for the percentage of valid, unique, and novel molecules on QM9 dataset.

| Method | $\alpha$ | $\Delta\epsilon$ | $\epsilon_{\text{HOMO}}$ | $\epsilon_{\text{LUMO}}$ | $\mu$ | $C_v$ |
|---|---|---|---|---|---|---|
| EDM | 87.0% | 84.1% | 79.8% | 84.7% | 73.0% | 68.0% |
| JODO | 86.5% | 87.3% | 86.7% | 86.2% | 86.8% | 85.6% |
| Geo2Seq with Mamba | 82.8% | 82.8% | 83.6% | 83.0% | 83.3% | 83.6% |

In addition, following Hoogeboom et al. (2022), we compare negative log-likelihood (NLL) performance on the random generation of QM9 dataset for Geo2Seq and baseline models that reports this metric. For this experiment, we use the same model and setting as the main paper. From Table 8, we can see the performance of our method is better than or comparable with all baseline methods, evidencing the validity of our model.

Table 8: Additional Negative Log Likelihood (NLL) comparisons of random generation on QM9 dataset.

| Method | NLL |
|--------|-----|
| E-NF | -59.7 |
| GDM | -94.7 |
| EDM | -110.7 |
| GEOLDM | **-335.0** |
| Geo2Seq with Mamba | -242.0 |

For more comprehensive comparisons, we also extend to include the metrics of Symphony (Daigavane et al., 2023) in our evaluation. As shown in Table 9,10,11, we compare the performances of baseline methods and Geo2Seq with Mamba on Symphony metrics. Multiple algorithms exist for bond order assignment: `xyz2mol` (Kim & Kim, 2015), OpenBabel (Banck et al., 2011) and a simple lookup table based on empirical pairwise distances in organic compounds (Hoogeboom et al., 2022). We perform the comparison between these algorithms for evaluating machine-learning generated 3D structures. In Table 9, we use each of these algorithms to infer the bonds and create a molecule from generated 3D molecular structure. A molecule is valid if the algorithm could successfully assign bond order with no net resulting charge. We also measure the uniqueness to see how many repetitions were present in the set of SMILES strings of valid generated molecules. Buttenschoen et al. (2023) showed that the predicted 3D structures from machine-learned protein-ligand docking models tend to be highly unphysical. Table 10 utilizes the PoseBusters framework to perform the following sanity checks to count how many of the predicted 3D structures are reasonable. The valid molecules from all models tend to be quite reasonable. Next, we evaluate models on how well they capture bonding patterns and the geometry of local environments found in the training set molecules as Table 11. We utilize the bispectrum (Uhrin, 2021) as a rotationally invariant descriptor of the geometry of local environments. Given a local environment with a central atom $u$, all of the neighbors of $u$ are projected according to the inferred bonds onto the unit sphere $S^2$. Then, the signal $f$ is computed as a sum of Dirac delta distributions along the direction of each neighbor. The bispectrum $\mathcal{B}(f)$ of $f$ is then defined as $\mathcal{B}(f) = \textsc{ExtractScalars}(f \otimes f \otimes f)$. Thus, $f$ captures the distribution of atoms around $u$, and the bispectrum $\mathcal{B}(f)$ captures the geometry of this distribution. The bispectrum varies smoothly when $f$ is varied and is guaranteed to be rotationally invariant. We follow Symphony and compute the bispectrum of local environments with atleast 2 neighboring atoms, and exclude the pseudoscalars in the bispectra. For comparing discrete distributions, we use the symmetric Jensen-Shannon divergence (JSD) as Hoogeboom et al. (2022). Given the true distribution $Q$ and the predicted distribution $P$, the Jensen-Shannon divergence between them is defined as: $D_{JS}(Q \,\|\, P) = \frac{1}{2} D_{KL}(Q \,\|\, M) + \frac{1}{2} D_{KL}(P \,\|\, M)$ where $D_{KL}$ is the Kullback–Leibler divergence and $M = \frac{Q+P}{2}$ is the mean distribution. For continuous distributions, estimating the Jensen-Shannon divergence from samples is tricky without further assumptions on the distributions. We follow Symphony and use the MMD scores to compare samples from continuous distributions. Overall, the performance of our method is better than or comparable with baseline methods across the metrics, showing the effectiveness of our 3D molecule generation.

Table 9: Additional validity and uniqueness percentages of molecules following Symphony.

| Metric ↑ | Symphony | EDM | G-SchNet | G-SphereNet | Geo2Seq |
|----------|----------|-----|----------|-------------|---------|
| Validity via `xyz2mol` | 83.50 | 86.74 | 74.97 | 26.92 | **95.42** |
| Validity via OpenBabel | 74.69 | 77.75 | 61.83 | 9.86 | **83.84** |
| Validity via Lookup Table | 68.11 | 90.77 | 80.13 | 16.36 | **97.55** |
| Uniqueness via `xyz2mol` | 97.98 | **99.16** | 96.73 | 21.69 | 98.88 |
| Uniqueness via OpenBabel | 99.61 | **99.95** | 98.71 | 7.51 | 99.91 |
| Uniqueness via Lookup Table | 97.68 | 98.64 | 93.20 | 23.29 | **98.95** |

Table 10: Percentage of valid molecules passing each PoseBusters test following Symphony.

| Test ↑ | Symphony | EDM | G-SchNet | G-SphereNet | Geo2Seq |
|---|---|---|---|---|---|
| All Atoms Connected | 99.92 | 99.88 | 99.87 | **100.00** | **100.00** |
| Reasonable Bond Angles | 99.56 | **99.98** | 99.88 | 97.59 | 99.90 |
| Reasonable Bond Lengths | 98.72 | **100.00** | 99.93 | 72.99 | **100.00** |
| Aromatic Ring Flatness | **100.00** | **100.00** | 99.95 | 99.85 | 99.98 |
| Double Bond Flatness | 99.07 | 98.58 | 97.96 | 95.99 | **99.45** |
| Reasonable Internal Energy | 95.65 | 94.88 | 95.04 | 36.07 | **96.10** |
| No Internal Steric Clash | 98.16 | **99.79** | 99.57 | 98.07 | 99.33 |

Table 11: Additional comparison statistics of generated molecules to the training set for QM9 dataset following Symphony.

| MMD of Bond Lengths ↓ | Symphony | EDM | G-SchNet | G-SphereNet | Geo2Seq |
|---|---|---|---|---|---|
| C-H: 1.0 | 0.0739 | 0.0653 | 0.3817 | 0.1334 | **0.0488** |
| C-C: 1.0 | 0.3254 | 0.0956 | 0.2530 | 1.0503 | **0.0705** |
| C-O: 1.0 | 0.2571 | 0.0757 | 0.5315 | 0.6082 | **0.0712** |
| C-N: 1.0 | 0.3086 | 0.1755 | 0.2999 | 0.4279 | **0.1056** |
| N-H: 1.0 | 0.1032 | 0.1137 | 0.5968 | 0.1660 | **0.0965** |
| C-O: 2.0 | 0.3033 | 0.0668 | 0.2628 | 2.0812 | **0.0667** |
| O-N: 1.5 | 0.3707 | 0.1736 | 0.5828 | 0.4949 | **0.1570** |
| O-H: 1.0 | 0.2872 | 0.1545 | 0.7899 | 0.1307 | **0.0990** |
| C-C: 1.5 | 0.4142 | 0.1749 | 0.2051 | 0.8574 | **0.0832** |
| C-N: 2.0 | 0.5938 | 0.3237 | 0.4194 | 2.1197 | **0.2676** |
| **MMD of Bispectra ↓** | Symphony | EDM | G-SchNet | G-SphereNet | Geo2Seq |
| C: C2,H2 | 0.2165 | 0.1003 | 0.4333 | 0.6210 | **0.0955** |
| C: C1,H3 | 0.2668 | 0.0025 | 0.0640 | 1.2004 | **0.0011** |
| C: C3,H1 | 0.1111 | 0.2254 | 0.2045 | 1.1209 | **0.0867** |
| C: C2,H1,O1 | 0.1500 | 0.2059 | 0.1732 | 0.8361 | **0.1058** |
| C: C1,H2,O1 | 0.3300 | 0.1082 | 0.0954 | 1.6772 | **0.0802** |
| O: C1,H1 | 0.0282 | 0.0056 | 0.0487 | 0.0030 | **0.0022** |
| C: C2,H1,N1 | 0.1481 | 0.1521 | 0.1967 | 1.3461 | **0.1111** |
| C: C2,H1 | 0.2525 | **0.0468** | 0.1788 | 0.2403 | 0.0851 |
| C: C1,H2,N1 | 0.3631 | 0.2728 | 0.1610 | 0.9171 | **0.1285** |
| N: C2,H1 | **0.0953** | 0.2339 | 0.2105 | 0.6141 | 0.1081 |
| **Jensen-Shannon Divergence ↓** | Symphony | EDM | G-SchNet | G-SphereNet | Geo2Seq |
| Atom Type Counts | 0.0003 | **0.0002** | 0.0011 | 0.0026 | **0.0002** |
| Local Environment Counts | 0.0039 | 0.0057 | 0.0150 | 0.1016 | **0.0035** |

## D.4 GENERATION EFFICIENCY ANALYSIS

We compare the generation efficiency of our method and the diffusion-based methods using a single NVIDIA A100 GPU and a batch size of 32. The results in Table 6 show that our method is much faster than diffusion-based methods, indicating the great efficiency of our method. Though we have take more memory compared to diffusion-based methods, our time efficiency is much better than diffusion-based methods. Throughput, or samples per second, is one of the most important metrics to measure generation efficiency. In particular, Geo2Seq with Mamba is more than 100 times faster than diffusion-based methods, indicating the high throughput of our method, a significant advantage in practical applications where speed is crucial.

Table 12: Generation efficiency comparison between diffusion-based methods and our LM-based method.

| Method | QM9 | | | DRUG | | |
|---|---|---|---|---|---|---|
| | Parameters | Memory | Sample/second | Parameters | Memory | Sample/second |
| EDM | 5.3M | 1.5GB | 1.4 | 2.4M | 7.4GB | 0.1 |
| GeoLDM | 11.4M | 1.5GB | 1.4 | 5.5M | 8.4GB | 0.1 |
| Geo2Seq with GPT | 87.7M | 2.4GB | 8.3 | 105.4M | 3.1GB | 0.2 |
| Geo2Seq with Mamba | 91.8M | 2.2GB | 100.0 | 108.4M | 2.6GB | 16.7 |

## D.5 RESULTS WITH PRETRAINING

To show the advantage of pretraining, we compare the random generation performance on QM9 for models with and without pretraining on Molecule3D (Xu et al., 2021c) dataset, which includes around 4M molecules. Specifically, we conduct experiments on an 8-layer GPT model and a 20-layer Mamba model. The models are pretrained for 20 epochs and then finetuned for 200 epochs. The results in Table 13 demonstrate the advantage of pretraining. Future studies could explore pretraining on larger datasets.

Table 13: Random generation performance on QM9 for models with and without pretraining on Molecule3D dataset.

| Method | Atom Sta (%) | Mol Sta (%) | Valid (%) | Valid & Unique (%) |
|---|---|---|---|---|
| Geo2Seq with GPT | 97.0 | 82.2 | 91.0 | 75.5 |
| Geo2Seq with GPT + pretraining | **98.5** | **89.7** | **94.8** | **76.6** |
| Geo2Seq with Mamba | 97.4 | 86.8 | 93.0 | 78.8 |
| Geo2Seq with Mamba + pretraining | **98.3** | **89.4** | **94.9** | **83.5** |

## E EXTENDED STUDIES

## E.1 SCALING LAWS

Scaling law refers to the relations between functional properties of interest, performance metrics in our case, and properties of the architecture or optimization process. In this section we explore the scaling laws of our models, specifically regarding parameter size, since they provide typical insights for LMs. Scaling laws in 3D molecule generation appears similar to that in NLP. We provide experiments on both GPT and Mamba in Table 14 and 15, respectively. As can be observed, LMs' performances on molecules grow significantly with parameter size increase, similar to the emergence abilities widely-recognized in NLP tasks. As known from NLP studies (Schaeffer et al., 2024), model capabilities grow consistently with model size, while emergence abilities are largely caused by nonlinear metrics. This matches our observations, since the chemical metrics are hardly linear.

Note that we evaluate all models after 250 epochs for fairness concerns, while this fixed hyperparameter setting is not optimal for performances at all parameter sizes. Other settings are the same as the ablation studies.

Table 14: Scaling laws on Geo2Seq with GPT model.

| Parameter size - GPT | 2556532 | 31309824 | 61650944 | 88012800 | 116342688 |
|---|---|---|---|---|---|
| Atom sta(%) | 76.2 | 89.6 | 96.5 | 98.3 | 98.5 |
| Mol sta(%) | 5.1 | 42.4 | 81.3 | 89.1 | 90.6 |
| Valid(%) | 45.5 | 73.1 | 90.9 | 94.3 | 95.1 |
| Valid & Unique(%) | 43.4 | 66.7 | 83.6 | 74.9 | 78.6 |

Table 15: Scaling laws on Geo2Seq with Mamba model.

| Parameter size - Mamba | 2180352 | 31458048 | 61631232 | 93088512 | 121977600 |
|---|---|---|---|---|---|
| Atom sta(%) | 81.6 | 95.7 | 97.4 | 97.8 | 97.9 |
| Mol sta(%) | 13.6 | 79.2 | 86.8 | 88.3 | 89.0 |
| Valid(%) | 51.2 | 89.4 | 93 | 93.7 | 94.4 |
| Valid & Unique(%) | 49.6 | 78.7 | 78.8 | 82.6 | 83.5 |

### E.2 ERROR CASE ANALYSIS

In the natural language domain, trained language models can produce error cases showing repetition or hallucinations. This is also a problem that often arises with LLMs. In this section, we provide the analysis of some error cases to introduce more insights into the field.

Similarly to NLP cases, our trained language models are showing repetition or hallucinations, especially when not trained to best convergence. This happens to both GPT and Mamba models. Below we show some error cases from a 16-layer Mamba model trained 150 epochs on the QM9 dataset. The error case below shows a typical repetition problem. The model generates repeated tokens for several periods, resulting in an invalid sample.

- H 0.00 0.00° 0.00° C 1.09 1.57° 0.00° N 2.02 2.15° 0.00° C 3.39 1.99° -0.02° H 3.98 2.10° 0.23° C 4.34 2.11° -0.35° H 4.43 2.38° -0.46° H 5.41 2.09° -0.29° H 4.29 1.96° -0.59° C 4.05 1.63° 0.04° H **4.09 4.09 4.09 4.09 4.09 4.09 4.09 4.09 4.09 4.09 4.09 4.09 4.09 4.09 4.09 4.09 4.09 4.09 4.09 4.09 4.09** 1.01° H **4.96 4.96 4.96** H **1.23° 1.23° 1.23°** C 3.27 1.74° -0.03° H 4.02 1.83° -0.23° H 3.79 1.91° 0.17° H 3.79 1.53° -0.03°

For hallucination, our tokenization design actually prevents token-level hallucination by defining elements and whole-numerical-values as tokens, instead of using single characters. This prevents token-level hallucination, *i.e.*, non-existent elements or numbers such as 'Hr' or '-0..15'. However, there can still be sequence-level hallucinations, such as the error case below. The model generates distance values in the place the should be angle values (and vice versa).

- H 0.00 0.00° 0.00° N 1.01 1.57° 0.00° H 1.70 2.14° 0.00° C 2.06 1.13° -0.48° O 3.13 1.34° -0.42° N 2.49 0.62° -0.87° C 2.94 0.10° -1.64° H 3.20 0.39° **3.91** H 2.81 0.33° -3.14° C 4.43 0.10° -1.77° H 5.15 0.22° 0.22° H 2.78 0.21° -0.95° C **1.86°** 1.86° **4.84** H **0.19°** 1.89° -2.06° H 8.28 1.94° -1.69° H 6.24 1.71° **5.70** C 3.97 0.67° -0.90° H 5.02 0.68° -0.77° H 4.15 0.91° -1.11° C 2.93 0.50° **6.97** H 3.54 0.42° 0.42° H 2.74 0.88° 0.88°

These error cases will be rarer if the model well converges. When trained for 150 epochs, the model would generate $\sim 15\%$ of invalid samples, including the above discussed syntax problems. When trained for 250 epochs, the model would generate $< 2\%$ of invalid samples.

## F VISUALIZATION RESULTS

### F.1 VISUALIZATION OF GENERATED MOLECULES

In this section, we provide visualizations of molecules generated from Geo2Seq with Mamba conditionally on the property of Polarizability $\alpha$ in Figure 3. The Polarizability of a molecule is the

tendency to acquire an electric dipole moment when the molecule is subject to an external electric field. Large $\alpha$ values usually correspond to less isometrically molecular geometries. This is consistent with our generated examples.

In addition, we provide visualizations of molecules generated from Geo2Seq with Mamba trained on QM9 and DRUG in Figure 4 and Figure 5, respectively. These examples are randomly generated without any cherry pick. From the figures, we can see that the model can generate realistic molecular geometries for both small and large size molecules. However, similar to previous methods (Hoogeboom et al., 2022; Xu et al., 2023a), there are disconnected components, especially for larger molecules. A possible future direction is to apply fragment-based methods to reduce the sequence length, thus benefiting the training of language models.

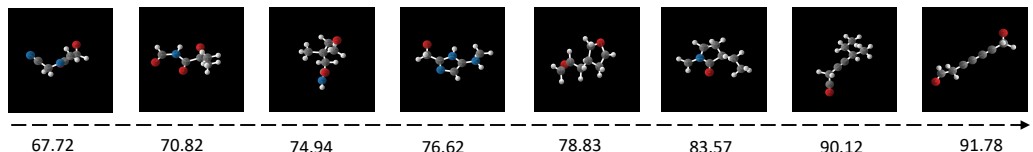

Figure 3: Visualization of generated molecules condition on the property of Polarizability $\alpha$.

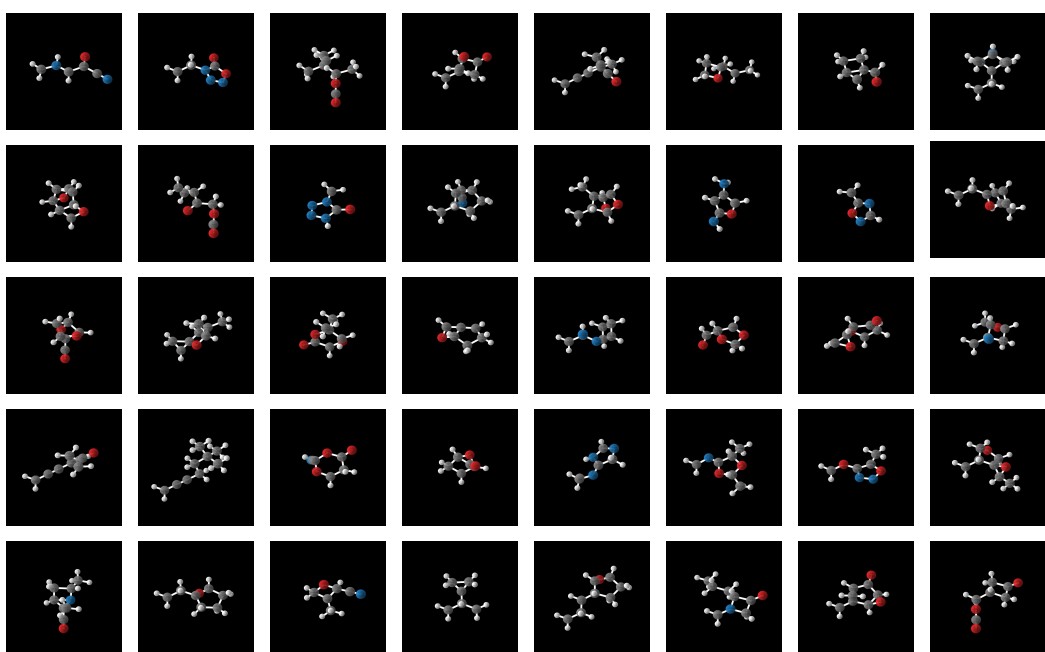

Figure 4: Visualization of molecules generated from Geo2Seq with Mamba trained on QM9.

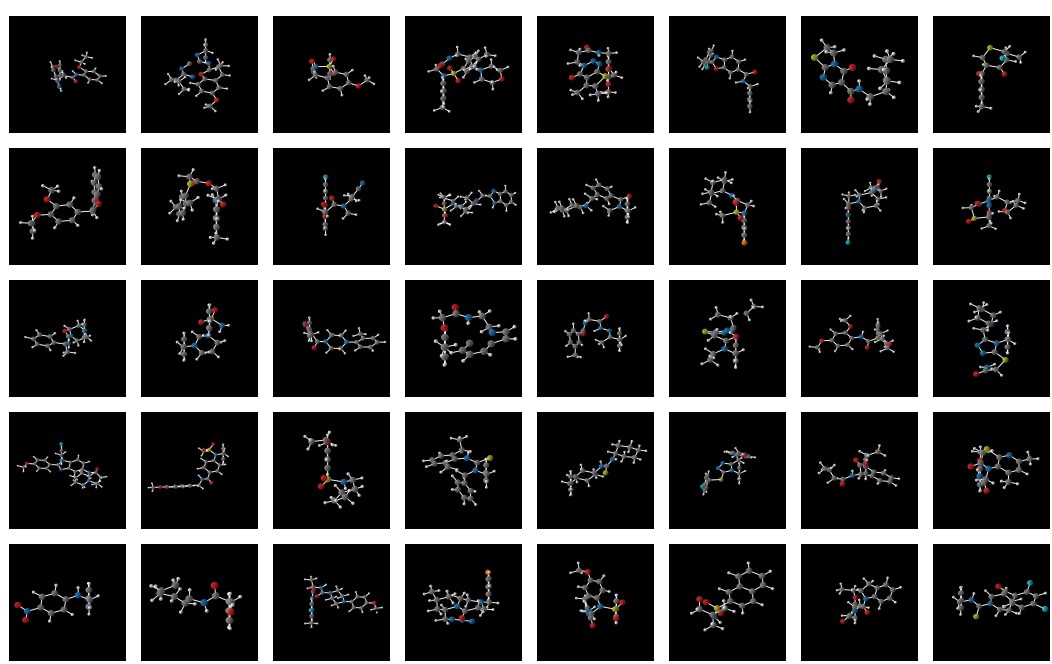

Figure 5: Visualization of molecules generated from Geo2Seq with Mamba trained on GEOM-DRUGS.

### F.2 VISUALIZATION OF LEARNED TOKEN EMBEDDINGS

In this section, we provide UMAP visualizations of different (atom type, distance, and angle) token embeddings learned by Mamba models trained on QM9 and GEOM-DRUGS datasets. Patterns of the embeddings indicate that the model has successfully learned structure information from the sequence data, showcasing LMs' capabilities to understanding molecules precisely in 3D space. For example, Figure 8 shows that similar angle tokens (*e.g.*, '1.41°' and '1.42°') are placed next to each other and the overall structure of all angles is a loop. Further, $\pi$-out-of-phase angles are placed near each other, such as '3.14°', '-3.14°', and '0°'. For atom type tokens, the model appears to capture the structure of the periodic table, although the rows and columns are not perfect in Figure 6. One reason is the limited atom types in the datasets (5 in QM9 and 16 in GEOM-DRUG), limiting the model's capabilities to learn chemical patterns from the entire periodic table. We provide analyses of the visualization results in the caption of each figure as Figure 6 - Figure 10.

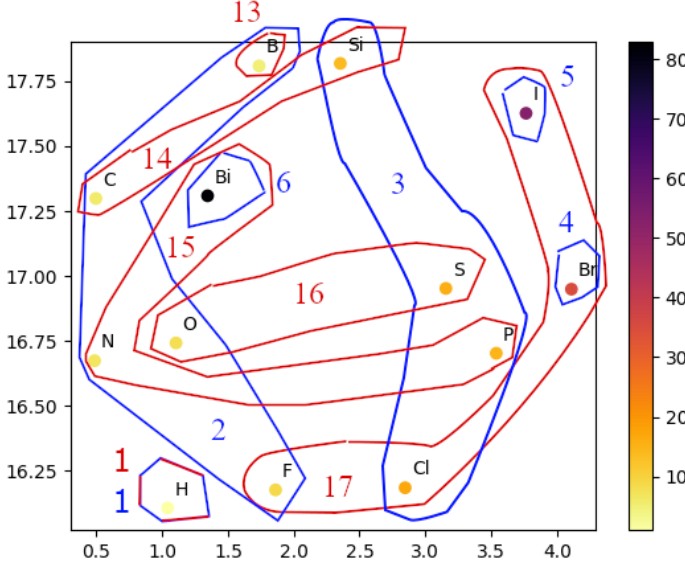

Figure 6: UMAP visualization of element token embeddings learned by a Mamba model trained on GEOM-DRUGS. Red groups indicate columns in the periodic table and blue groups indicate rows, which are both numbered. Points are colored by atomic weight. Overall, the model appears to capture the structure of the periodic table. The column generally increases from top to bottom, and the row generally increases from left to right.

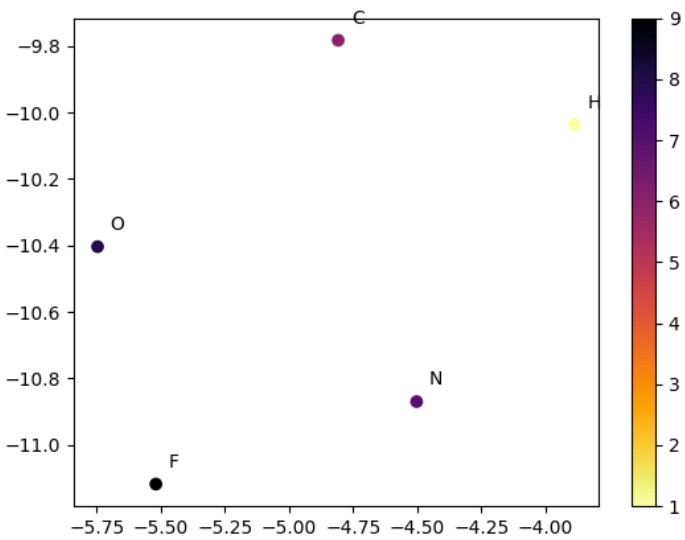

Figure 7: UMAP visualization of element token embeddings learned by a Mamba model trained on QM9. Points are colored by atomic weight. Overall, the model appears to distinguish well between different elements. All different elements are distributed distantly from each other in the embedding space.

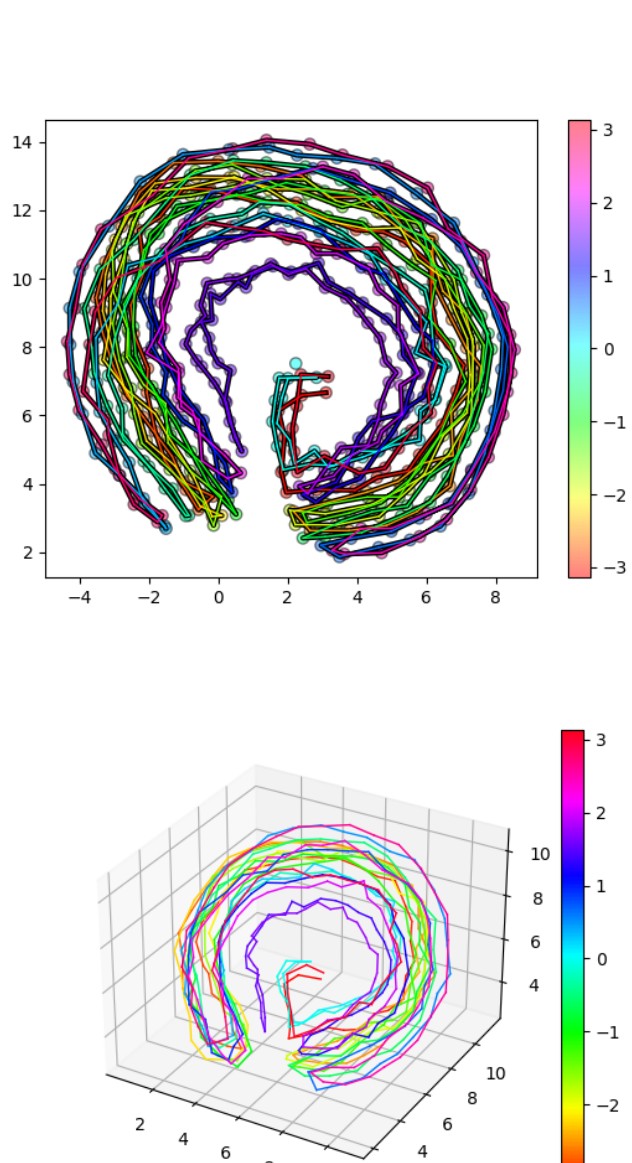

Figure 8: 2D and 3D UMAP visualization of angle token embeddings learned by a Mamba model trained on GEOM-DRUGS. It can be observed that similar tokens (e.g., '1.41°' and '1.42°') are placed next to each other and the overall structure is a loop. Further, $\pi$-out-of-phase angles are placed near each other, such as '3.14°', '-3.14°', and '0°'.

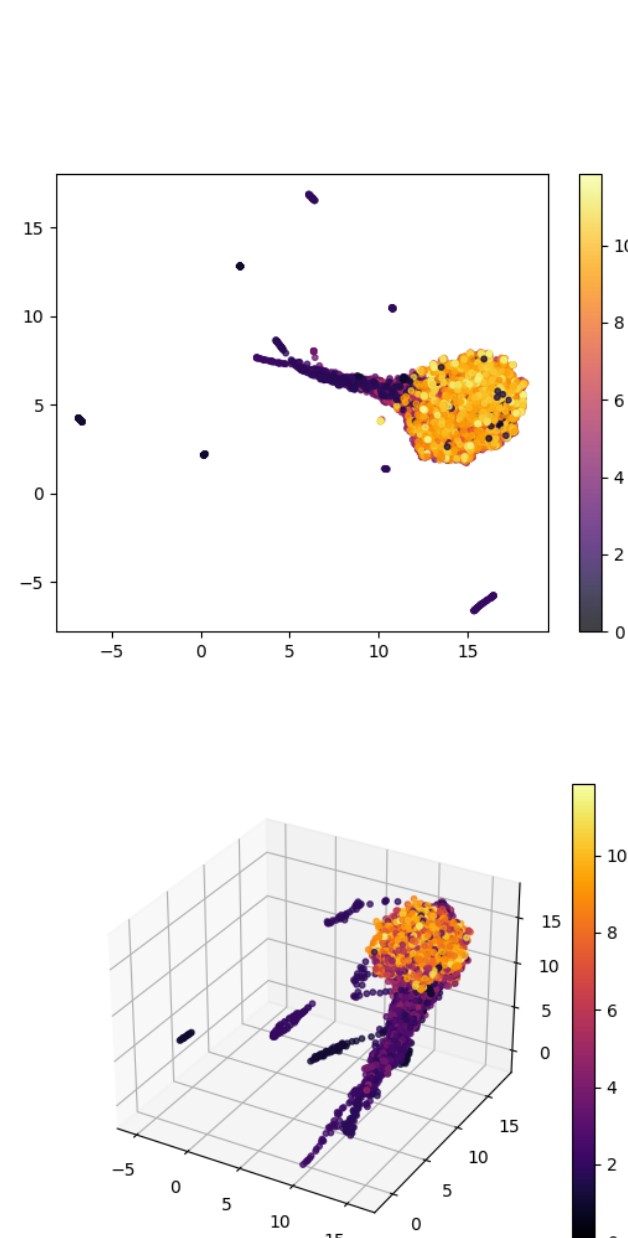

Figure 9: 2D and 3D UMAP visualization of distance token embeddings learned by a Mamba model trained on QM9. Representations of distances lower than 6 form relatively distinct patterns. This is likely because these values are much more frequently seen in the training data. Values over 20 cluster into a clump, suggesting that they are also recognized by the model.

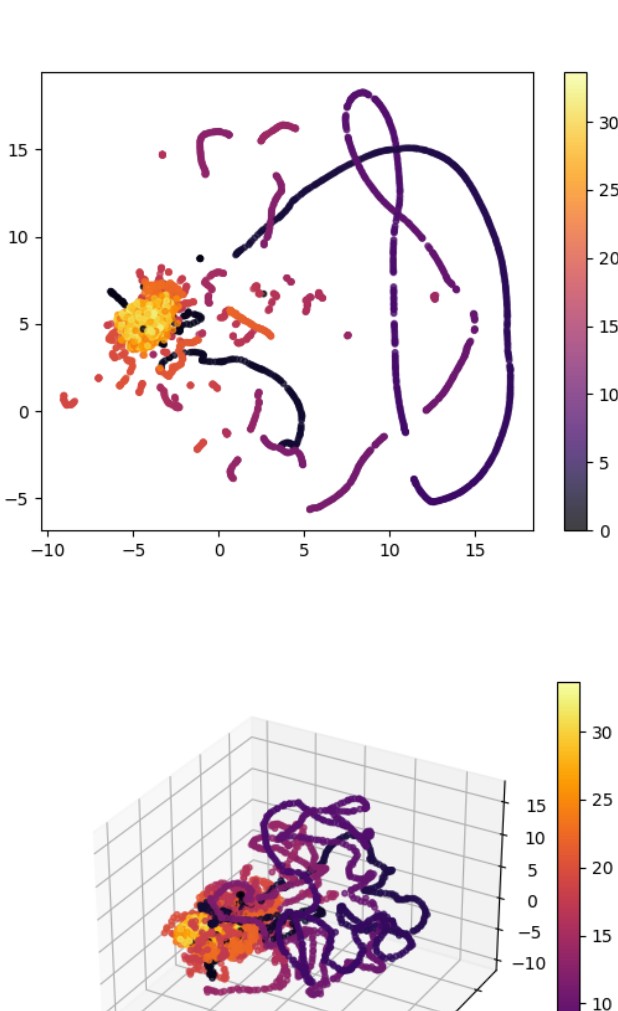

Figure 10: 2D and 3D UMAP visualization of distance token embeddings learned by a Mamba model trained on GEOM-DRUGS. It is notable that the best and most distinct representations seem to arise from between 5 and 20. This is likely because these values are much more frequently seen in the training data. Values over 20 form an indistinct clump. Interestingly, values > 20 are near values < 3, which is initially unintuitive; however, they are likely placed in a similar location in the embedding space since both small and large distances are rarely seen in the data.

