# OpenReview forum: "Geometry Informed Tokenization of Molecules for Language Model Generation"
_ICLR.cc/2025/Conference — Submitted to ICLR 2025_

### Official Review · Reviewer_aUY2 · 2024-10-27

**Soundness:** 3
**Presentation:** 4
**Contribution:** 3
**Rating:** 8
**Confidence:** 4

**Summary:**

This paper introduces Geo2Seq, a language model-based method for generating molecules. The method converts a 3D molecular graph into a 1D sequential representation through canonical ordering. It employs a spherical representation to encapsulate the 3D structural information, preserving SE(3)-invariance while ensuring no information loss. Additionally, the paper presents a specialized 3D tokenization approach tailored for generating 3D molecules.

**Strengths:**

1. Understanding how to use a sequential representation for modeling a 3D molecule is critical to language-based molecule generation.
2. The paper theoretically demonstrates that the application of canonical and spherical representations preserves SE(3)-invariance without any loss of information.
3. The paper is well-written and easy to understand.

**Weaknesses:**

1. My primary concern is that since the use of canonical ordering and spherical representation is not novel in the context of 3D molecular tasks, it would be beneficial to thoroughly discuss and compare their applications as documented in existing literature. This analysis should be included in the preliminary and related work section of the paper. This comparison will help in highlighting the unique contributions and potential advancements introduced by the Geo2Seq method.
2. Given that different tokenization approaches may significantly affect the generation quality, it would be beneficial to have additional ablation studies compare the proposed tokenization method with some traditional word-based or subword tokenization methods.

**Questions:**

Please refer to Weaknesses section.

---

> ### Author Response · Authors · 2024-11-20
> **Response to Reviewer aUY2**
>
> Dear Reviewer aUY2,
>
> Thank you for your constructive comments and valuable suggestions! We have made much effort to thoroughly improve our work accordingly and provide responses for each concern here. Please also refer to the revised paper PDF.
>
> > W1: ... it would be beneficial to thoroughly discuss and compare their applications as documented in existing literature. This analysis should be included in the preliminary and related work section of the paper ...
>
> - Thanks for your constructive suggestions. We understand your point that canonical ordering and spherical representation have been used in existing literature. We would like to first emphasize that our major contribution lies in providing a solution for LM learning of 3D molecules which can achieve both structural completeness and geometric invariance. Theoretically we demonstrate that using canonicalization and spherical representation can achieve our goal.
> - Following your advice, we added a more thorough literature review as follows, which is also included in the **paper revision**:
>   - Spherical representation has been applied in various molecule-related tasks [1], including molecular property prediction [2, 3] and molecule generation [4]. A crucial step in using spherical representation is defining the coordinate frame, i.e. x, y, and z axes. One straightforward approach is to directly use the frame as for the input coordinates, i.e. (1, 0, 0) as x-axis, but this method fails to ensure SE(3) invariance. Instead, SphereNet [2] defines local frames based on a central edge and one reference node to ensure invariance [3]. Similarly, G-SphereNet defines local frames based on focal atoms to compute distance and angle for model generation [4]. These approaches demonstrate the importance of carefully designing the coordinate frame when working with spherical representations in molecule-related tasks.
>   - Canonical labeling (CL) has been adopted from the graph theory and used in molecular representation, enabling the conversion of molecules into 1D sequences. This allows for efficient processing and analysis of chemical structures. One of the most popular canonical sequence is canonical SMILES [5], which represents molecules as a string of characters based on the Morgan Algorithm [7] and additional defined rules. SELFIES [6] provides a more robust sting representation to overcome the limitation that some strings do not correspond to valid molecules. However, the application of CL to 3D molecules has yet been studied.
>
> > W2: ... it would be beneficial to have additional ablation studies compare the proposed tokenization method with some traditional word-based or subword tokenization methods.
>
> - Thanks for your insightful comment. We agree that different tokenization approaches may significantly affect the generation quality, and we have included ablation study on tokenization approaches in our initial version (see Appendix C, Table 5), including Comp-tokenization (taking the complete real number as a token) and Sub-tokenization (splitting a real number by the decimal point and treat every part as an individual token). We have also tried Char-tokenization (treating each character as a token) and BPE during our method design, but they didn't work well. A detailed comparison is show in the following table, which is also included in the **paper revision**:
>
> | Tokenization| Atom Sta (%) | Mol Sta (%) | Valid (%) | Valid & Unique (%) |
> |--|--|--|--|--|
> | Char-tokenization |90.5|43.7|71.5| 71.0|
> | BPE |85.3|55.3|74.4| 57.6|
> | Sub-tokenization |96.4|80.3|89.9| 74.4|
> | Comp-tokenization|**97.4**|**86.8**|**93.0**|**84.7**|
>
> We sincerely thank you for your time! Hope we have addressed your concerns through practical efforts and shown the contributions and significance of our work. We look forward to your reply and further discussions, thanks!
>
> Sincerely,
>
> Authors
>
> [1] Fast and accurate protein structure search with Foldseek, 2024
>
> [2] Spherical message passing for 3D graph networks, 2022
>
> [3] PiFold: Toward effective and efficient protein inverse folding, 2022
>
> [4] An autoregressive flow model for 3D molecular geometry generation from scratch, 2022
>
> [5] SMILES, a chemical language and information system, 1988
>
> [6] Self-referencing embedded strings (SELFIES): A 100% robust molecular string representation, 2020
>
> [7] The generation of a unique machine description for chemical structures-a technique developed at chemical abstracts service, 1965

---

> > ### Comment · Reviewer_aUY2 · 2024-11-26
> > **Response to Rebuttal**
> >
> > I sincerely appreciate reviewers detailed rebuttal. My questions have been resolved. I will increase my score.

---

> > > ### Author Response · Authors · 2024-11-26
> > > **Thank you to Reviewer aUY2**
> > >
> > > Dear Reviewer  aUY2,
> > >
> > > Thank you so much for replying to our rebuttal and increasing our score! We are grateful for your recognition that we have resolved your questions. We sincerely thank you for your time and welcome any additional discussions or feedback.
> > >
> > > Sincerely,
> > >
> > > Authors

---

### Official Review · Reviewer_dPhN · 2024-10-28

**Soundness:** 3
**Presentation:** 3
**Contribution:** 2
**Rating:** 5
**Confidence:** 5

**Summary:**

This paper proposes Geo2Seq, which extracts invariant coordinates of molecules and supports language model-based 3D molecular structure modeling through digit-level encoding. The method demonstrates strong performance across datasets.

**Strengths:**

The benchmarks show strong performance, especially in controllable generation.

**Weaknesses:**

1. This is a technically solid paper but with relatively limited contributions.
2. This paper does not require extensive theoretical exposition. Both isomorphism and invariance are well-studied concepts with no new conclusions presented here, and the heavy use of theoretical formulations affects readability.
3. From the existing experimental analysis, it remains unclear whether the model is merely replicating token patterns or genuinely learning underlying molecular principles.

**Questions:**

1. Although it may compromise invariance, would training directly on centered Cartesian coordinates improve performance on this task?
2. How does Geo2Seq ensure that generated molecules adhere to chemical and physical constraints (e.g., bond lengths and steric hindrance)? Are there methods to introduce explicit constraints?
3. Would increasing coordinate precision (i.e., more decimal places) impact the quality or accuracy of the model's generated outputs?
4. How does the GPT model perform in controllable generation? The model structure does not appear to be significantly different.

---

> ### Author Response · Authors · 2024-11-20
> **Response to Reviewer dPhN (Part I)**
>
> Dear Reviewer dPhN,
>
> Thank you for your constructive comments and valuable suggestions!
> We have made much effort to thoroughly improve our work accordingly and provide responses for each concern here. Please also refer to the revised paper PDF.
>
> > W1. Limited contributions
>
> We first extend our gratitude for sharing your specific concerns on contributions. We understand that contribution concerns are very subjective. However, the contribution consideration should be also comprehensive. In order to demonstrate the contributions of our paper clearly, we mainly focus on the distinctions between this paper and the previous works.
> 1. **The track is under-explored:** Different from mainstream methods, we convert 3D molecules to 1D discrete sequences and generate using LMs for 3D molecule generation tasks, positioning in an under-explored field.
> 2. **Technique distinctions:** The only comparable work using LMs applies coordinates from XYZ files directly. In contrast, Geo2Seq achieves **structural completeness** and **geometric invariance** for **LM training**, which is distinct from all existing works.
> 3. **Flexibility & comprehensive architecture:** Geo2Seq operates solely on the input data, which allows independence from model architecture and training techniques and provides reuse flexibility. To further demonstrate the universal applicability of our method, we not only include conventional Transformer architecture of CLMs, but also employ SSMs as LM backbones.
> 4. **Effectiveness & Efficiency:**  Results under both architectures prove the effectiveness of Geo2Seq. In addition, compared to diffusion models, our method outperforms them with significantly higher efficiency, *i.e.*, 100x faster generation speed.
>
> Although this work stands on the shoulders of techniques from multiple fields, we have demonstrated our unique contributions, which is beneficial to the 3D molecular community.
>
> > W2. Does not require extensive theoretical exposition...no new conclusions presented here
>
> Thank you for your comment.
> - We understand the importance of balancing theoretical depth with readability. To address this, we have restructured the paper to move a few theoretical details to the appendix, while highlighting necessary contents and the methodology. Please refer to the revised paper.
> - Still, we would like to clarify our theoretical contributions the importance of the theoretical formulations presented:
>   - While isomorphism and invariance are concepts in graph theory, our paper extends these concepts to another domain by introducing 3D graph isomorphism and its application to molecular systems. Specifically, we propose an extension that incorporates SE(3)-invariance into graph isomorphism, ensuring equivalence of molecular modeling. **To the best of our knowledge, this has not been explicitly studied in prior works**. The proposed 3D graph isomorphism is a more general and rigorous definition of 3D molecular equivalence and provides a framework for comparing molecules at multiple levels of abstraction.
>   - The theoretical formulations also serve as the foundation for our methodology, ensuring the correctness and invariance of the sequence representation. We demonstrated how our theoretical framework directly supports the development of Geo2Seq, which provides a complete SE(3)-invariant mapping of 3D molecular structures into sequences, also **a new conclusion we present**. The guarantee for this mapping is established via our theoretical derivations.
>   - Our work introduces a theoretical contribution that bridges graph theory, geometry, and language models for 3D molecular modeling. The conclusions we present benefit the development of these communities.

---

> > ### Author Response · Authors · 2024-11-20
> > **Response to Reviewer dPhN (Part II)**
> >
> > > W3. Unclear whether the model replicating token patterns or learning underlying molecular principles
> >
> > Thanks for your comment. Here we provided detailed responses to this point from the perspectives of our method design and generated molecule analysis.
> >
> > - In order to capture physical and chemical information, in the method design, we consider spherical coordinates and SE(3)-invariance, which are well-known domain-specific molecular modeling techniques (Q7) (some related molecular modeling papers: SchNet, DimeNet, SphereNet, Equiformer). These techniques ensure structural completeness and geometric invariance for LM training.
> > - Experimentally, Table 6 reports further random generation results on QM9, including novelty metric, while Table 7 presents the novelty results of controllable generation. Results show that our method achieves reasonably high **novelty** scores, which demonstrates that our method is **not simply memorizing training data**.
> > - In addition, in Appendix F.2, we provided UMAP visualizations of learned (atom type, distance, and angle) token embeddings, which indicates that the model has successfully learned structure information from the sequence data, showcasing LMs' capabilities to understand molecules precisely in 3D space. **For example**, Figure 8 shows that similar angle tokens (e.g., 1.41° and 1.42°) are placed next to each other and the overall structure of all angles is a loop. Further, π-out-of-phase angles are placed near each other, such as 3.14°, -3.14°, and 0°. For atom type tokens, the model appears to capture
> > the structure of the periodic table.
> > - We believe the current results are sufficient to prove the correctness and capabilities of our design. Geo2Seq with LMs can model 3D molecular structure distribution and capture the underlying chemical rules. In addition, our method moves a significant step in pushing the performance of controllable generation task towards the lower bound.
> >
> > > Q1. Centered Cartesian coordinates
> >
> > - We would like to study whether normalizing the xyz coordinates is effective. Following your advice, we subtract the xyz coordinates with the mass-center coordinates of each molecule. We provide experiment results using 8-layer GPT models on QM9 dataset:
> >
> > |3D representation|Atom Sta(%)|Mol Sta(%)|Valid(%)|Valid&Unique(%)|
> > |---|---|---|---|---|
> > |Original coordinates|91.1|58.1|75.6|55.1|
> > |Normalized coordinates|92.7|63.2|83.1|72.5|
> > |Invariant Cartesian coordinates|96.0|78.5|89.7|74.1|
> > |Invariant spherical coordinates|**97.3**|**83.4**|**91.0**|**82.7**|
> >
> > - As shown above, normalizing the xyz coordinates brings certain performance gain over the original coordinates, but does not completely match the performance of invariant Cartesian coordinates. Compared to normalization, invariance shows more effectiveness as molecular inductive bias, and the xyz coordinates are not as advantageous as spherical angles for LM learning.
> >
> > > Q2. Ensure chemical and physical constraints, introduce explicit constraints
> >
> > - Firstly, we would like to clarify that existing AI methods on molecule generation can't ensure that all generated molecules adhere to all physical and chemical constraints, which is also why **validity** is one of the most important evaluation metrics for molecule generation quality. Experiments indicate that our generated molecules generally satisfy the constraints well. We can further improve the validity by incorporating chemical constraints as validity checks (e.g. stop generation if some constraints are not satisfied) during the generation process, but it is an add-on outside the model learning process with additional cost and we leave it as future work.
> > - From Table 1, our method achieved SOTA results on most metrics, including validity, atom stability (the percentage of atoms with correct bond valencies), and molecule stability (percentage of molecules whose all atoms have correct bond valencies), indicating our method can learn physical and chemical constraints better compared to existing methods. In addition, we provided more evaluation results on **various chemical metrics in Appx. D, Tables 6, 7, 8, 9, 10, and 11, including bond lengths and steric hindrance**. **Vast results** show that we can outperform existing methods across metrics regarding various chemical constraints.

---

> > > ### Author Response · Authors · 2024-11-20
> > > **Response to Reviewer dPhN (Part III)**
> > >
> > > > Q3. Increasing coordinate precision
> > >
> > > - Yes. As we specified in the paper, with $b$ decimal places, we have a information loss of $ϵ\leq|10-b|/2$. More decimal places produce less information loss and slightly better performance, but has larger vocabulary size and thus larger computational cost, forming a trade-off.
> > > - We also conduct experiments to empirically show comparisons under precisions of 1, 2 and 3 decimal places. We provide experiment results using 8-layer GPT models on QM9 dataset:
> > >
> > > | Precisions| Atom Sta (%) | Mol Sta (%) | Valid (%) | Valid & Unique (%) |
> > > |--|--|--|--|--|
> > > | 1-dp|95.9|81.6|90.4|75.1|
> > > | 2-dp|97.4|86.8|93.0|84.7|
> > > | 3-dp|97.9|86.9|93.5|84.7|
> > >
> > > - As shown above, using more decimal places enable LMs to capture 3D molecular structures more comprehensively and produce better results.
> > >
> > > > Q4. GPT model in controllable generation
> > >
> > > - Thank you for this suggestion. We would like to extend the performance of GPT in Table 2. Initially, we only explore Mamba in conditional generation due to its better time efficicy. The results are as below(just including important baselines):
> > >
> > > | Property (Units)       | α (Bohr³) | Δε (meV) | ε_HOMO (meV) | ε_LUMO (meV) | μ (D)  | C_v (cal/mol·K) |
> > > |--|-|-|--|--|--|--|
> > > |Data|0.10| 64| 39 | 36| 0.043| 0.040|
> > > |Random |9.01| 1470| 645| 1457|1.616| 6.857|
> > > |GEOLDM|2.37| 587| 340| 522 | 1.108  | 1.025           |
> > > |Geo2Seq with Mamba| **0.46**| **98**| 57 | 71| 0.164 | **0.275**|
> > > |Geo2Seq with GPT| 0.53|102|**48**|**53**|**0.097**|0.325|
> > >
> > > - As shown above, Geo2Seq with GPT achieves significantly better results over the baselines and outperforms Geo2Seq with Mamba in several properties. We have included the results and discussions in the paper revision.
> > >
> > >
> > > We sincerely thank you for your time! Hope we have addressed your concerns through practical efforts and shown the contributions and significance of our work. We look forward to your reply and further discussions, thanks!
> > >
> > > Sincerely,
> > >
> > > Authors

---

> > > > ### Author Response · Authors · 2024-12-03
> > > > **Follow-up with Reviewer dPhN**
> > > >
> > > > Dear Reviewer dPhN,
> > > >
> > > > Thank you again for your valuable additional comments, which helps to improve our work a lot. Since the discussion period is quickly approaching the end, we sincerely hope you can let us know if we have fully addressed your concerns and reevaluate our work if we have. Also, please let us know if there are any additional questions or feedback. Thank you!
> > > >
> > > > Sincerely,
> > > >
> > > > Authors

---

### Official Review · Reviewer_Sjkt · 2024-11-01

**Soundness:** 2
**Presentation:** 2
**Contribution:** 2
**Rating:** 5
**Confidence:** 4

**Summary:**

The paper proposes a novel tokenization method, Geo2Seq, for generating 3D molecular structures using language models (LMs). Geo2Seq transforms molecular geometries into SE(3)-invariant 1D discrete sequences, making them compatible with LMs while preserving essential structural details. The method uses canonical labeling to establish a unique node order and spherical coordinates for SE(3)-invariance, enabling robust encoding of molecular geometry. This technique allows LMs to handle 3D molecule generation more effectively, surpassing existing models in generating diverse, chemically valid 3D structures.

**Strengths:**

1. This is a very interesting task, and it’s great to see a method for integrating multimodal data, especially 3D molecular structures.
2. The motivation is clear and straightforward.
3. The experiments in this paper are comprehensive, covering various LMs and ablations.
4. The writing is clear and accessible.

**Weaknesses:**

I believe there is still room for improvement in the paper:

1. The canonical labeling method discussed in Section 3.1 resembles the SMILES canonicalization commonly used in cheminformatics. Intuitively, this functionality could be implemented using SMILES canonicalization in RDKit—why not utilize that?
2. The molecular bonds are generated using RDKit, also there is no strict definition of bonds in quantum chemistry. What is the necessity of introducing graph isomorphism in this paper?
3. The authors suggest that limiting the range could enhance the language model’s performance. However, in spherical coordinates, only the azimuth and polar angles are restricted in range, while the distance remains a continuous variable. This could still limit applications for larger molecules, such as proteins.

**Questions:**

1. Would using SMILES canonicalization be a better approach? How does it differ from the method you've employed?
2. Since the LLM is already generating iteratively, I believe adopting a local coordinate system could be more effective, as it would help keep distances manageable.
3. I'm curious whether normalizing the xyz coordinates (for example, by subtracting the center of mass coordinates) could achieve a similar effect. According to the authors, normalization could also assist in reducing the range to some extent.

---

> ### Author Response · Authors · 2024-11-20
> **Response to Reviewer Sjkt (Part I)**
>
> Dear Reviewer Sjkt,
>
> Thank you for your constructive comments and valuable suggestions!
> We have made much effort to thoroughly improve our work accordingly and provide responses for each concern here. Please also refer to the revised paper PDF.
>
> > W1. Why not use SMILES canonicalization
>
> - Thank you for the point. As we specified in Sec 3.1 and Appx. C, our theoretical analyses and derivations apply to all rigorous CL algorithms. We do not propose a new CL algorithm, and all rigorous CL algorithms are usable here, while our contribution lies in achieving structural completeness and geometric invariance for LM learning of 3D molecules. Thus we believe that the choice of CL algorithm is not a weakness of our work. In the paper, we implement Nauty Algorithm for 3D molecules because:
>   1. Its implementation has the best time efficiency among all existing CL algorithms.
>   2. Canonical SMILES is based on the Morgan CL Algorithm, which is proven to be incomplete for isomorphism corner cases (such as two triangles versus one hexagon). While canonical SMILES solve corner cases by manual restrictions, it's a bit less elegant than Nauty Algorithm.
> - We have studied the influence of ordering algorithms in `Ablation on atom order` of Appendix C. Following your advice, we further extend to include canonical SMILES order, as shown below.
>
> |Order|Atom Sta(%)|Mol Sta(%)|Valid(%)|Valid&Unique(%)|
> |---|---|---|---|---|
> |Canonical-locality|**97.39**|**86.77**|**92.97**|**84.71**|
> |Canonical-SMILES|97.35|85.86|92.97|84.05|
> |Canonical-nonlocality|96.45|81.36|90.89|83.37|
> |DFS|95.95|81.54|90.45|82.48|
> |BFS|96.85|80.92|90.49|76.13|
> |Dijkstra|95.29|77.25|88.97|73.52|
> |Cuthill–McKee|93.56|71.57|85.36|76.23|
> |Hilbert-curve|90.11|64.99|80.40|67.83|
> |Random|64.87|20.14|43.16|38.44|
>
> - Our implemented Nauty Algorithm for 3D molecules can perform partitioning of graph vertices (a step in Nauty) with/without locality considered. Canonicalization with locality considered can lead to better results, due to the importance of neighboring atom interactions in molecular evaluations. Given the similar nature, canonical SMILES produces a very similar ordering with "Nauty with locality", thus close in performances.
>
> > W2. No strict definition of bonds in quantum chemistry...graph isomorphism necessity?
>
> - Thank you for your thoughtful question. While molecular bonds in our paper are generated using RDKit, the introduction of graph isomorphism serves a foundational role for several reasons:
>   - In graph theory, graph isomorphism is not restricted to comparing the presence or absence of edges; the concept allows us to model and compare node attributes, and thus features such as molecular geometry or spatial configurations. Graph isomorphism provides a general framework for comparing molecules at multiple levels of abstraction, beyond bond-centric representation.
>   - Also, the introduction of graph isomorphism lays necessary theoretical groundwork for our proposed 3D graph isomorphism, which extends to incorporate spatial geometry through SE(3)-invariant transformations. The defined 3D graph isomorphism allows us to compare 3D molecular graphs, ensuring that we account for both topological and geometric equivalence of molecules, and can generalize to different bonding schemes.
>   - The introduction of 3D graph isomorphism is a more general and rigorous definition of 3D molecular equivalence. We believe this is meaningful theoretical contribution providing a unified perspective. Our method, Geo2Seq, also uses this theoretical foundation to guarantee the correctness and invariance of the sequence representation. The process uses the 3D graph isomorphism to ensure that equivalent 3D molecular graphs are mapped to the same representation, thus achieving SE(3)-invariant sequences.
>
> > W3 & Q2. Limiting the range...distance remains
>
> Thank you for the insightful question and advice! We consider W3 and Q2 together here.
> - It is true that in our current setting, the distance is not restricted numerically like the azimuth and polar angles. We appreciate your suggestion regarding adopting a local coordinate system to manage distances. To further analyze, we provide ablation studies on `local distances`, where our `distances to the global frame` are replaced with `relative distances to the previous atom` (except for the first atom) while the angles remain the same. This reduces the scale of the distances.
> - We provide experiment results using 8-layer GPT models on QM9 dataset:

---

> > ### Author Response · Authors · 2024-11-20
> > **Response to Reviewer Sjkt (Part II)**
> >
> > |3D representation|Atom Sta(%)|Mol Sta(%)|Valid(%)|Valid&Unique(%)|
> > |---|---|---|---|---|
> > |Original coordinates|91.1|58.1|75.6|55.1|
> > |Inv-Cartesian coordinates|96.0|78.5|89.7|74.1|
> > |Inv-spherical coordinates|**97.3**|**83.4**|91.0|**82.7**|
> > |Inv-spherical coordinates-local distances|97.1|82.8|**91.7**|79.6|
> >
> > - As shown above, both our spherical coordinates and that replaced with local distances achieve comparable results, while outperforming Cartesian coordinates. From these empirical results, we can analyze that the representation of azimuth and polar angles has brought sufficient advantage for LM learning over Cartesian coordinates, thus spherical representations with both distance schemes are showing promising performances. In addition, the similar performances could be attributed to that molecular systems often exhibit localized spatial structures (e.g., compact subunits or functional groups), which naturally constrain distances for most small molecules. We have included these studies in the paper revision.
> > - For larger compounds, such as proteins, it is indeed possible that global distances could become less manageable. In this work, we focus on the scope of small molecules, while we agree that using local distances for protein tasks could be interesting future work. Moreover, in scenarios where excessive distances do occur (e.g., highly extended proteins), additional techniques such as fragmentation, hierarchical or multi-scale modeling can be applied to further reduce distance scales.
> >
> > > Q1. Comparison to SMILES canonicalization
> >
> > - Please refer to the response to W1.
> >
> > > Q2. Local coordinate system
> >
> > - Please refer to the response to W3.
> >
> > > Q3. Normalizing the xyz coordinates
> >
> > Thank you for the question.
> > - We would like to study whether normalizing the xyz coordinates is effective. Following your advice, we subtract the xyz coordinates with the mass-center coordinates of each molecule. We provide experiment results using 8-layer GPT models on QM9 dataset:
> >
> > |3D representation|Atom Sta(%)|Mol Sta(%)|Valid(%)|Valid&Unique(%)|
> > |---|---|---|---|---|
> > |Original coordinates|91.1|58.1|75.6|55.1|
> > |Normalized coordinates|92.7|63.2|83.1|72.5|
> > |Invariant Cartesian coordinates|96.0|78.5|89.7|74.1|
> > |Invariant spherical coordinates|**97.3**|**83.4**|**91.0**|**82.7**|
> >
> > - As shown above, normalizing the xyz coordinates brings certain performance gain over the original coordinates, but does not completely match the performance of invariant Cartesian coordinates. Compared to normalization, invariance shows more effectiveness as molecular inductive bias, and the xyz coordinates are not as advantageous as spherical angles for LM learning.
> >
> > > Novelty and Significance
> >
> > Finally, we would like to demonstrate the contributions and distinctions of our paper clearly:
> > 1. **The track is under-explored:** Different from mainstream methods, we convert 3D molecules to 1D discrete sequences and generate using LMs for 3D molecule generation tasks, positioning in an under-explored field.
> > 2. **Technique distinctions:** The only comparable work using LMs applies coordinates from XYZ files directly. In contrast, Geo2Seq achieves **structural completeness** and **geometric invariance** for **LM training**, which is distinct from all existing works.
> > 3. **Flexibility & comprehensive architecture:** Geo2Seq operates solely on the input data, which allows independence from model architecture and training techniques and provides reuse flexibility. To further demonstrate the universal applicability of our method, we not only include conventional Transformer architecture of CLMs, but also employ SSMs as LM backbones.
> > 4. **Effectiveness & Efficiency:**  Results under both architectures prove the effectiveness of Geo2Seq. In addition, compared to diffusion models, our method outperforms them with significantly higher efficiency, *i.e.*, 100x faster generation speed.
> >
> > We believe our work would be beneficial to the 3D molecular community and would genuinely appreciate your recognition.
> >
> > We sincerely thank you for your time! Hope we have addressed your concerns through practical efforts and shown the contributions and significance of our work. We look forward to your reply and further discussions, thanks!
> >
> > Sincerely,
> >
> > Authors

---

> > > ### Comment · Reviewer_Sjkt · 2024-11-25
> > >
> > > Thank you for your response. Most of my concerns have been addressed; however, one issue remains regarding the **local distances**. If you replace the global frame distances with relative distances to the previous atom while keeping the angle component in the global frame, I believe the reconstruction task cannot be completed in theory. Could you please share your code with an anonymous repo?

---

> ### Author Response · Authors · 2024-11-26
> **Further response to Reviewer Sjkt**
>
> Dear Reviewer Sjkt,
>
> Thank you for replying to our rebuttal! We are glad to know that most of your concerns have been addressed, and would like to respond to your remaining concern as follows.
>
> >  Local distances ... relative distances to the previous atom while keeping the angle component in the global frame ... reconstruction task cannot be completed
>
> - We would like to clarify that in our ablation studies on local distances, when we replace `distances to the global frame` with `relative distances to the previous atom`, the angles are **not constructed in the global frame** but **constructed along with the same relativity**. You are definitely correct in that reconstruction cannot be completed if relative distances are used while keeping the angle component in the global frame. The **distances and two angles** need to be constructed under the same frame, thus we adopt relative distances and use the same way as Geo2Seq to define azimuth and polar angles under relativity. Specifically:
>   - For each atom $A_i$, we compute local distance $d = \| \mathbf{r}\_{i} - \mathbf{r}\_{i-1} \|$ where $\mathbf{r}\_{I}$ and $\mathbf{r}\_{i-1}$ are the positions of $A\_i$ and $A\_{i-1}$. We compute angles, where polar angle $\theta$ is the angle between $\mathbf{r}\_i - \mathbf{r}\_{i-1}$ and the local $z$-axis, and Azimuth angle $\phi$ is the angle between the projection of $\mathbf{r}\_i - \mathbf{r}\_{i-1}$ onto the local $x-y$ plane and the local $x$-axis. For the locality, we define a reference frame with local axes using two adjacent atoms, $z$-axis as $\mathbf{v} = \mathbf{r}\_{i-1} - \mathbf{r}\_{i-2}$, $x$-axis as vector orthogonal to $\mathbf{v}$, and $y$-axis as $\mathbf{y} = \mathbf{z} \times \mathbf{x}$. For $A\_1$, the global frame is still used as the reference.
>   - For reconstruction we can do the following steps. First we perform global frame initialization similar to Geo2Seq; the global coordinate of the first atom is $\mathbf{r}\_1 = [0, 0, 0]$, and the coordinate of the second atom is set along direction $\mathbf{r}\_2 = [d\_2, \pi/2, 0]$. The third atom defines the initial reference frame. We place it in 3D space based on the distance $d\_3$ and angles $\theta\_3, \phi\_3$, $$\mathbf{r}\_3 = \mathbf{r}\_2 + d\_3 \cdot \begin{bmatrix} \sin(\theta\_3) \cos(\phi\_3) \\\\ \sin(\theta\_3) \sin(\phi\_3) \\\\ \cos(\theta\_3) \end{bmatrix}.$$ For $i \geq 4$, compute the local frame at $A\_{i-1}$, where local $z$-axis is $\mathbf{z}\_{i-1} = \frac{\mathbf{r}\_{i-1} - \mathbf{r}\_{i-2}}{\|\mathbf{r}\_{i-1} - \mathbf{r}\_{i-2}\|}$, local $x$-axis is $\mathbf{x}\_{i-1} = \frac{\mathbf{x}\_g - (\mathbf{z}\_{i-1} \cdot \mathbf{x}\_g) \cdot \mathbf{z}\_{i-1}}{\|\mathbf{x}\_g - (\mathbf{z}\_{i-1} \cdot \mathbf{x}\_g) \cdot \mathbf{z}\_{i-1}\|}$, and local $y$-axis: $\mathbf{y}\_{i-1} = \mathbf{z}\_{i-1} \times \mathbf{x}\_{i-1}$. We can reconstruct global cartesian coordinate for $A\_i$. Using $d\_i, \theta\_i, \phi\_i$, calculate the Cartesian displacement of $A\_i$ in the local reference frame $$\Delta \mathbf{r}\_i = d\_i \cdot \begin{bmatrix} \sin(\theta\_i) \cos(\phi\_i) \\\\ \sin(\theta\_i) \sin(\phi\_i) \\\\ \cos(\theta\_i) \end{bmatrix}.$$ Transform this displacement into the global frame using the local axes, $$\Delta \mathbf{r}\_i^\text{global} = \Delta \mathbf{r}\_i \cdot \begin{bmatrix} \mathbf{x}\_{i-1} \\\\ \mathbf{y}\_{i-1} \\\\ \mathbf{z}\_{i-1} \end{bmatrix}^T.$$ And compute the global coordinate $\mathbf{r}\_i = \mathbf{r}\_{i-1} + \Delta \mathbf{r}\_i^\text{global}$. Wo do this for all subsequent atoms $A\_{i}$ to reconstruct the global coordinates for the molecule.
> - At your request, we would also like to share our code through this anonymous link: https://anonymous.4open.science/r/DGTHM/
>
> We sincerely thank you for your time! Hope we have addressed your concerns through practical efforts and shown the contributions and significance of our work. We believe our work would be beneficial to the 3D molecular community and would genuinely appreciate your recognition. We hope you can let us know if we have addressed your concern and reconsider the score if we have. We also welcome any additional discussions or feedback, thanks!
>
> Sincerely,
>
> Authors

---

> > ### Author Response · Authors · 2024-12-02
> > **Follow-up with Reviewer Sjkt**
> >
> > Dear Reviewer Sjkt,
> >
> > Thank you again for your valuable additional comments, which helps to improve our work a lot. Since the discussion period is quickly approaching the end, we sincerely hope you can let us know if we have fully addressed your concerns and reevaluate our work if we have. Also, please let us know if there are any additional questions or feedback. Thank you!
> >
> > Sincerely,
> >
> > Authors

---

### Official Review · Reviewer_Y6S8 · 2024-11-01

**Soundness:** 3
**Presentation:** 4
**Contribution:** 3
**Rating:** 6
**Confidence:** 4

**Summary:**

The paper proposes a way to tokenize 3D molecules into sequences for language model generation which preserve structural completeness and SE(3)-invariance. They include detailed derivations and proofs for the claims. The method demonstrates promising results for random generation and controllable generation. They also include comprehensive ablations showing the importance of the canonicalization and 3D representation.

**Strengths:**

- The method and derivations are sound and rigorous
- The ablations are comprehensive and justify the method's design choices
- The random generation results are promising
- The presentation is beautiful

**Weaknesses:**

- There should be more LM-based baselines for random generation
- There should be more baselines for controllable generation

**Questions:**

For this work, the LM is used as a generative model for learning patterns of the proposed representation from data. There can be more case studies and discussion on how additional capabilities (e.g. in-context learning) of LLMs can be leveraged with this representation. Do the authors see any emergent abilities? Are prompt-based approaches possible?

---

> ### Author Response · Authors · 2024-11-20
> **Response to Reviewer Y6S8**
>
> Dear Reviewer Y6S8,
>
> Thank you for your appreciation of our work and insightful comments! We have made much effort to thoroughly improve our work accordingly and provide responses for each concern here. Please also refer to the revised paper PDF.
>
> > W1 & W2: more (LM-based) baselines:
>
> - Thanks for your valuable comments. We would like to clarify that the track of `using LMs for 3D molecule generation` is largely under-explored, and the only comparable work we noticed is [1], which applies coordinates from XYZ files directly. The result is included in the Ablation study (please see Table 4 original coordinates, implemented by ourselves).
> - In addition, we have included 3 more baselines, JODO [2], MiDi [3], and Symphony [4] in our initial paper with many additional evaluation metrics [2,3,4]. Please refer to **Appendix D.3, Tables 6, 7, 8, 9, 10, and 11**. Vast experiments show the effectiveness of our method. Note that JODO and MiDi are diffusion models jointly generating 2D and 3D molecular information. Since the settings of these papers are not the same as ours, we exclude them in experiments of the main paper and provide in the appendix.
>
> > Q1: ... There can be more case studies and discussion on how additional capabilities (e.g. in-context learning) of LLMs can be leveraged with this representation. Do the authors see any emergent abilities? Are prompt-based approaches possible?
>
> - Thanks for your insightful comment. First, we would like to emphasize that one of our major motivation is to make use of the rapidly developing power of LMs; thus our Geo2Seq is orthogonal to the selected LM architectures. In the paper, we focus on the design of Geo2Seq and evaluate it using two backbone models (GPT and Mamba). Since our current method is well grounded and has shown empirical significance and great performances, we leave the exploration of further LLM techniques as a future direction.
> - However, we agree that some advanced LLMs techniques and case studies can introduce valuable insights into the field. We have provided some **scaling law studies (Appendix E.1)** since this provides typical insights regarding LMs. As shown in Table 14 and 15, LMs' performances on molecules grow significantly with parameter size increase, which is very similar to the **emergent abilities** widely-recognized in NLP tasks. We also provide **case studies ( Appendix E.2)** in our paper, where we show error cases with repetition or hallucinations, similar to those observed in NLP tasks. We observed these cases especially when not trained to best convergence, but they will be rarer if the model well converges.
> - Regarding prompt-based approaches, we believe that they can be applied here. In fact, for the controllable generation setting, we provide the model with **property values as the prompt**, and let the model generate novel molecules with this desired property. Moreover, we can leverage in-context learning to perform controllable generation on new properties without retraining a model for new properties. In-context learning can be also be extended to property prediction task, which could be an interesting research direction and we leave it for future work.
>
> We sincerely thank you for your time! Hope we have addressed your concerns through practical efforts and shown the contributions and significance of our work. We look forward to your reply and further discussions, thanks!
>
> Sincerely,
>
> Authors
>
> [1] Language models can generate molecules, materials, and protein binding sites directly in three dimensions as XYZ, CIF, and PDB files, 2023
>
> [2] Learning joint 2d & 3d diffusion models for complete molecule generation, 2023
>
> [3] Midi: Mixed graph and 3d denoising diffusion for molecule generation, 2023
>
> [4] Symphony: Symmetry-equivariant point-centered spherical harmonics for molecule generation

---

> > ### Comment · Reviewer_Y6S8 · 2024-11-29
> >
> > Thanks for your rebuttal. With only diffusion model baselines, it's difficult for me to evaluate how competitive your method is in the context of LM-based generation. I find it ironic you include a lot of discussion around CLMs, but none of them are baselines for your method. You could include baselines where a CLM generates the SMILES, and use standard conformation tools to generate the geometry. Anyways, I will keep my score.

---

> > > ### Author Response · Authors · 2024-12-02
> > > **Further Response to Reviewer Y6S8**
> > >
> > > Dear Reviewer Y6S8,
> > >
> > > Thank you for replying to our rebuttal! We would like to respond to your remaining concerns as follows.
> > >
> > > > CLM baselines
> > >
> > > - Following your suggestions, we would like to extend our experiments with several CLM baselines for more comprehensive evaluation. We add three baselines of CLMs for the random generation task on QM9 dataset, where CLMs generate SMILES while atom and molecule stability are calculated using conformers generated by RDKit and optimized with UFF (Universal Force Field) and MMFF (Merck Molecular Force Field). For MolGPT [1], we train the model on QM9 SMILES, for smiles-gpt [2], we fine-tune the model checkpoint on QM9 SMILES, and for MolReGPT [3], we perform 10-shot generation using QM9 SMILES and the backend model GPT-3.5-turbo. We provide the comparison results as below.
> > >
> > > | Method               | Atom Sta (%) | Mol Sta (%) | Valid (%)     | Valid & Unique (%) |
> > > |----------------------|--------------|-------------|---------------|---------------------|
> > > | Data                | 99.0         | 95.2        | 97.7          | 97.7               |
> > > | E-NFs               | 85.0         | 4.9         | 40.2          | 39.4               |
> > > | G-SchNet            | 95.7         | 68.1        | 85.5          | 80.3               |
> > > | GDM                 | 97.0         | 63.2        | -             | -                  |
> > > | GDM-AUG             | 97.6         | 71.6        | 90.4          | 89.5               |
> > > | EDM                 | 98.7±0.1     | 82.0±0.4    | 91.9±0.5      | 90.7±0.6           |
> > > | EDM-Bridge          | 98.8±0.1     | 84.6±0.3    | 92.0±0.1      | 90.7±0.1           |
> > > | GEOLDM              | **98.9**±0.1     | 89.4±0.5    | 93.8±0.4      | **92.7**±0.5           |
> > > | `MolGPT [1]`       | 92.6±0.2     | 29.0±0.3   | 94.5±0.2    | 94.2±0.2       |
> > > | `smiles-gpt [2] `   | 90.9±0.5     | 47.8±0.3    | 95.2±0.3   | 87.6±0.2       |
> > > | `MolReGPT [3]`    | 87.5±0.8     | 23.3±0.7    | 80.7±0.5     | 74.2±0.3           |
> > > | **Geo2Seq with GPT**    | 98.3±0.1     | 90.3±0.1    | 94.8±0.2      | 80.6±0.4           |
> > > | **Geo2Seq with Mamba**  | **98.9**±0.2     | **93.2**±0.2    | **97.1**±0.2      | 81.7±0.4           |
> > >
> > > - As shown in the table, the CLM methods does not perform as well as the strong diffusion baselines and our proposed method, especially in 3D-conformation-related metrics such as atom/molecule stability. Geo2Seq enables models to learn 3D structures, which directly encode spatial and geometric information. Inputting 3D information ensures a higher fidelity representation of molecular structures, reducing noise introduced by imprecise conformer generation, since RDKit-generated conformations are approximations and often lack precision.
> > >
> > > We hope we have shown the significance of our work through all the discussions. Since the discussion period is quickly approaching the end, we sincerely hope you can let us know if we have fully addressed your concerns. Also, we welcome any additional discussions or feedback. Thank you!
> > >
> > > Sincerely,
> > >
> > > Authors
> > >
> > >
> > > [1] MolGPT: Molecular Generation using a Transformer-Decoder Model
> > >
> > > [2] Generative Pre-Training from Molecules
> > >
> > > [3] Empowering Molecule Discovery for Molecule-Caption Translation with Large Language Models: A ChatGPT Perspective

---

### Official Review · Reviewer_EoEL · 2024-11-03

**Soundness:** 4
**Presentation:** 3
**Contribution:** 3
**Rating:** 6
**Confidence:** 3

**Summary:**

This paper explores language model (LM) based  generation of molecules in 3D space. To that end the paper proposes tokenization of molecular 3D graphs that converts molecular geometries into SE(3)-invariant 1D discrete sequences to be used as input to a LM. The paper proposes to a) first use graph canonicalization to create a 1D sequential ordering of the 3D molecule. b) use spherical representation to represent the 3D coordinate of the atoms. This ensures that any two isomorphic graphs have the same representation and different graphs have different representations. The LM model is next trained using typical next token prediction on the created token.  The experimental results on QM9 and GEOM-DRUGS dataset shows that their proposed Geo2Seq performs better than baselines both for random and controlled generations.

**Strengths:**

-- The problem description of converting 3D graphs to 1D tokens preserving SE(3)-invariance is very interesting and relevant to the community.

-- Moreover their proposed method Geo2Seq, which combines the two ideas from Graph theory (canonicalization and spherical representation) is novel (even though parts of it might be used in previous applications, for example canonicalization is used in SMILES).

-- Experimental studies on random and controlled generations across two dataset showcases the efficacy of the proposed method

-- Detailed Ablation Studies (sec C). One of the immediate questions I had was understanding the importance of using different types of ordering (one of which could be canonical) and different types of representation (including the proposed spherical). These questions have been asked and answered in the ablation (shown in the appendix).

**Weaknesses:**

-- As stated in the strength section, even though the combination is novel, the individual components have been used before.

-- One thing that is not clear to me from the very thorough experiment is from the following observation: From table-4 the gap between the top 3-4 methods is not very large. Under that circumstance, should efficiency and memory required to perform the ordering be used to pick a winner? Did the author do such an analysis?

**Questions:**

Other than the points mentioned in the weakness section, I have the following questions:

-- The proposed combination of canonical representation and spherical representation is nothing very specific to 3D molecules but can also be used by any 3D point cloud, did the authors try to see whether their proposed method works for other available 3D point cloud generation tasks (see pointflow or related papers for the tasks)?

---

> ### Author Response · Authors · 2024-11-20
> **Response to Reviewer EoEL (Part I)**
>
> Dear Reviewer EoEL,
>
> Thank you for your constructive comments and valuable suggestions! We have made much effort to thoroughly improve our work accordingly and provide responses for each concern here. Please also refer to the revised paper PDF.
>
> > W1: As stated in the strength section, even though the combination is novel, the individual components have been used before.
>
> - We appreciate your acknowledgement of the novelty of our proposed Geo2Seq. We understand your point that parts of our method have been used in previous application. However, we would like to emphasize that our major contribution lies in providing a solution for LM learning of 3D molecules which can achieve both structural completeness and geometric invariance. Theoretically we demonstrate that using canonicalization and spherical representation can achieve our goal.
> - More specifically, we summarize the key distinctions of our work as follows:
> 1. **The track is under-explored:** Different from mainstream methods, we convert 3D molecules to 1D discrete sequences and generate using LMs for 3D molecule generation tasks, positioning in an under-explored field.
> 2. **Technique distinctions:** The only comparable work using LMs applies coordinates from XYZ files directly. In contrast, Geo2Seq achieves **structural completeness** and **geometric invariance** for **LM training**, which is distinct from all existing works.
> 3. **Flexibility & comprehensive architecture:** Geo2Seq operates solely on the input data, which allows independence from model architecture and training techniques and provides reuse flexibility. To further demonstrate the universal applicability of our method, we not only include conventional Transformer architecture of CLMs, but also employ SSMs as LM backbones.
> 4. **Effectiveness & Efficiency:**  Results under both architectures prove the effectiveness of Geo2Seq. In addition, compared to diffusion models, our method outperforms them with significantly higher efficiency, *i.e.*, 100x faster generation speed.
>
> - In short, although this work stands on the shoulders of techniques from multiple fields, we have demonstrated our unique contributions, which is beneficial to the 3D molecular community.
>
> > W2: ...From table-4 (ablation study on different atom orders) the gap between the top 3-4 methods is not very large. ... should efficiency and memory ... be used to pick a winner? Did the author do such an analysis?
>
> - Thank you for your comment. We agree that the differences may not seem very large, but we would like to point out that there are still noticeable differences of around 0.5% (atom stability), **5.8% (molecule stability), 2.5% (valid), 8.6% (valid & unique)** between these top methods (Canonical-locality vs BFS). Notably, atom stability is a relatively easy metric, thus most methods can achieve 90%. Moreover, while the absolute improvement in these metrics is not large, utilizing canonical rigorousness to achieve these performance gains towards SoTA is both theoretically and practically meaningful. Thus we emphasize the information completeness and select canonical ordering, which grounds our contributions in well-defined theoretical principles.
> - Regarding efficiency, yes, we take it into consideration when evaluating our method. In fact, the results presented in the main tables (Table 1 and Table 2) are based on **Canonical-locality (Nauty algorithm) atom order because of its top performance, high efficiency, and rigorousness**. In detail:

---

> > ### Author Response · Authors · 2024-11-20
> > **Response to Reviewer EoEL (Part II)**
> >
> > | Algorithm | Average Time Complexity  | Space Complexity  | Note |
> > |--|--|--|--|
> > | Nauty (Canonical)  | O(n^2)| O(n^2)| Fastest for most graphs; efficient search and symmetry pruning|
> > | Bliss (Canonical) | O(n^3) | O(n^2) | Optimized for sparse graphs; scalable but slightly slower than Nauty|
> > | Saucy (Canonical)  | O(2^(n/2)) | O(n^2) | Focuses on sparse and symmetric graphs; slower for dense graphs |
> > | Traces (Canonical) | O(n^2) to O(n^3) | O(n^2) | Extension of Nauty; focuses on stable automorphism group ordering |
> > | McKay’s Algorithm (Canonical)  | O(n^4) | O(n^2) | Predecessor to Nauty; slower and less optimized |
> > | Brendan McKay’s Isomorphism Algorithm (Canonical)| O(n^4) | O(n^2) | Older implementation; lacks pruning optimizations|
> > | Morgan Algorithm | O(V + E)| O(V + E)| Fingerprinting for molecular graphs; not canonical|
> > | DFS | O(V + E) | O(V + E)| Linear in graph size, used for traversal |
> > | BFS| O(V + E)| O(V + E)| Linear; for unweighted shortest paths|
> > | Dijkstra| O((V + E) log V)| O(V + E)| For weighted graphs|
> > | Cuthill–McKee| O(V + E) | O(V + E)| Linear, optimized for sparse matrices|
> > | Hilbert-curve| O(n log n)| O(n)| Suitable for spatial data (e.g., geometry)|
> > | Random | O(1)| O(1)| Trivial, no computation involved|
> >
> >  1. The implementation of Nauty algorithm has the best time efficiency among all existing CL algorithms, as shown above.
> >  2. Canonical SMILES is based on the Morgan CL Algorithm, which is proven to be incomplete for isomorphism corner cases (such as two triangles versus one hexagon). While canonical SMILES solve corner cases by manual restrictions, it's a bit less elegant than Nauty Algorithm.
> >
> > > Q1: for 3D point cloud generation tasks (see pointflow or related papers for the tasks)
> >
> > - Thank you for your insightful comments. We appreciate your suggestion to explore the interesting applicability of our Geo2Seq towards 3D point cloud datasets. Indeed, our method can be extended to this domain, as the main difference between our 3D molecule data and 3D point cloud data (e.g., ShapeNet) is that node types (e.g. atom types for molecules) are not necessary in the latter case. Therefore, for 3D point cloud data, each node can be represented with three tokens (d, θ, φ). Note that we can also consider node attributes such as colors using additional node tokens if needed.
> > - Meanwhile, we also acknowledge that applying Geo2Seq to 3D point cloud datasets poses challenges due to the large size of point clouds. As mentioned in the paper, even for the large molecule dataset GEOM-DRUGS, the number of nodes is up to 181 atoms and 44.2 atoms on average. However, to our knowledge, the number of nodes for a point cloud should be much larger, e.g. over 2k nodes for ShapeNet. Training language models on such long sequences (over 6k if we consider 3 tokens for each node) is harder and requires further exploration of techniques like hierarchical methods and advanced LMs techniques for long sequences. In addition, 3D point cloud tasks poses a different setting with various baselines from the graphics fields. Thus, we focus on the field of molecules in this work and leave this exploration for future work.
> >
> >
> > We sincerely thank you for your time! Hope we have addressed your concerns through practical efforts and shown the contributions and significance of our work. We look forward to your reply and further discussions, thanks!
> >
> > Sincerely,
> >
> > Authors

---

> > > ### Comment · Reviewer_EoEL · 2024-11-24
> > >
> > > Thanks for the rebuttal. My questions have been answered. Scalability of the proposed approach (applicability to point cloud datasets with thousands of nodes) is a limitation but doesn't take away from the paper.  I still believe that the paper is above the acceptance threshold and my score would be somewhat in between 6 and 8, but I will keep my current score.  Thanks!

---

> > > > ### Author Response · Authors · 2024-11-26
> > > > **Thank you to Reviewer EoEL**
> > > >
> > > > Dear Reviewer EoEL,
> > > >
> > > > Thank you for replying to our rebuttal! We are grateful for your appreciation of our work and your recognition that we have addressed your questions. The applicability of our Geo2Seq towards 3D point cloud datasets is an interesting future direction for us. We sincerely thank you for your time and welcome any additional discussions or feedback.
> > > >
> > > > Sincerely,
> > > >
> > > > Authors

---

### Official Review · Reviewer_JuWn · 2024-11-03

**Soundness:** 4
**Presentation:** 3
**Contribution:** 2
**Rating:** 6
**Confidence:** 2

**Summary:**

This paper provides a novel approach to tokenization of molecules for the purposes of conditional and unconditional generation, focusing not on the model, but on ways of representing molecules as symbols for any model requiring a symbolic approach in a manner that is SE-3 invariant and efficient.

**Strengths:**

This work enables bridging geometric approaches to molecule representation with sequence-based molecule representation in a highly original and apparently successful manner.  The authors correctly identify the reasons why it can be difficult to translate inherently graphical information into a sequence representation, and then present both rigorous theoretical and extensive experimental evidence to show that they have arrived at a better solution than those existing to this problem.  I particularly appreciate the ablations and would suggest putting them earlier in the paper, as they convincingly show that these differences actually matter in real LLM's (not a trivial claim, since often apparently problematic objects to serialize/tokenize can be tokenized effectively given large enough models, e.g., in computer vision).

This paper is also particularly compelling as a work which includes extensive mathematical foundations, a very significant use case for the mathematical framework built up, and evidence that the mathematical framework is effective at solving that real-world problem.  I am not fully qualified to comment on the correctness of the proofs; however, to the best of my knowledge they seemed both valid and novel.

I appreciated the attention brought not just to the number of valid molecules, but also to the number of valid and unique and novel molecules, as arguably the most common "hidden failure mode" of transformers is merely regurgitating their training data.  Overall, the experiment section was clear and convincing.

**Weaknesses:**

I felt the controllable generation experiment was under-explained.  While they do list the six quantum properties in line 491 and discuss per-quantum-property efficacy in line 437, I would have preferred an analysis for each of these cases of instances where it succeeded and other methods failed or where their method failed and other methods succeeded.  More importantly, since the paper itself was on the mathematical foundation of a 1-d tokenization of 3-d molecules, I would have liked some explanation for each property of how the property relates to the 3-d configuration and what information relevant to that configuration is preserved or removed under the new framework, especially for readers with general knowledge of LLM's but without knowledge of chemistry.

Creating novel molecules is an interesting task; however, its usefulness was not made obvious in the paper (when would an applied researcher benefit from being able to generate a random valid molecule, or a molecule with a particular LUMO energy)?  The benefits of models such as AlphaFold were well documented in terms of medicinal chemistry; I was not able to find a similar reference explaining the benefits of random molecule generation, While it may be self-evident to researchers in the field of computational chemistry, to general readers of ICLR it would help to provide such a use case.  More ambitiously, it would be fascinating to see whether this approach could be applied to other geometric tokenization problems, such as generating valid representations of social networks with 2-d or 3-d constraints, or generating valid configurations of cellular automata.

Most LLM papers benefit strongly from looking at performance statistics as a function of model size and other hyperparameters; which I did not see here (there is a section on hyperparameters in the appendix, but not a comprehensive study involving the type of LLM used).

**Questions:**

1.  What do you think the use cases are for your work?
2. I was somewhat confused by line 258 - why would the coordinates being bound in a smaller region be helpful?  A simple clarification would be satisfactory
3. Could you be more specific about, of the proofs in the appendix, which are taken from existing literature, and which are novel to this paper?
4. When it says in line 1735 "π-out-of-phase angles are placed near each other, such as ‘3.14°’, ‘-3.14°’, and ‘0°’" - isn't this a sign that you're not capturing some aspect of invariance?

---

> ### Author Response · Authors · 2024-11-20
> **Response to Reviewer JuWn (Part I)**
>
> Dear Reviewer JuWn,
>
> Thank you for your constructive comments and valuable suggestions! We have made much effort to thoroughly improve our work accordingly and provide responses for each concern here. Please also refer to the revised paper PDF.
>
> > W1: Controllable generation experiment was under-explained ... analysis for each of these cases of instances where it succeeded and other methods failed or ... some explanation for each property of how the property relates to the 3-d configuration and what information relevant to that configuration is preserved or removed under the new framework, especially for readers with general knowledge of LLM's but without knowledge of chemistry.
>
> Thanks for your comments.
> - As shown in Table 2, among all six properties, our method outperforms the diffusion methods EDM and GEOLDM by a large margin, moving a significant step towards the lower bound. We can observe that, our method has significant improvement over baseline methods on the controllable generation, while the improvement on random generation is relatively small. This means that although baseline methods (diffusion-based methods) can generate novel molecules, they can't use conditional information appropriately, showing the advantage of our LM-based methods.
>
>   In detail, compared to LMs which can use prompt as conditions, baseline methods (e.g. diffusion based methods) take a property c as additional input which is concatenated to the nodes features. This makes the model hard to train and limit their performance.
>
> - Regarding the properties, all properties are related to the 3D configurations. For example, for different conformers of the same molecules (considering the same atoms and chemical bonds, but some bonds are rotated, therefore, the 3D structure becomes different), the property value of mu (Dipole moment) can change a lot. This is because the dipole moment is a vector quantity that depends on the orientation of the molecule in 3D space. As the theoretical analysis guarantees, Geo2Seq can achieve structural completeness and geometric invariance with the numerical precision of $ϵ\leq|10-b|/2$, since canonical labeling reduces graph structures to 1D sequences with no information loss regarding graph isomorphism, and SE(3)-invariant spherical representations that ensure no 3D information loss. Thus atomic information and nearly-precise 3D information are preserved, which the properties depend on. Below are the details of these properties:
>    - α Polarizability: Tendency of a molecule to acquire an electric dipole moment when subjected to an external electric field.
>    - HOMO: Highest occupied molecular orbital energy.
>    - LUMO: Lowest unoccupied molecular orbital energy.
>    - Gap: The energy difference between HOMO and LUMO.
>    - µ: Dipole moment which depends on the orientation of the molecule in 3D space.
>    - Cv: Heat capacity at 298.15K which is related to the vibration of molecules
> - We have added these information in the paper revision.

---

> > ### Author Response · Authors · 2024-11-20
> > **Response to Reviewer JuWn (Part II)**
> >
> > > W2 & Q1: Creating novel molecules is interesting task ... usefulness was not made obvious in the paper (when would an applied researcher benefit from being able to generate a valid molecule or a molecule with a particular LUMO energy) ... fascinating to see whether this approach could be applied to other geometric tokenization problems, such as generating valid representations of social networks with 2-d or 3-d constraints, or generating valid configurations of cellular automata. (Q1) Use cases…
> >
> > Thanks for your comments.
> > - We provide some detailed examples below to show the use case.
> >
> >   For controllable generation task, we aim to generate molecules with specific properties.
> >   One example property is gap which refers to HOMO-LUMO energy gap. This property is an important measure for designing organic semiconductors, an essential component of e.g. organic solar cells and OLED displays [1,2]. A smaller HOMO-LUMO gap can improve electrical conductivity and optoelectronic properties.
> >
> >   Our controlled generation method has the potential to generate new molecules with smaller HOMO-LUMO gaps, thereby accelerating the discovery of new materials. We have added these information in the paper revision.
> >
> > - Regarding the application to other geometric tokenization problems, we believe that our method can be extended to different domains and Geo2Seq can work on the new datasets. Reviewer EoEL also suggested to explore the interesting applicability of our Geo2Seq towards 3D point cloud datasets. Indeed our method can be extended to these domains, as the main difference between our 3D molecule data and social network/cellular data/3D point cloud data (e.g., ShapeNet) is that node types (e.g. atom types for molecules) might not be necessary in the latter cases. Therefore, for these geometric data, each node can be represented with three tokens (d, θ, φ), while we can also consider node attributes such as colors using additional node tokens if needed. Meanwhile, there can be challenges due to the large size of these 3D graphs. As mentioned in the paper, for the molecule dataset GEOM-DRUGS, the number of nodes is up to 181 atoms and 44.2 atoms on average. However, the number of nodes for, e.g., a point cloud, should be much larger, e.g. over 2k nodes for ShapeNet. Training language models on such long sequences (over 6k if we consider 3 tokens for each node) is harder and requires further exploration of techniques like fragmentation/hierarchical methods and advanced LMs techniques for long sequences. Still, these other tasks pose different settings with various baselines from fields such as graphics. Thus, we focus on the field of molecules in this work and leave these fascinating explorations for future work.
> >
> > > W3: ... looking at performance statistics as a function of model size and other hyperparameters; which I did not see here (there is a section on hyperparameters in the appendix, but not a comprehensive study involving the type of LLM used).
> >
> > - Thanks for your point. We would like to clarify that our paper does include results with different model sizes ("scaling laws") in **Appendix E.1**, since this provides typical insights for LMs.
> >
> >   As shown in Table 14 and 15 (also provided below), LMs' performances on molecules grow significantly with parameter size increase, similar to the emergent abilities widely-recognized in NLP tasks.
> >
> > |Parameter size - GPT| 2,556,532| 31,309,824| 61,650,944| 88,012,800| 116,342,688|
> > |---|---|---|---|---|---|
> > |Atom sta(%)| 76.2| 89.6| 96.5| 98.3| 98.5|
> > |Mol sta(%)| 5.1| 42.4| 81.3| 89.1| 90.6|
> > |Valid(%)| 45.5| 73.1| 90.9| 94.3| 95.1|
> > |Valid & Unique(%)| 43.4| 66.7| 83.6| 74.9| 78.6|
> >
> > |Parameter size - Mamba| 2,180,352| 31,458,048| 61,631,232| 93,088,512| 121,977,600|
> > |---|---|---|---|---|---|
> > |Atom sta(%)| 81.6| 95.7| 97.4| 97.8| 97.9|
> > |Mol sta(%)| 13.6| 79.2| 86.8| 88.3| 89.0|
> > |Valid(%)| 51.2| 89.4| 93| 93.7| 94.4|
> > |Valid & Unique(%)| 49.6| 78.7| 78.8| 82.6| 83.5|
> >
> > > Q2: I was somewhat confused by line 258 - why would the coordinates being bound in a smaller region be helpful? A simple clarification would be satisfactory
> >
> > - The 3D coordinates are real numbers. In our method, we need to discretize these numbers in order to tokenize and make use of LMs. Thus in practice, we round up numerical values to certain decimal places. Spherical coordinate values are bounded in a smaller region compared to original coordinates, therefore,
> >   - given the same decimal place fixed, spherical coordinates creates a **smaller vocabulary size**;
> >   - and given the same vocabulary size constraint, spherical coordinates present **less information loss**.
> > - In addition, we have empirically observed that this benefits LM learning significantly.

---

> > > ### Author Response · Authors · 2024-11-20
> > > **Response to Reviewer JuWn (Part III)**
> > >
> > > > Q3: Could you be more specific about, of the proofs in the appendix, which are taken from existing literature, and which are novel to this paper?
> > >
> > > Thanks for your point. We provide the following clarification:
> > > - B.1 Proof of Lemma 3.2: we didn't provide detailed proof but provided a reference, as it can be found from existing literature.
> > > - B.2 proof of Lemma 3.3: we proved this Lemma using coordinate transformation under local frames, which is novel to this paper.
> > > - B.3 and B.4: we prove our Theorem, and it is novel to this paper.
> > >
> > > > Q4: When it says in line 1735 "π-out-of-phase angles are placed near each other, such as '3.14°', '-3.14°', and '0°' - isn't this a sign that you're not capturing some aspect of invariance?
> > >
> > > - Thank you for your insightful comment. The observation that π-out-of-phase angles are placed near each other in the learned embedding space does not indicate a failure to capture invariance. Instead, it highlights a meaningful representation of angular periodicity. Below, we clarify why this placement aligns with the desired invariance properties.
> > > - The angle ranges from -π to π, and '3.14°' (-π) and '-3.14°' (π) are mathematically equivalent under angular periodicity (the difference is produced by the minor differences of the precise angle values) as angles wrap around a 2π-range in spherical coordinates. This is consistent with our semantic closeness visualization ('3.14°' and '-3.14°' are near) because they represent the same orientation in physical space. In addition, '3.14°' and '0°' angle values can correspond to chiral molecules (a pair of very similar molecules, can be mapped through a mirror), therefore it is reasonable that there are near in the plot. This behavior reflects the model's ability to capture rotational invariance and periodicity, indicating that the model has successfully understood 3D geometric information from the tokens and generalize well across angular periodic boundaries.
> > >
> > > We sincerely thank you for your time! Hope we have addressed your concerns through practical efforts and shown the contributions and significance of our work. We look forward to your reply and further discussions, thanks!
> > >
> > > Sincerely,
> > >
> > > Authors
> > >
> > > [1] Symmetry-adapted generation of 3d point sets for the
> > > targeted discovery of molecules, 2019
> > > [2] Design and Synthesis of Low HOMO-LUMO Gap N-Phenylcarbazole-Substituted Diketopyrrolopyrroles, 2016

---

> > > > ### Comment · Reviewer_JuWn · 2024-11-24
> > > > **Response to Authors**
> > > >
> > > > Thank you, you have addressed my concerns.  I will keep the score the same due to a new understanding of the issues other reviewers brought up, but am less concerned about the original concerns, and believe this paper is (as my original score indicates) not unworthy of being accepted.

---

> > > > > ### Author Response · Authors · 2024-12-02
> > > > > **Follow-up to Reviewer JuWn**
> > > > >
> > > > > Dear Reviewer JuWn,
> > > > >
> > > > > Thank you for replying to our rebuttal! We are grateful for your support of our work's acceptance and your recognition that we have addressed your questions. Regarding the questions other Reviewers brought up, we have provided detailed responses and further experiments to all remaining questions. Please refer to our discussions with other Reviewers and let us know if we have fully addressed any questions you might have.
> > > > >
> > > > > We sincerely thank you for your time! Hope we have shown the significance of our work through all the discussions.  Also, we welcome any additional discussions or feedback. Thank you!
> > > > >
> > > > >
> > > > > Sincerely,
> > > > >
> > > > > Authors

---

### Official Review · Reviewer_3p53 · 2024-11-04

**Soundness:** 4
**Presentation:** 4
**Contribution:** 3
**Rating:** 10
**Confidence:** 3

**Summary:**

This paper tackles the problem of 3D molecular generation from scratch. The authors propose the tokenization method Geo2Seq, that converts 3D molecular graphs into SE(3)-invariant sequences for further uses in LMs.
It is shown that by using this novel tokenization method, various LMs can generate 3D molecular structures with high validity, outperforming diffusion-based baselines.

**Strengths:**

- Proposed tokenization method Geo2Seq is agnostic to the used LMs, which provides flexibility in method usage [SIgnificance].
- [Clarity] The paper outline and structure are well defined and easy to follow.
- [Quality] Provision of additional random generation results for QM9 dataset, as well as additional experiments on more baselines and metrics perfectly visualize the novel method performance compared with existing methodologies.
- [Originality] Usage of spherical representations for 3D structures without information loss.

- This work is provided with a substantial number of definitions and lemmas that help navigate the problem.

**Weaknesses:**

I believe this article has everything to be called a good paper without any major concerns.

**Questions:**

I could find the answers to most of my questions in paper. Thank you for providing such a detailed report of the work.

---

> ### Author Response · Authors · 2024-11-20
> **Response to Reviewer 3p53**
>
> Dear Reviewer 3p53,
>
> Thank you for your appreciation of our work and insightful comments! We provide responses for each concern here. Following other reviewers' advice, we have made much effort to thoroughly improve our work. Please also refer to the revised paper PDF and our responses to other reviewers. Also, we welcome any additional discussions or feedback. Thank you!
>
> Sincerely,
>
> Authors

---

### Official Review · Reviewer_J6yi · 2024-11-04

**Soundness:** 2
**Presentation:** 2
**Contribution:** 2
**Rating:** 3
**Confidence:** 3

**Summary:**

This paper proposes Geo2Seq tokenization scheme that is meant to enable generation of 3D molecules using autoregressive LLMs.
It canonicalizes the graph and creates an input dependent frame of reference with spherical coordinates to encode positions as strings to be provided for the autoregressive model.

**Strengths:**

The work provides a way to generate 3D conformers using autoregressive generation.
The bijective mapping between the molecular graph and the sequences produced by Geo2Seq is demonstrated.

**Weaknesses:**

A few minor stylistic points:
- Line 111: what is 'molecular science' ?
- Line 80, 401: perhaps "conditional generation" would be a more precise term than "Controllable generation".
- Line 107: "LLMs have revolutionized the landscape of NLP and beyond." - what is the "NLP and beyond" field?

The proposed tokenization scheme relies on the model to be able to 'understand' the 'number tokens'. The appendix shows UMAP plots demonstrating that the models learn some spherical-like structure from those tokens. Could adding the SE(3) invariant position embeddings be a more efficient way to achieve this?
The vocabulary size is reported to be '1.8K for QM9 dataset' and '16K for Geom-Drug' for problems with single digit number of possible atoms. Could this be a sign of overfitting?
The autoregressive models used are also much larger than the GNN baselines (~90M parameters vs ~5M), could the performance gains be a result of this?

**Questions:**

How would the method perform if the number of tokens in the tokenizer would be restricted to 50, 100, 500, 1000 ?
Why does mamba seems to perform better than gpt in your evaluation of random generation (table 1)? Why is gpt not featured in table 2?
Would adding the position information on embedding level make more sense?
How is the Geo2Seq canonicalization different from canonical SMILES?
Why would the cartesian coordinate tokens with the same frame as Geo2Seq not be rotation invariant?

---

> ### Author Response · Authors · 2024-11-20
> **Response to Reviewer J6yi (Part I)**
>
> Dear Reviewer J6yi,
>
> Thank you for your constructive comments and valuable suggestions!
> We have made much effort to thoroughly improve our work accordingly and provide responses for each concern here. Please also refer to the revised paper PDF.
>
> > W1. Stylistic points
>
> 1. Thank you for your question. "Molecular science" in this context refers to the interdisciplinary study of molecules, their structures, properties, interactions, and functions. This encompasses fields such as computational chemistry, drug discovery, material science, and biochemistry, where understanding and modeling molecular behavior are crucial. In our paper, we specifically highlight how language models (LMs) contribute to molecular science by representing molecules and enabling tasks like molecular property prediction, molecule generation, and interaction modeling. We use this term following prior works such as [1,2] and chemistry journals such as [3,4,5].
> 2. Thank you for the suggestion. It's true that both terms have been used for similar tasks [6,7,8], and we choose to use "controllable generation" following our baselines EDM[9] and GeoLDM[10] in order to keep consistency. To improve clarity, we have added the discussion on the equivalence of these two terms.
> 3. By "NLP and beyond," we refer to the application of large language models not only in traditional natural language processing (NLP) tasks such as text generation, sentiment analysis, and machine translation but also in domains outside conventional NLP. These include areas like daily Q&A, education, software development, scientific discovery, and multi-modal tasks. We have added explanations in the paper revision.
>
> > W2. Position embeddings for token understanding
>
> - Thank you for the question. We would like to clarify that one of our major motivation is to make use of the rapidly developing power of LMs; thus our goal is to design a **model-agnostic** method, without any need of model-level modification. We admit that `position embeddings for coordinates` might help the model achieve similar token understanding; however, it requires specific embedding design, which needs to be done separately for each LM architecture and can be infeasible for modern LMs released via APIs. Geo2Seq operates solely on the input data, which allows independence from model architecture and training techniques and provides reuse flexibility. In this case, we can apply Geo2Seq on newly released LMs, which we believe is more efficient compared to the workload of positional embedding design.
> - To improve clarity, we have further emphasized this motivation and advantage of Geo2Seq in the paper revision.
>
> > W3. Vocabulary size... overfitting?
>
> - First, we would like to clarify that the vocabulary size reported ('1.8K for QM9' and '16K for Geom-Drug') is not for `single digit number`; we use the precision of 2 decimal places, and this vocabulary covers the numerical token range of distance and angle values, as specified in Sec 3.3. In addition, we believe a vocabulary size of 1.8k/16k is not abnormal for LMs with 80M/100M parameters, since for a 100M-parameterLM in NLP, a vocabulary size of 10k to 50k tokens is typical [11,12].
> - Next, we would like to clarify that our results are not from overfitting. We use the cross entropy loss for LM training, which does not have direct mathematical relationships with any of the chemical metrics, such as validity, stability, uniqueness, novelty, and so on. We use a validation set to monitor whether overfitting happens. For QM9 dataset, we select checkpoint before overfitting happens to perform final evaluation so that we get good performances across all metrics. Further experiments show that one metric validity does get better as the LM overfits, but other metrics such as stability, uniqueness and novelty would suffer when the LM overfits. For GEOM-DRUG dataset, we have not observed overfitting throughout our training of 20 and 25 epoch for GPT and Mamba models, respectively. We include these information in the paper revision as well.

---

> > ### Author Response · Authors · 2024-11-20
> > **Response to Reviewer J6yi (Part II)**
> >
> > > W4. Larger parameters than GNN
> >
> > Thank you for the point.
> > - First, we would like to clarify that our baseline EDM (and GeoLDM) which reports 5M parameters is a diffusion-based method rather than a GNN, since GNNs are primarily used for prediction tasks. Diffusion methods are the previous SoTA in this task, but have disadvantages in efficiency, which is also our motivation of using LMs. From Table 12, we can observe that though we need larger parameter and more memory compared to diffusion-based methods, our time efficiency is much better than diffusion-based methods. Throughput, or samples per second, is one of the most important metrics to measure generation efficiency. In particular, Geo2Seq with Mamba is more than 100 times faster than diffusion-based methods, which is a significant advantage in practical applications where speed is crucial.
> > - Considering whether the results rely on an extensive parameterization, we believe that LMs do need a relatively large parameter size in order to achieve decent performances. However, we do not think this undermines our contributions. Our major technical contribution lies in the Geo2Seq tokenization, which can then be used together with any LM. We adopt existing language models architectures, which typically need large parameter size to exhibit capabilities. For diffusion models which learns to gradually denoise data, their learning process doesn't inherently require as many parameters as LMs. Nevertheless, our method also show several advantages of LMs over diffusion models, including excellence in certain tasks and metrics, and significant better generation speed. Moreover, the innovation of using LMs in this domain can be a pathway to new methods and breakthroughs.
> > - We also provide scaling law studies in Appx. E.1, which show the the performance comparisons of models at various parameter sizes.
> >
> > > Q1. Performance under restricted tokens
> >
> > - Thanks for the question. We would like to provide ablation studies on vocabulary size. Our tokens are naturally defined chemical symbols and numbers rather than subword extractions such as BPE or SentencePiece. Therefore, in order to maintain semantic information we cannot require the vocabulary size to be rigorously `500`. Instead, we use different numerical tokenization rules to control the vocabulary size.
> >   - Following the same number split, we get vocabulary sizes of 194 and 1.8K if we consider 1 and 2 decimal places, respectively.
> >   - Instead of simply taking the complete real number as a token, we can split it by the decimal point and treat every part as an individual token, which will result in vocabulary sizes of 26 and 122 if we consider 1 and 2 decimal places, respectively.
> > - We provide experiment results using 8-layer GPT models on QM9 dataset:
> >
> > | Vocab Size| Atom Sta (%) | Mol Sta (%) | Valid (%) | Valid & Unique (%) |
> > |--|--|--|--|--|
> > | 28 |84.2|71.3|80.5| 73.3|
> > | 124 |96.4|80.3|89.9| 74.4|
> > | 196|95.9|81.6|90.4|75.1|
> > | 1.8K|97.4|86.8|93.0|84.7|
> >
> > - As shown above, our used tokenization leads to better performance. This shows that treating the complete real number as an individual token and using more decimal places enable LMs to capture 3D molecular structures more effectively.
> >
> > > Q2. Mamba v.s. GPT in Table 1&2
> >
> > - Thank you for the question. We do observe that Mamba performs slightly better than GPT in our random generation task on QM9 dataset. This could be attributed to the below reasons. Given similar parameter size, the layer count of Mamba doubles that of a Transformer, as two Mamba blocks are needed for each "layer" (MHA block + MLP block) of a Transformer [13]. This could allow Mamba to build deeper and thus more expressive hierarchical representations. Additionally, state-space models handle sequential dependency modeling more directly, while GPT's attention mechanism captures global relationships well. When sequential dependency is critical, Mamba might be performing better. Still, the performance of both models are generally comparable.
> > - Thank you for this insightful suggestion. We would like to extend the performance of GPT in Table 2. Initially, we only explore Mamba in conditional generation due to its better time efficiency. The results are as below(just including important baselines):
> >
> > | Property (Units)       | α (Bohr³) | Δε (meV) | ε_HOMO (meV) | ε_LUMO (meV) | μ (D)  | C_v (cal/mol·K) |
> > |--|-|-|--|--|--|--|
> > |Data|0.10| 64| 39 | 36| 0.043| 0.040|
> > |Random |9.01| 1470| 645| 1457|1.616| 6.857|
> > |GEOLDM|2.37| 587| 340| 522 | 1.108  | 1.025           |
> > |Geo2Seq with Mamba| **0.46**| **98**| 57 | 71| 0.164 | **0.275**|
> > |Geo2Seq with GPT| 0.53|102|**48**|**53**|**0.097**|0.325|
> >
> > - As shown above, Geo2Seq with GPT achieves significantly better results over the baselines and outperforms Geo2Seq with Mamba in several properties. We have included the results and discussions in the paper revision.

---

> > > ### Author Response · Authors · 2024-11-20
> > > **Response to Reviewer J6yi (Part III)**
> > >
> > > > Q3. Adding position information on embedding level
> > >
> > > - Please refer to our response to W2.
> > >
> > > > Q4. Canonicalization comparison to canonical SMILES
> > >
> > > - Thank you for the point. As we specified in Sec 3.1 and Appx. C, our theoretical analyses and derivations apply to all rigorous CL algorithms. We do not propose a new CL algorithm; all rigorous CL algorithms are usable here, while our contribution lies in achieving structural completeness and geometric invariance for LM learning of 3D molecules. In the paper, we implement Nauty Algorithm for 3D molecules because:
> > >   1. Its implementation has the best time efficiency among all existing CL algorithms.
> > >   2. Canonical SMILES is based on the Morgan CL Algorithm, which is proven to be incomplete for isomorphism corner cases (such as two triangles versus one hexagon). While canonical SMILES solve corner cases by manual restrictions, it's a bit less elegant than Nauty Algorithm.
> > > - We studied the influence of ordering algorithms in `Ablation on atom order` of Appendix C. Following your advice, we further extend to include canonical SMILES order, as shown below.
> > >
> > > |Order|Atom Sta(%)|Mol Sta(%)|Valid(%)|Valid&Unique(%)|
> > > |---|---|---|---|---|
> > > |Canonical-locality|**97.39**|**86.77**|**92.97**|**84.71**|
> > > |Canonical-SMILES|97.35|85.86|92.97|84.05|
> > > |Canonical-nonlocality|96.45|81.36|90.89|83.37|
> > > |DFS|95.95|81.54|90.45|82.48|
> > > |BFS|96.85|80.92|90.49|76.13|
> > > |Dijkstra|95.29|77.25|88.97|73.52|
> > > |Cuthill–McKee|93.56|71.57|85.36|76.23|
> > > |Hilbert-curve|90.11|64.99|80.40|67.83|
> > > |Random|64.87|20.14|43.16|38.44|
> > >
> > > - Our implemented Nauty Algorithm for 3D molecules can perform partitioning of graph vertices (a step in Nauty) with/without locality considered. Canonicalization with locality considered can lead to better results, due to the importance of neighboring atom interactions in molecular evaluations. Given the similar nature, canonical SMILES produces a very similar ordering with "Nauty with locality", thus close in performances.
> > >
> > > > Q5. Cartesian coordinate with the same frame not rotation invariant?
> > >
> > > - It is invariant. As we discussed in Sec 3.2 and `Ablation on 3D representation` of Appx. C, we can have invariant Cartesian coordinates when our proposed equivariant frame is applied.
> > >  - Specifically, we compare the spherical coordinates in Geo2Seq with the direct use of Cartesian coordinates of atoms from xyz data files as well as the use of SE(3)-invariant Cartesian coordinates that are projected to the equivariant frame proposed in Section 3.2. Results in **Table 4** demonstrate that LMs achieve the best performance on invariant spherical coordinates, showing the superiority of invariant spherical coordinates over invariant Cartesian coordinates.
> > > - In Sec 3.2, we discuss the advantage of spherical coordinates. Compared to Cartesian coordinates, spherical coordinate values are bounded in a smaller region, namely, a range of $[0,\pi]$/$[0,2\pi]$. Given the same decimal place constraints, spherical coordinates require a smaller vocabulary size, and given the same vocabulary size, spherical coordinates present less information loss. This makes spherical coordinates advantageous in discretized representations and thus easier to be modeled by LMs. Considering our theoretical results, Lemma 3.3 and its proof aim to guarantee the validity that our proposed invariant spherical representations possess $SE(3)$-invariance. We consider it as a part of our theoretical contribution towards the derivation of Theorem 3.5 instead of an advantage over other possible 3D representations.
> > >
> > > > Novelty and Significance
> > >
> > > Finally, we would like to demonstrate the contributions and distinctions of our paper clearly:
> > > 1. **The track is under-explored:** Different from mainstream methods, we convert 3D molecules to 1D discrete sequences and generate using LMs for 3D molecule generation tasks, positioning in an under-explored field.
> > > 2. **Technique distinctions:** The only comparable work using LMs applies coordinates from XYZ files directly. In contrast, Geo2Seq achieves **structural completeness** and **geometric invariance** for **LM training**, which is distinct from all existing works.
> > > 3. **Flexibility & comprehensive architecture:** Geo2Seq operates solely on the input data, which allows independence from model architecture and training techniques and provides reuse flexibility. To further demonstrate the universal applicability of our method, we not only include conventional Transformer architecture of CLMs, but also employ SSMs as LM backbones.
> > > 4. **Effectiveness & Efficiency:**  Results under both architectures prove the effectiveness of Geo2Seq. In addition, compared to diffusion models, our method outperforms them with significantly higher efficiency, *i.e.*, 100x faster generation speed.

---

> > > > ### Author Response · Authors · 2024-11-20
> > > > **Response to Reviewer J6yi (Part IV)**
> > > >
> > > > We believe our work would be beneficial to the 3D molecular community and would genuinely appreciate your recognition.
> > > >
> > > > We sincerely thank you for your time! Hope we have addressed your concerns through practical efforts and shown the contributions and significance of our work. We look forward to your reply and further discussions, thanks!
> > > >
> > > > Sincerely,
> > > >
> > > > Authors
> > > >
> > > > [1] New Basis Set Exchange: An Open, Up-to-Date Resource for the Molecular Sciences Community, 2019
> > > >
> > > > [2] PotentialNet for molecular property prediction, 2018
> > > >
> > > > [3] International Journal of Molecular Sciences
> > > >
> > > > [4] WIREs Computational Molecular Science
> > > >
> > > > [5] Molecular Sciences (ISSN: 2998-8977)
> > > >
> > > > [6] Retrieval-based controllable molecule generation, 2022
> > > >
> > > > [7] C5t5: Controllable generation of organic molecules with transformers, 2021
> > > >
> > > > [8] CProMG: controllable protein-oriented molecule generation with desired binding affinity and drug-like properties, 2023
> > > >
> > > > [9] Equivariant diffusion for molecule generation in 3d
> > > >
> > > > [10] Geometric Latent Diffusion Models for 3D Molecule Generation
> > > >
> > > > [11] Scaling Laws with Vocabulary: Larger Models Deserve Larger Vocabularies, 2024
> > > >
> > > > [12] Scaling Laws for Neural Language Models, 2020
> > > >
> > > > [13] Mamba: Linear-Time Sequence Modeling with Selective State Spaces, 2023

---

> > > > > ### Comment · Reviewer_J6yi · 2024-11-25
> > > > >
> > > > > I would like to thank the authors for the effort invested into this rebuttal.
> > > > > Many of the points raised previously have been addressed.
> > > > > However, some of the key issues are still unresolved:
> > > > >
> > > > > - Q1. Performance under restricted number of tokens:
> > > > >
> > > > >    It can be clearly observed, that the approach underperforms at reasonable number of tokens, suggesting that the 1.8K tokenization is overfitting.
> > > > >    Indeed, 100k+ vocabulary size is appropriate for general purpose language models.
> > > > >    However, the chemistry problems involve much fewer unique 'words', e.g. atomic symbols and (optionally) frequent substructures.
> > > > >    Are 'naturally defined chemical symbols and numbers rather than subword extractions such as BPE or SentencePiece' truly a better way to split the textual representation into tokens?
> > > > >
> > > > > - On the general applicability of the tokenization scheme.
> > > > >
> > > > >    Throughout the rebuttal, the general applicability of the tokenization scheme is emphasised.
> > > > >    However, using this approach requires the ability to at least fine tune the model embeddings.
> > > > >    Therefore one could use positional embeddings instead.
> > > > >    This would save the context length used, among other things, and possibly be more versatile than discretizing rational numbers into vocabulary.
> > > > >
> > > > > Although I have read and understand the assessment of other reviewers, I find the paper in its current form not to be ready for publication and have therefore decided to retain my score.

---

> > > > > > ### Author Response · Authors · 2024-11-25
> > > > > > **Further Response to Reviewer J6yi (Part I)**
> > > > > >
> > > > > > Dear Reviewer J6yi,
> > > > > >
> > > > > > Thank you for your reply. We are glad to know that many of your concerns have been addressed, and would like to respond to your remaining concerns as follows.
> > > > > >
> > > > > > > Q1. Performance under restricted number of tokens
> > > > > >
> > > > > > We would like to address the points raised about tokenization, vocabulary size, and performance under different tokenization schemes.
> > > > > > - **Performance should not be judged solely by vocabulary size, and the experiments evidence against overfitting.**
> > > > > >   - First, we respectfully disagree with the conclusion that a larger vocabulary size inherently suggests overfitting. The number of tokens is a byproduct of the chosen tokenization scheme, and a larger vocabulary size can also reflect better semantic granularity and preservation of the original information. In our experiments, we observed that the **performance difference is primarily influenced by tokenization design**, rather than solely by the absolute size of the vocabulary. A smaller vocabulary size often results in semantic over-fragmentation (e.g., splitting meaningful symbols into too many smaller subparts), which degrades the quality of downstream tasks.
> > > > > >   - Second, we believe the observation of "underperformance under reasonable token counts" is a misunderstanding. We would like to emphasize that our [experiments on different vocabulary size](https://openreview.net/forum?id=HbZrxBXzks&noteId=n6RJxSTRfn) is `performed with different tokenization schemes`, while our Geo2Seq uses complete numbers in tokenization. We provide an expanded experimental table here. As shown below, **comparing across the different tokenization methods, performances are largely influenced by tokenization rather than the absolute vocabulary size**. previously, we have also conducted experiments using Byte Pair Encoding (BPE). However, as noted in our response to `Reviewer aUY2` and shown below, BPE-based tokenization significantly underperformed compared to naturally defined chemical tokens, as it breaks down chemical semantics (e.g., numeric properties) into subwords that lack chemical interpretability. In contrast, our Comp-tokenization leads to better performance, showing that treating the complete real number as an individual token and using more decimal places enable LMs to capture 3D molecular structures more effectively. More evidently, **even with smaller vocabulary size, our adopted Comp-tokenization performs significantly better than BPE**, which clearly shows that our performance gain is not from overfitting but from the model’s better understanding of information.
> > > > > >
> > > > > > | Tokenization (Vocab Size)| Atom Sta (%) | Mol Sta (%) | Valid (%) | Valid & Unique (%) |
> > > > > > |--|--|--|--|--|
> > > > > > | Char-tokenization (25) |90.5|43.7|71.5| 71.0|
> > > > > > | Sub-tokenization (28) |84.2|71.3|80.5| 73.3|
> > > > > > | Sub-tokenization (124) |96.4|80.3|89.9| 74.4|
> > > > > > | Comp-tokenization (196) |95.9|81.6|90.4|75.1|
> > > > > > | BPE (551) |85.3|55.3|74.4| 57.6|
> > > > > > | Comp-tokenization (1.8K) |97.4|86.8|93.0|84.7|
> > > > > >
> > > > > > - **Further experimental evidence against overfitting:**
> > > > > >   - Moreover, while we cannot know the quantitative optimal of vocabulary size for chemistry problems, we observe experimental evidence against overfitting.
> > > > > >   - We use the cross entropy loss for LM training, which does not have direct mathematical relationships with any of the chemical metrics, such as validity, stability, uniqueness, novelty, and so on. Moreover, Table 6 reports further random generation results on QM9, including novelty metric, while Table 7 presents the novelty results of controllable generation. Results show that our method achieves reasonably high **novelty scores**, which is surely **not achievable through overfitting**.
> > > > > >   - We also provided evaluation results on various chemical metrics in Appx. D, Table 6, 7, 8, 9, 10, and 11 . Vast results show that we can outperform existing methods across metrics, demonstrating that our model is not merely memorizing or overfitting on the training data, but understands underlying chemical information.
> > > > > >
> > > > > > - Analyses on why naturally defined tokens are better:
> > > > > >   - The chemical domain has unique characteristics, such as atomic symbols, molecular substructures, and numeric properties, that differ significantly from general-purpose natural language. Subword-based tokenization schemes like BPE or SentencePiece tend to break down chemically meaningful units, especially numerical information, into smaller, non-interpretable pieces. This loss of semantic integrity leads to degraded performance on chemistry-specific tasks. Our tokenization scheme ensures that atomic symbols and numbers are represented as standalone tokens, which directly aligns with the underlying chemistry and enhances interpretability.

---

> > > > > > > ### Author Response · Authors · 2024-11-25
> > > > > > > **Further Response to Reviewer J6yi (Part II)**
> > > > > > >
> > > > > > > > On the general applicability of the tokenization scheme
> > > > > > >
> > > > > > > - We appreciate the suggestion of using positional embeddings as an alternative. However, we would like to emphasize why our proposed tokenization scheme is designed to ensure general applicability and flexibility, and why it is better suited for leveraging the rapidly evolving capabilities of modern LMs.
> > > > > > > - One of the core motivations of our work is to design a model-agnostic tokenization scheme that operates solely at the input data level. This ensures that Geo2Seq can be applied to any LM. In contrast, using positional embeddings for coordinates requires model-level modifications, which may include injecting additional positional encoding layers and are infeasible for modern LMs released as APIs. Moreover, we emphasize the practical efficiency of our method compared to designing custom positional embeddings for each architecture. For example, it is not straightforward to design such SE(3)-invariant positional embeddings for an SSM architecture such as Mamba.
> > > > > > > - Finally, while we agree some potential viability of using positional embeddings as an alternative, **we do not think this undermines our contributions or constitutes a weakness of our paper**. Geo2Seq was specifically designed to address the limitations of existing methods in a model-agnostic and practical manner, enabling seamless integration with a wide range of LMs, including those that are not accessible for in-layer modification.
> > > > > > >
> > > > > > > We believe our work would be beneficial to the 3D molecular community and would genuinely appreciate your recognition. We hope we have shown the significance of our work through all the discussions. Since the discussion period is quickly approaching the end, we sincerely hope you can let us know if we have fully addressed your concerns and reevaluate our work if we have. Also, we welcome any additional discussions or feedback. Thank you!
> > > > > > >
> > > > > > > Sincerely,
> > > > > > >
> > > > > > > Authors

---

> > > > > > > > ### Author Response · Authors · 2024-12-02
> > > > > > > > **Follow-up with Reviewer J6yi**
> > > > > > > >
> > > > > > > > Dear Reviewer J6yi,
> > > > > > > >
> > > > > > > > Thank you again for your valuable additional comments, which helps to improve our work a lot.
> > > > > > > > Since the discussion period is quickly approaching the end, we sincerely hope you can let us know if we have fully addressed your concerns and reevaluate our work if we have.
> > > > > > > > Also, please let us know if there are any additional questions or feedback. Thank you!
> > > > > > > >
> > > > > > > > Sincerely,
> > > > > > > >
> > > > > > > > Authors

---

> > > > > > > > > ### Comment · Reviewer_J6yi · 2024-12-03
> > > > > > > > > **Concluding remarks**
> > > > > > > > >
> > > > > > > > > The architecture-agnostic molecular conformation generation capabilities of the proposed representation could be valuable contributions.
> > > > > > > > > However, my concerns that the experimental setup put the method in an overly positive daylight have not been sufficiently addressed.
> > > > > > > > >
> > > > > > > > > Although the evaluation is in line with prior work, evaluating generative models in a meaningful way is known to be difficult and it is easy to overfit commonly used metrics [1].
> > > > > > > > > Therefore, I believe this paper would benefit from a more thorough revision that provides a more solid experimental evaluation, using more robust metrics (e.g. [2]) to reinforce the claims concerning overfitting, addressing the effects of vocabulary size (and corresponding model size and capacity), and considering alternative embedding approaches (e.g. using special tokens for rotational and translational values instead of digits).
> > > > > > > > > This is why I decided to keep my score.
> > > > > > > > >
> > > > > > > > > ###### References
> > > > > > > > > [1] Renz et al. On Failure Modes of Molecule Generators and Optimizers. ChemRxiv (2020) doi:10.26434/chemrxiv.12213542.v1.
> > > > > > > > > [2] Preuer et al. Fréchet ChemNet Distance: A Metric for Generative Models for Molecules in Drug Discovery (2018)  Journal of chemical information and modeling, 58(9), 1736-1741.

---

> > > > > > > > > > ### Author Response · Authors · 2024-12-03
> > > > > > > > > > **Further Response to Reviewer J6yi**
> > > > > > > > > >
> > > > > > > > > > Dear Reviewer J6yi,
> > > > > > > > > >
> > > > > > > > > > Thank you for your reply. We appreciate your recognition that architecture-agnostic molecular conformation generation capabilities of our proposed representation is valuable contributions, and would like to respond to your remaining concern as follows.
> > > > > > > > > >
> > > > > > > > > > > Concern: more solid experimental evaluation
> > > > > > > > > >
> > > > > > > > > > - We would like to point out that other than the metrics following EDM, **we have already included many more chemical metrics in the original paper, including Fréchet ChemNet Distance [2], in Appx. D.3, as presented in lines 1404-1435 of the paper**. We agree that evaluating generative models is complex, and thus we have provided further evaluation results on various chemical metrics in Appx. D, **Tables 6, 7, 8, 9, 10, and 11**, including bond lengths and steric hindrance. Vast results show that we consistently outperform existing methods across metrics regarding various chemical constraints, which is **surely not achievable through overfitting**. We believe that our experiments have provided sufficiently comprehensive evaluations in support of the solidity of our methodology. We welcome additional discussion on other meaningful metrics you have in mind.
> > > > > > > > > > - Regarding the `effects of vocabulary size`, in our [previous experiments](https://openreview.net/forum?id=HbZrxBXzks&noteId=ruZy5qoxKy) we have shown that across the different tokenization methods, performances are largely influenced by tokenization rather than the absolute vocabulary size. And even with smaller vocabulary size, our adopted Comp-tokenization performs significantly better than BPE, which clearly shows that our performance gain is not from overfitting but from the model’s better understanding of information. Regarding `alternative embedding approaches`, Geo2Seq has achieved SE(3)-invariance representation in a model-agnostic and practical manner, and we appreciate your recognition that it is valuable contributions. We focus on the scope of our paper and agree that alternative embedding approaches can be an interesting direction of future explorations.
> > > > > > > > > >
> > > > > > > > > > > Summary: Novelty and Significance
> > > > > > > > > >
> > > > > > > > > > Finally, we would like to demonstrate the contributions and distinctions of our paper clearly:
> > > > > > > > > > 1. **The track is under-explored:** Different from mainstream methods, we convert 3D molecules to 1D discrete sequences and generate using LMs for 3D molecule generation tasks, positioning in an under-explored field.
> > > > > > > > > > 2. **Technique distinctions:** The only comparable work using LMs applies coordinates from XYZ files directly. In contrast, Geo2Seq achieves **structural completeness** and **geometric invariance** for **LM training**, which is distinct from all existing works.
> > > > > > > > > > 3. **Flexibility & comprehensive architecture:** Geo2Seq operates solely on the input data, which allows independence from model architecture and training techniques and provides reuse flexibility. To further demonstrate the universal applicability of our method, we not only include conventional Transformer architecture of CLMs, but also employ SSMs as LM backbones.
> > > > > > > > > > 4. **Effectiveness & Efficiency:**  Results under both architectures prove the effectiveness of Geo2Seq. In addition, compared to diffusion models, our method outperforms them with significantly higher efficiency, *i.e.*, 100x faster generation speed.
> > > > > > > > > >
> > > > > > > > > > We believe our work would be beneficial to the 3D molecular community and would genuinely appreciate your recognition.
> > > > > > > > > > We sincerely thank you for your time and valuable additional comments, which helps to improve our work a lot! Hope we have addressed your concerns through practical efforts and shown the contributions and significance of our work, and we hope you can consider reevaluating our work if we have. Also, we welcome additional feedbacks of any form. Thank you!
> > > > > > > > > >
> > > > > > > > > > Sincerely,
> > > > > > > > > >
> > > > > > > > > > Authors
> > > > > > > > > >
> > > > > > > > > >
> > > > > > > > > >  [2] Preuer et al. Fréchet ChemNet Distance: A Metric for Generative Models for Molecules in Drug Discovery (2018) Journal of chemical information and modeling, 58(9), 1736-1741.

---

### Meta-Review · Area_Chair_QXho · 2024-12-19

**Metareview:**

**Summary**
This work proposes a geometry-informed tokenization method Geo2Seq to enable 3D molecule generation with language models (LMs). Specifically, Geo2Seq employs canonical labeling and invariant spherical representations to transform, reversibly, 3D structures into SE(3)-invariant sequences, which can be fed into LMs. Experimental validation is provided on GEOM-Drugs and QM9 datasets.

**Strengths**
The reviewers appreciated several merits of this work including clarity of presentation, flexibility of the method in terms of being agnostic to the choice of LM, solid theoretical arguments, and the empirical performance.

**Weaknesses**
Some reviewers raised concerns about the novelty of the approach (while disagreeing with others), pointing out that canonical labeling and invariant spherical coordinates have already been extensively studied in the prior work, and the submission spent disproportionate space on related work and theoretical results that they felt were rather straightforward did not provide any new or interesting insights.

During the author-reviewer discussion period and the follow-up with AC, some of the reviewers also raised concerns about the possibility that the improvement in results might be due to memorization in spherical coordinates and/or overparameterization compared to the much smaller diffusion-based models. They also attributed the performance improvements due to using decimals in tokens, and the apparent arbitrariness in the choice of decimal precision, as factors that might help tailor to the specific datasets, potentially limiting the generalizability of the findings. One of these reviewers mentioned that while their concerns were addressed to some extent, they were not fully resolved;  another was not convinced by the response at all.

Two of the reviewers specifically expressed dissatisfaction with a perfect 10 score by one of the reviewers, arguing that their review did not provide substantial arguments to support such a high score and overly inflated the perceived impact of this paper.

**Recommendation**
During the rebuttal period, the authors provided additional experimental evidence and clarifications that were largely appreciated by the reviewers. I commend the authors for investing time and effort in preparing thoughtful responses.

Noting the discrepancy in reviewers’ perception of this work, the AC decided to carefully review the work.  Despite the merits of this work, I agree with the dissenting reviewers about their concerns regarding the (methodological) novelty of this work, and believe that further empirical investigations are required to substantiate this work.

Notably, there is a large body of prior work, beginning with [1], on using language models for generating  (macro-)molecules such as proteins given the 3D-structure.  In fact, [1] proposes an efficient Transformer-based approach that uses a quaternion-based representation, guaranteeing SE(3)-invariance and reconstruction of the local neighborhoods (up to isomorphism). See also [2] for a more recent approach that follows [1] to incorporate symmetries.

A comprehensive empirical comparison with  [1] and [2], adapting them to the current setting (by leveraging their reconstructions), would address the concerns about the fairness of evaluation.  I also recommend that the authors reposition (the novelty of) their contributions accordingly.

In the current form, however, the work does not meet the bar for acceptance at ICLR.

[1] Ingraham et al. Generative models for graph-based protein design. NeurIPS 2019.

[2] Dauparas et al. Robust deep learning-based protein sequence design using ProteinMPNN. Science 2022.

**Additional Comments On Reviewer Discussion:**

Please see above for all the relevant information.

---

### Decision · Program_Chairs · 2025-01-22

Reject